# BLOCK-WISE ADAPTIVE CACHING FOR ACCELERATING DIFFUSION POLICY

**Kangye Ji[1], Yuan Meng[2][†], Hanyun Cui[1][*], Ye Li[1], Jianbo Zhou[1][*], Shengjia Hua[1][*]**
**Lei Chen[1], Zhi Wang[1][†]**
[1]Tsinghua Shenzhen International Graduate School, Tsinghua University
[2]Department of Computer Science and Technology, Tsinghua University

## ABSTRACT

Diffusion Policy has demonstrated strong visuomotor modeling capabilities, but its high computational cost renders it impractical for real-time robotic control. Despite huge redundancy across repetitive denoising steps, existing diffusion acceleration techniques fail to generalize to Diffusion Policy due to fundamental architectural and data divergences. In this paper, we propose **B**lock-wise **A**daptive **C**aching (**BAC**), a method to accelerate Diffusion Policy by caching intermediate action features. BAC achieves lossless action generation acceleration by adaptively updating and reusing cached features at the block level, based on a key observation that feature similarities exhibit non-uniform temporal dynamics and distinct block-specific patterns. To operationalize this insight, we first design an Adaptive Caching Scheduler to identify optimal update timesteps by maximizing the global feature similarities between cached and skipped features. However, applying this scheduler for each block leads to significant error surges due to the inter-block propagation of caching errors, particularly within Feed-Forward Network (FFN) blocks. To mitigate this issue, we develop the Bubbling Union Algorithm, which truncates these errors by updating the upstream blocks with significant caching errors before downstream FFNs. As a training-free plugin, BAC is readily integrable with existing transformer-based Diffusion Policy and vision-language-action models. Extensive experiments on multiple robotic benchmarks demonstrate that BAC achieves up to $3\times$ inference speedup for free. Project page: https://block-wise-adaptive-caching.github.io.

## 1 INTRODUCTION

Diffusion Policy has gained substantial attention in robotic control, due to its ability to model action distributions via conditional denoising processes (Chi et al., 2023). Recently, it has also been widely adopted by vision-language-action models (Wen et al., 2025; Liu et al., 2025b; Hou et al., 2025) to perform highly dexterous and complex tasks. However, its massive computational burden in the denoising process makes the action frequency unable to satisfy real-time and smooth control. For instance, on a 6-DoF robotic arm executing block pick-and-place, 50 diffusion denoising steps at 1 ms per step restrict the action update rate to 10 Hz, well below 30–50 Hz usually needed for smooth real-time control (Shih et al., 2023).

Despite the aforementioned necessity, the acceleration of Diffusion Policy remains an underexplored field. Cache-based methods have recently gained significant attention in accelerating diffusion models on image-generation tasks (Ma et al., 2024b; Wimbauer et al., 2024; Selvaraju et al., 2024; Chen et al., 2024; Zou et al., 2025) and video-generation tasks (Liu et al., 2024; Kahatapitiya et al., 2024; Lv et al., 2024). However, they cannot be directly applied to Diffusion Policy, due to differences in data characteristics and model architectures.

To address this issue, we aim to propose a customized feature caching method for Diffusion Policy. We first explore the distinct characteristics of Diffusion Policy models and identify two key observations

---

[†]Corresponding authors
[*]Work performed during internship at MMLab@SIGS, Tsinghua University

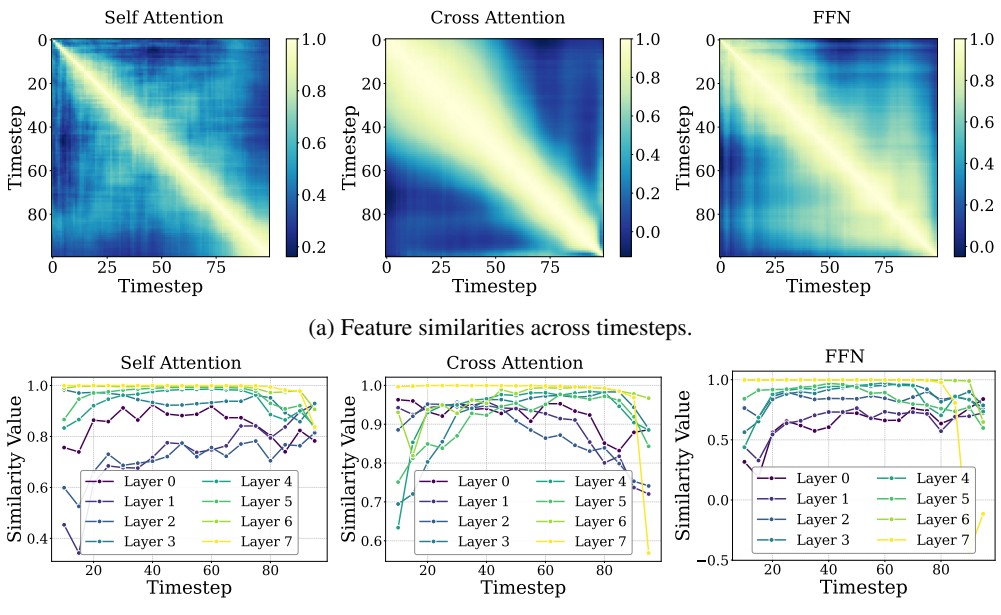

(a) Feature similarities across timesteps.

(b) Similarity change curves, measured between timestep $t$ and $t-10$.

Figure 1: Temporal and block-wise feature similarity patterns. (a) Similarity matrices of blocks in the third decoder layer. (b) Similarity change curves of different blocks. The feature similarity between consecutive timesteps varies non-uniformly over time and differs across blocks.

in feature similarities: (1) feature similarities across timesteps vary non-uniformly, and (2) different blocks exhibit distinct temporal similarity patterns as shown in Fig. 1. Motivated by this observation, we propose **B**lock-wise **A**daptive **C**aching (**BAC**), a training-free method that accelerates transformer-based Diffusion Policy by adaptively updating and reusing cached action features at the block level. BAC integrates an Adaptive Caching Scheduler (ACS) to allocate block-specific caching schedules and the Bubbling Union Algorithm (BUA) to truncate inter-block error propagation.

Specifically, the Adaptive Caching Scheduler aims to identify a set of cache update timesteps that maximize the global feature similarities between cached and skipped features. However, directly searching this set within an exponential search space is unacceptable. To address this challenge, we reformulate the problem as a dynamic programming optimization, where the global similarity serves as the objective and the block-specific similarity matrix defines the scores. Leveraging the high episode homogeneity within a single task, the scheduler computes once before inference, incurring virtually no additional cost.

While the Adaptive Caching Scheduler effectively determines update timesteps, extending the scheduler to the block level can trigger significant error surges, leading to performance collapse. We examine this problem theoretically and experimentally and attribute this failure to inter-block caching error propagation: FFN blocks introduce the caching errors from upstream blocks during their updates, due to the lack of intermediate normalization. To truncate the error propagation, we propose the Bubbling Union Algorithm, which first selects the upstream blocks with large caching errors and then enforces them to update their cache if downstream FFNs do.

Our main contributions are as follows:

1. We propose Block-wise Adaptive Caching, a training-free acceleration method for transformer-based Diffusion Policy, which adaptively updates and reuses cached features at the block level.

2. We develop the Adaptive Caching Scheduler that optimally determines cache update timesteps by maximizing the global feature similarity with a dynamic programming solver.

3. We design the Bubbling Union Algorithm to further extend the caching schedule to the block level by truncating inter-block caching error propagation, based on the theoretical and empirical analysis of the error surge phenomenon in Diffusion Policy.

4. We conduct extensive robotic experiments to evaluate our method. The results demonstrate that our method efficiently boosts Diffusion Policy by $3\times$ for free.

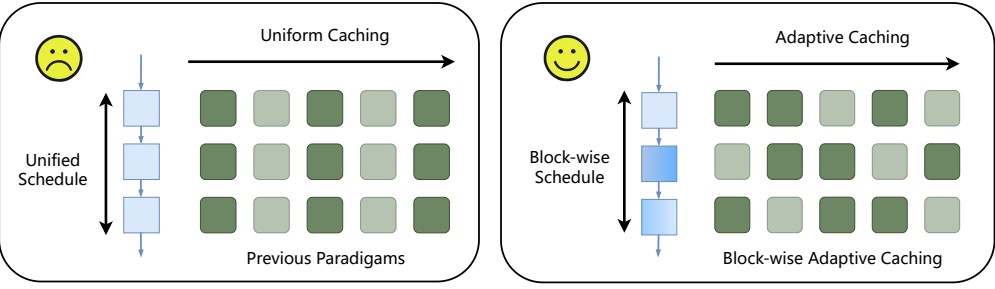

Figure 2: Comparison of Block-wise Adaptive Caching and previous caching paradigms.

## 2 RELATED WORK

### 2.1 DIFFUSION POLICY

Diffusion models, initially developed for image generation (Sohl-Dickstein et al., 2015; Ho et al., 2020; Esser et al., 2024; Bar-Tal et al., 2024), have been adapted for robot policy learning (Martinez-Cantin et al., 2007). Within the Diffusion Policy framework (Chi et al., 2023), both U-Net (Ronneberger et al., 2015) and Diffusion Transformer (DiT) (Peebles & Xie, 2023; Yuan et al., 2024; Zhao et al., 2024) denoisers are supported, enabling scalable backbone designs. Recent VLA methods (Brohan et al., 2023; Kim et al., 2024; Wen et al., 2024; 2025) increasingly adopt Transformer-based denoisers for stronger expressivity, but the iterative denoising steps induce significant inference latency, motivating acceleration techniques (Xu et al., 2025; Song et al., 2025). Detailed discussion is provided in Appendix A.2.

### 2.2 DIFFUSION MODELS CACHING

Despite the success of cache-based methods for diffusion models, their adaptation to Diffusion Policy remains underexplored. Existing caching methods primarily target U-Net-based diffusion models (Ma et al., 2024b; Wimbauer et al., 2024). For example, DeepCache (Ma et al., 2024b) exploits the temporal redundancy inherent in U-Nets by caching high-level feature representations. Nevertheless, these methods cannot be generalized to transformer backbones. Recently, some methods (Selvaraju et al., 2024; Ma et al., 2024a; Chen et al., 2024; Zou et al., 2025) explore the caching mechanism in transformer-based diffusion models. These methods typically operate at a coarse granularity, with all the blocks sharing a uniform caching schedule (Selvaraju et al., 2024; Ma et al., 2024b), i.e., updating the cache at uniform intervals. Despite some works extending this schedule in a finer architectural granularity, they either require extra training (Ma et al., 2024a) or are specifically designed for the patterns of the image generation process (Chen et al., 2024).

## 3 BLOCK-WISE ADAPTIVE CACHING

As illustrated in Fig. 3, BAC achieves a finer-grained cache schedule by first applying the Adaptive Caching Scheduler to compute optimal update timesteps for each block and then employing the Bubbling Union Algorithm to truncate inter-block error propagation. In this section, we first present the preliminaries in Sec. 3.1. Next, we introduce the Adaptive Caching Scheduler in Sec. 3.2. To extend the scheduler to the block level, we analyze the error surge phenomenon in Sec. 3.3 and describe the Bubbling Union Algorithm in Sec. 3.4.

### 3.1 PRELIMINARIES

**Diffusion Policy.** Diffusion Policy treats robot visuomotor control as sampling from a conditional denoising diffusion model (Chi et al., 2023). At each time step $t$, we first draw an initial noisy action $\mathbf{a}_t^{(K)}$ from a standard Gaussian prior and then apply $K$ learned reverse-diffusion steps:

$$\mathbf{a}_t^{(k-1)} = f_\theta\big(\mathbf{a}_t^{(k)}, \mathbf{o}_{1:t}, k\big), \quad k = K, K-1, \ldots, 1, \tag{1}$$

where $f_\theta$ parameterizes the conditional reverse kernel (i.e. the denoiser) and the final sample $\mathbf{a}_t^{(0)}$ is used as the control action.

**Diffusion Transformer (DiT).** The DiT architecture in Diffusion Policy utilizes an MLP to encode observation embeddings, which are then passed into a transformer-based decoder. The decoder consists of $L$ layers, where each layer $l$ contains a cross-attention (CA) block that conditions on timesteps and observations, a self-attention (SA) block, and a feed-forward network (FFN) block. For a given input $\mathbf{h}_k^{(l-1)}$ at denoising step $k$, the output of layer $l$ is computed by summing the residual outputs of these blocks:

$$\mathbf{h}_k^{(l)} = \mathbf{h}_k^{(l-1)} + \text{SA}_k^{(l)} + \text{CA}_k^{(l)} + \text{FFN}_k^{(l)}. \tag{2}$$

**Problem Formulation.** To reduce redundant computations across timesteps in the denoising process of diffusion models, cache-based methods reuse intermediate features to skip repeated computations partially. Following existing caching methods (Ma et al., 2024b; Selvaraju et al., 2024), we adopt an update-then-reuse paradigm.

Let $\mathbf{b}_k$ denote the output of a target block at step $k$. A caching mechanism defines a set of update steps $\mathcal{C} \subseteq \{1, \ldots, K\}$, where:

- The update step: If $k \in \mathcal{C}$, computes $\mathbf{b}_k$ and updates its cached features.
- The reuse step: The block reuses the cached feature $\mathbf{b}_{k'}$, which is computed in the most recent update step $k' = \min\{i \in \mathcal{C} \mid i > k\}$.

Following prior work (Zou et al., 2025; Selvaraju et al., 2024; Ma et al., 2024b), we construct a baseline in which all blocks share a unified caching schedule, with a fixed update interval $\mathcal{C}$ (e.g., updating the cache every three timesteps). BAC aims to improve upon this baseline by allocating an optimal $\mathcal{C}^*$ for each block.

### 3.2 ADAPTIVE CACHING SCHEDULER

**Optimization Objective.** In this work, we use cosine similarity to measure the similarity between features due to its superior performance in measuring directional consistency between high-dimensional feature vectors. The consecutive similarity is calculated as:

$$s_k = \cos(\mathbf{b}_k, \mathbf{b}_{k-1}) = \frac{\mathbf{b}_{k-1}^\top \mathbf{b}_k}{\|\mathbf{b}_{k-1}\|_2 \|\mathbf{b}_k\|_2}, \quad k = 1, \ldots, K, \tag{3}$$

We define the interval similarity between timesteps $i$ and $j$ as $\phi(i, j) = \sum_{k=i+1}^{j} s_k$. A larger $\phi(i, j)$ indicates lower caching errors incurred by reusing the cached feature $b_j$ over the interval $[i, j]$. The value function is then:

$$\max_{\substack{\mathcal{C} \subseteq \{1, \ldots, K\} \\ |\mathcal{C}| = M}} \sum_{m=0}^{M} \phi(c_m, c_{m+1} - 1), \quad \text{with} \quad c_0 = 0, \quad c_{M+1} = K, \tag{4}$$

where $c_0$ and $c_{M+1}$ are boundary conditions representing the start step and the end step of the path.

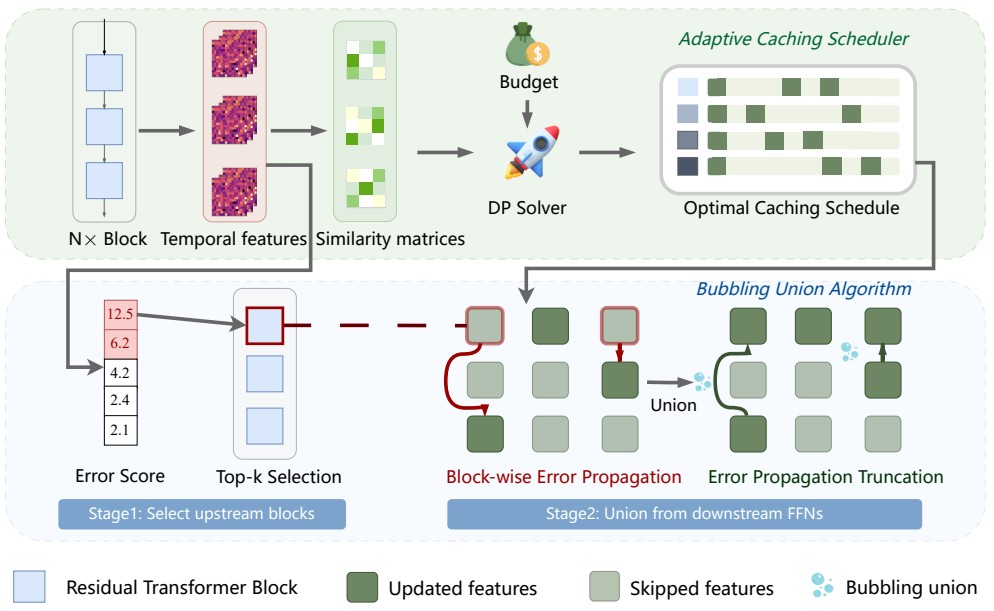

Figure 3: Framework of Block-wise Adaptive Caching (BAC). BAC enables adaptive feature caching by introducing the Adaptive Caching Scheduler, and further supports block-wise scheduling through the Bubbling Union Algorithm.

**Optimal Schedule Solver.** The combinatorial nature of selecting $M$ update steps from $K$ timesteps renders exhaustive search computationally infeasible for large $K$. To address this, we design a dynamic programming (DP) solver that efficiently computes the optimal cache schedule.

Define the DP state $\text{DP}[m][j]$ as the maximum cumulative similarity achievable when the $m$-th cache update occurs at timestep $j$:

$$\text{DP}[m][j] = \max \sum_{i=0}^{m} \phi\big(c_i,\ c_{i+1} - 1\big). \tag{5}$$

The corresponding state transition equation is given by:

$$\text{DP}[m][j] = \max_{0 \le i < j} \left\{ \text{DP}[m-1][i] + \phi(i, j) \right\}, \tag{6}$$

To recover the optimal update schedule $\mathcal{C}^*$ from the DP table, we introduce a pointer matrix:

$$\text{PTR}[m][j] = \arg \max_{0 \le i < j} \left\{ \text{DP}[m-1][i] + \phi(i, j) \right\}, \quad m = 1, \ldots, M,\ j = 1, \ldots, K. \tag{7}$$

Once both DP table and PTR table are filled, we dertermine the final endpoint as:

$$j^* = \arg \max_{1 \le j \le K} \text{DP}[M][j], \tag{8}$$

and backtrack from $j^*$ to reconstruct the full update schedule:

$$c_M^* = j^*, \quad c_{m-1}^* = \text{PTR}[m][c_m^*], \quad m = M, \ldots, 1. \tag{9}$$

The solved optimal update step set is given by $\mathcal{C}^* = \{c_1^* < \cdots < c_M^*\}$. Adaptive Caching Scheduler maximizes the performance efficiency trade-off by computing $\mathcal{C}^*$ for each block under the given computation budget.

### 3.3 THE ERROR SURGE PHENOMENON AND ANALYSIS.

However, purely extending the Adaptive Caching Scheduler to the block level can trigger significant error surges, particularly within FFN blocks. We provide a detailed analysis of this failure mode in the following section and introduce our remedy Bubbling Union Algorithm in Sec. 3.4.

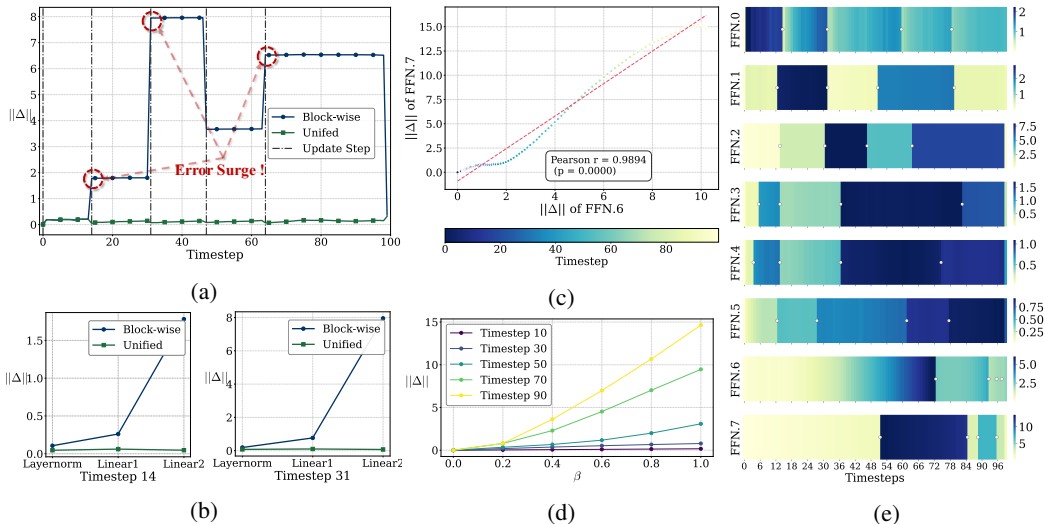

Figure 4: (a) Caching error of the third FFN block when updated using the block-wise versus a unified schedule. (b) Caching error of sub-layers within the third FFN block at update timesteps 14 and 31. (c) Correlation between the caching errors of the seventh and eighth FFN blocks. (d) Update-induced error under varying input error magnitudes, controlled by the scaling factor $\beta$. (e) Caching error across all blocks throughout the diffusion process, with white dots indicating update steps. The experiments are conducted on the Square task.

**Identifying Error Surge.** Extending Adaptive Caching Scheduler to the block level leads to unexpected performance collapse. Our observation uncovers a surprising phenomenon: instead of reducing errors, block-wise updates amplify them, resulting in sudden error surges in the FFN blocks, as illustrated in Fig. 4a.

Generally, caching errors arise from either feature reuse or feature update. In the reuse case, the error comes from a mismatch between cached features and the shifted ground-truth distribution. In the update case, the error results from inaccurate inputs caused by errors from upstream blocks. We observe that error surges often occur during update steps of FFN blocks, where update-induced errors exceed reuse-induced errors, indicating a failure in the update process. We first elucidate how FFN blocks incorporate these upstream errors during updates, then delineate the complete inter-block error propagation process.

**Error Propagation in FFN Blocks.** To understand how FFN blocks incorporate the upstream errors, we begin by formalizing the error propagation process. Let

$$\text{FFN}(X) = W_{\text{out}}\phi(W_{\text{in}}\text{LN}(X) + b_1) + b_2, \tag{10}$$

**Proposition 3.1.** *Given an upstream error $\delta$, we have*

$$\Delta = W_{\text{out}} \, \text{diag}\big(\phi'(U)\big) \, W_{\text{in}} \, (A - B) \, \delta + O(\|\delta\|^2), \tag{11}$$

*where*

$$A = \frac{\text{diag}(\gamma) \left(I - \frac{1}{d}\mathbf{1}\mathbf{1}^\top\right)}{\sigma(X)}, \quad B = \frac{\text{diag}(\gamma) \left(X - \mu(X)\mathbf{1}\right)(X - \mu(X)\mathbf{1})^\top}{d \, \sigma(X)^3}. \tag{12}$$

To further analyze the correlation between inter-block errors, we design a toy experiment where the upstream block (FFN.6) uses only cached activations, while the downstream block (layer.7.FFN) performs full computation. The relationship between the update-induced error and its corresponding upstream error is depicted in Fig. 4c, with a Pearson correlation coefficient of $r = 0.9894$, indicating a strong correlation. To isolate the influence of the timestep, we fix it and use a factor $\beta$ to control the magnitude of the upstream error. The result in Fig. 4d also shows a strong positive correlation between upstream and downstream errors, further confirming the effect of inter-block error propagation.

**Inter-block Error Propagation.** A complete propagation chain is shown in Fig. 4e. When a block updates at a timestep when its upstream block does not update and has a larger caching error, an error surge occurs, visually manifested by a sudden deepening of block colors without any gradual transition. Although the upstream block updates later, the surging error in the downstream block still persists, indicating the failure of this update.

## 3.4 Bubbling Union Algorithm

To truncate inter-block error propagation, we propose a simple yet effective algorithm to revise the original scheduler. The core insight of our algorithm is that if an FFN block updates its cache, its upstream blocks with large errors should also update. Therefore, the updated error $\Delta$ can be mitigated due to the suppressed propagated upstream error $\delta$.

Our algorithm consists of two stages:

**Stage 1: Selecting Upstream Blocks with Large Caching Errors.** To estimate the caching error magnitude for each block $j$, we compute the average of $\ell_1$ norm across features over all pairs of denoising timesteps:

$$\ell_j = \frac{1}{K^2} \sum_{t=1}^{K} \sum_{u=1}^{K} \left\| X_j^{(t)} - X_j^{(u)} \right\|_1. \tag{13}$$

A larger $\ell_j$ indicates that the block has larger reuse-induced errors. We then select the top $n$ blocks with the largest $\ell_j$ and denote this block set by $U$.

**Remark 3.1.** *As discussed in Sec. 3.3, caching error consists of both reuse-induced errors and update-induced errors. We choose not to account for the update-induced error of upstream blocks because it occurs less frequently and is difficult to approximate reliably. Moreover, incorporating it would require treating all FFN blocks as upstream blocks, which would compromise the overall trade-off between efficiency and precision.*

**Stage 2: Unioning Update Timesteps of FFNs from Downstream to Upstream.** Our algorithm truncates error propagation by enforcing that each upstream block in $U$ updates its cache before its downstream FFN blocks. Concretely, let $C(u)$ denote cache update timesteps set of block $u$. Let $D(u)$ be the set of all FFN blocks downstream of block $u$. Then for each $u \in U$, we update $C(u)$ as:

$$C(u) = C(u) \cup \bigcup_{v \in D(u)} C(v) \tag{14}$$

## 4 Experiments

We first outline the experimental setup, covering benchmarks, baselines, implementation details, and metrics in Sec. 4.1. Following that, we evaluate BAC on the real-world and simulation benchmarks in Sec. 4.2 and Sec. 4.3, respectively. Finally, we present an ablation study of BAC in Sec. 4.4.

## 4.1 Experimental Setup

**Benchmarks.** We comprehensively evaluate BAC across standard real-world and simulation benchmarks. For real-world evaluation, we deploy the Diffusion Policy on a Franka Research 3 arm equipped with a UGREEN CM717 RGB camera. The task requires the robot to grasp, pick, and release a soft, deformable bag whose diameter is approximately 80% of the gripper's maximum jaw opening. This setup requires precise temporal coordination to prevent the object from being toppled. For simulation, we utilize DP-T on four robotic manipulation benchmarks: Robomimic in PH/MH data, Push-T, Multimodal Block Pushing, and Kitchen. To test generalization on VLA models, we employ RDT-1B on the ManiSkill benchmark, focusing on four representative tasks, e.g., PegInsertionSide, PickCube, and StackCube.

**Baselines.** We report the result of DP-T as Full Precision and utilize the baseline constructed in Sec. 3.1, in which all blocks update and reuse their cache at uniform intervals simultaneously. We refer to this baseline as *Uniform*. We migrate the cache update strategy from TeaCache (Liu et al.,

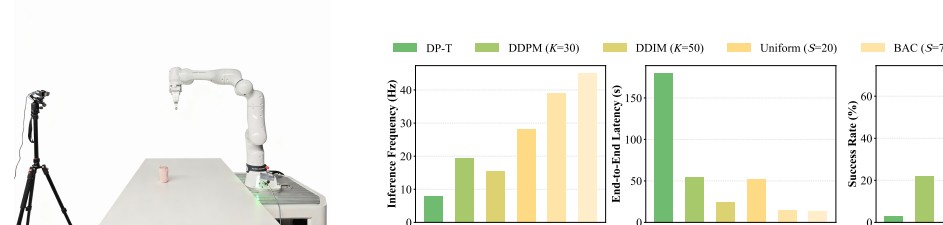

Figure 5: Left: Hardware setup for real-world evaluation. Right: Performance comparisons on inference frequency, end-to-end latency, and success rate.

2025a) and adapt it to our setting, which serves as another baseline, denoted as TeaCache. We also include DDIM as the representative sampling-based method.

**Implementation and Metrics.** BAC is implemented as a training-free plugin. We set the hyperparameter $n = 5$ for simulation tasks and $n = 3$ for real-world experiments. All experiments are conducted on a single NVIDIA GeForce RTX 4090D GPU. We report Success Rate (or target coverage for Push-T) as the primary precision metric, alongside FLOPs and Speedup for efficiency. In real-world settings, we additionally measure Inference Frequency (Hz) and End-to-End Latency to quantify practical speedup.

## 4.2 REALWORLD BENCHMARK

**Quantitative Results.** As shown in Fig. 5, BAC significantly outperforms all baselines in both success rate and inference frequency. With $\mathcal{S} = 7$ and $n = 3$, BAC achieves a 71% success rate at 39.2 Hz. By adopting more aggressive caching ($\mathcal{S} = 5$), BAC reaches a peak inference frequency of 45.1 Hz while retaining a competitive 63% success rate. In contrast, the standard DDPM with $K = 100$ steps yields only a 3% success rate at an impractical 7.8 Hz. Even with acceleration techniques such as DDIM with $K = 50$ or *Uniform* ($\mathcal{S} = 20$), success rates remain limited to 52% and 40%, respectively, due to degraded generation quality.

**Qualitative Analysis.** Fig. 6 illustrates clear behavioral differences across acceleration methods. BAC ($\mathcal{S} = 5$) achieves both high generation fidelity and low latency, enabling smooth and successful task execution. In comparison, baselines exhibit severe failure modes: DDPM with $K = 30$ triggers premature picking and fails to release, while *Uniform* $\mathcal{S} = 20$ stays still due to generated joint poses lying outside the robot's kinematic reachability and subsequently releases too early. Moreover, both baselines suffer from high end-to-end latency of up to 54 seconds, primarily caused by severe observation–action desynchronization. For DDPM methods, inference alone can exceed 900 ms per action chunk, rendering predictions stale by the time they are executed. This lag induces wandering motions and chunk rollbacks, as the controller attempts to reach states that no longer match the current state and observations.

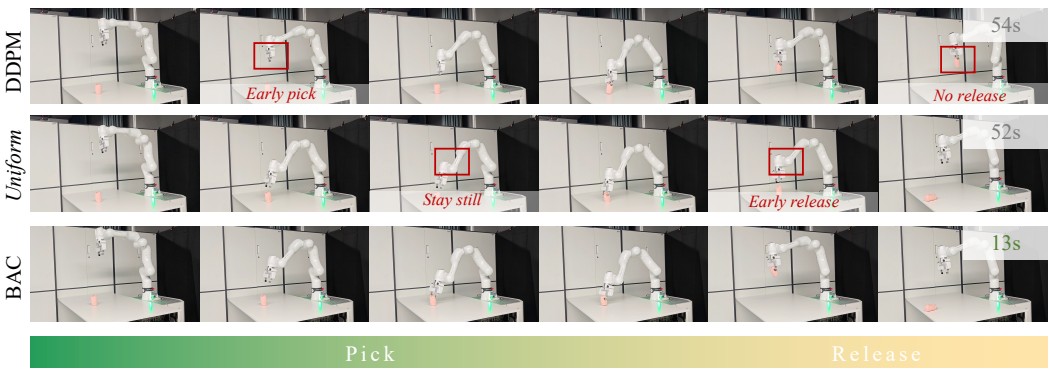

Figure 6: Qualitative results of real-world evaluations from different methods.

Table 1: Results on different benchmarks. We present success rates of different checkpoints in the format of (max performance) / (average of last 10 checkpoints), with each averaged across 3 training seeds. The overall average success rate is denoted as AVG, with average flops and speedup reported as FLOPs and Speed×, respectively.

Benchmark on Proficient Human (PH) demonstration data.

| Method | Success Rate ↑ | | | | | | AVG | FLOPs | Speed × |
|---|---|---|---|---|---|---|---|---|---|
| | Lift | Can | Square | Transport | Tool | Push–T | | | |
| Full Precision | 1.00/1.00 | 0.95/0.97 | 0.82/0.88 | 0.78/0.81 | 0.43/0.53 | 0.59/0.64 | 0.76 | 15.77G | – |
| *Uniform*(fast) | 0.99/1.00 | 0.93/0.96 | **0.86**/0.88 | 0.78/0.77 | 0.39/0.50 | 0.58/0.64 | 0.79 | 3.15G | 2.69 |
| *Uniform*(fastest) | 0.99/1.00 | 0.79/0.95 | 0.73/0.83 | 0.73/0.78 | 0.23/**0.64** | 0.57/**0.65** | 0.76 | 2.72G | 3.20 |
| Teacache(fast) | 0.99/1.00 | **0.97**/0.95 | 0.57/0.82 | **0.79**/0.56 | 0.34/0.23 | **0.65**/0.65 | 0.71 | 3.40G | 2.23 |
| Teacache(fastest) | 1.00/1.00 | 0.96/0.96 | 0.67/0.82 | 0.77/0.52 | 0.44/0.38 | 0.63/0.52 | 0.72 | 2.78G | 3.14 |
| BAC($\mathcal{S} = 7$) | 1.00/1.00 | 0.90/0.95 | 0.78/0.87 | 0.75/0.81 | 0.36/0.47 | 0.57/0.61 | 0.74 | 2.02G | 3.54 |
| BAC($\mathcal{S} = 10$) | **1.00/1.00** | 0.94/**0.97** | 0.82/**0.89** | 0.77/**0.82** | **0.49**/0.55 | 0.59/0.62 | **0.79** | 2.66G | 3.40 |

Benchmark on Mixed Human (MH) demonstration data.

| Method | Success Rate ↑ | | | | AVG | FLOPs | Speed × |
|---|---|---|---|---|---|---|---|
| | Lift | Can | Square | Transport | | | |
| Full Precision | 0.99/1.00 | 0.92/0.97 | 0.76/0.79 | 0.35/0.46 | 0.76 | 15.77G | – |
| *Uniform*(fast) | 0.99/0.99 | 0.91/0.96 | **0.80**/0.75 | 0.24/0.42 | 0.76 | 3.15G | 2.69 |
| *Uniform*(fastest) | 0.95/**1.00** | 0.65/0.92 | 0.73/0.78 | 0.01/0.06 | 0.64 | 2.72G | 3.20 |
| Teacache(fast) | 0.79/0.85 | **0.97**/0.92 | 0.76/0.66 | **0.50**/0.40 | 0.73 | 5.25G | 1.41 |
| Teacache(fastest) | 0.00/0.02 | 0.97/0.90 | 0.26/0.22 | 0.35/0.31 | 0.38 | 2.62G | 3.44 |
| BAC($\mathcal{S} = 7$) | 0.96/0.99 | 0.39/0.79 | 0.56/0.53 | 0.17/0.41 | 0.60 | 2.03G | 3.48 |
| BAC($\mathcal{S} = 10$) | **0.99**/0.98 | 0.95/**0.97** | 0.77/**0.79** | 0.30/**0.46** | **0.77** | 2.64G | 3.41 |

Benchmark on multi-stage tasks. For Block-Pushing, px is the frequency of pushing x blocks into the targets. For Kitchen, px is the frequency of interacting with x or more objects (e.g. bottom burner).

| Method | Success Rate ↑ | | | | | | AVG | FLOPs | Speed × |
|---|---|---|---|---|---|---|---|---|---|
| | $BP_{p1}$ | $BP_{p2}$ | $Kit_{p1}$ | $Kit_{p2}$ | $Kit_{p3}$ | $Kit_{p4}$ | | | |
| Full Precision | 0.98/0.98 | 0.98/0.96 | 1.00/1.00 | 0.98/1.00 | 0.97/1.00 | 0.95/0.97 | 0.98 | 15.77G | – |
| *Uniform*(fast) | 1.00/1.00 | 0.97/**0.97** | 0.97/**1.00** | 0.93/**1.00** | 0.91/0.99 | 0.79/0.93 | 0.96 | 3.15G | 2.85 |
| *Uniform*(fastest) | 0.99/0.99 | 0.95/0.95 | 0.66/0.89 | 0.42/0.79 | 0.29/0.63 | 0.08/0.34 | 0.66 | 2.72G | 3.34 |
| Teacache(fast) | 0.33/0.33 | 0.33/0.49 | 0.75/0.76 | 0.24/0.18 | 0.27/0.13 | 0.00/0.00 | 0.52 | 1.82G | 4.42 |
| Teacache(fastest) | 0.33/0.33 | 0.00/0.00 | 0.75/0.76 | 0.24/0.18 | 0.27/0.13 | 0.00/0.00 | 0.25 | 1.80G | 4.46 |
| BAC($\mathcal{S} = 7$) | 0.99/0.99 | 0.91/0.93 | 0.80/0.85 | 0.61/0.69 | 0.45/0.57 | 0.23/0.39 | 0.69 | 1.92G | 3.78 |
| BAC($\mathcal{S} = 10$) | **1.00/0.99** | **0.97**/0.95 | **1.00**/0.99 | **1.00**/0.99 | **1.00/0.99** | **0.94/0.97** | **0.98** | 2.44G | 3.60 |

## 4.3 Simulation Benchmark

**Acceleration on Diffusion Policy.** We compare BAC against the baselines across multiple simulation benchmarks in Table 1. We control the computation budget by setting the number of cache update steps $\mathcal{S}$. BAC achieves lossless acceleration with $\mathcal{S} = 10$ on all the benchmarks, with an average success rate of 0.79, 0.77, 0.98 versus 0.76, 0.76, 0.98 for the full-precision DP-T, even exhibiting a modest improvement. BAC consistently achieves stable acceleration rates above $3.4\times$ across most of the tasks. Compared with *Uniform* and Teacache, BAC has two advantages: (1) BAC improves the success rate significantly on the hard tasks such as Kitchen$_{p4}$, where *Uniform* and Teacache fail to

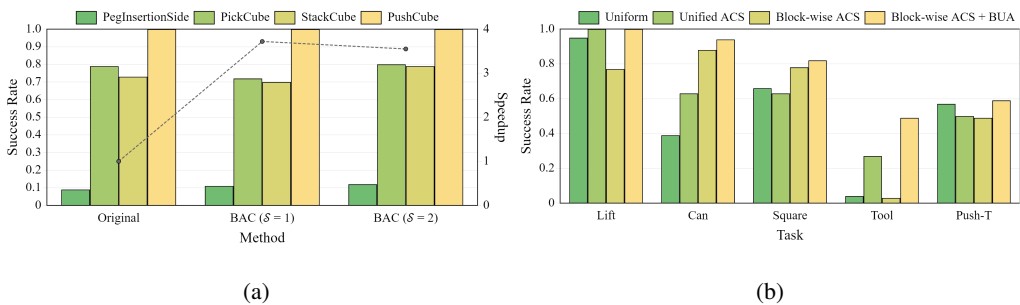

(a)                                                      (b)

Figure 7: (a) Performance of BAC on RDT-1B. (b) Ablation study on BAC.

restore the correct action generation. (2) BAC maintains a strong and stable performance in all tasks, demonstrating the reliability of a lossless acceleration plugin. We attribute this to the ability of BAC to reduce the reuse-induced error by ACS and precisely avoid the update-induced errors by BUA.

**Acceleration on VLA.**   As summarized in Fig. 7a, BAC achieves up to **3.55× acceleration** with a negligible performance drop compared to the original RDT-1B (Liu et al., 2025c) at $\mathcal{S} = 2$, while maintaining lossless performance at $\mathcal{S} = 1$. These results demonstrate that BAC delivers significant speed gains even when combined with DPMSolver, providing strong evidence of its robust generalization across Diffusion Policy-based VLA models.

## 4.4    ABLATION STUDY

**Ablation Study Methods.**   We consider three variants of our caching method. To evaluate the effectiveness of ACS, we build **Unified ACS**, the Adaptive Caching Scheduler is applied solely to the self-attention block in layer 0, which is the very first block in the decoder. The computed update steps are then used by every block. To evaluate the effectiveness of BUA, we build **Block-wise ACS**, the scheduler is naively applied to each block, producing a distinct set of update steps for all blocks. Finally, in **Block-wise ACS + BUA**, we first compute block-wise update steps via the Adaptive Caching Scheduler and then integrate the Bubbling Union Algorithm, yielding our full BAC method. Results of these methods are presented in Fig. 7b.

**Effectiveness of ACS.**   Experiments with the Unified ACS schedule demonstrate a clear performance improvement over *Uniform*. This result confirms the necessity and effectiveness of reducing the reuse-induced error.

**Effectiveness of BUA.**   The performance of Block-wise ACS unexpectedly falls below that of the Unified ACS, empirically substantiating the Error Surge Phenomenon. Integrating BUA into the block-wise schedule recovers full-precision performance across all tasks with the highest score of 0.79, demonstrating the effectiveness of the Bubbling Union Algorithm.

## 5    CONCLUSION

In this paper, we propose BAC, a novel training-free acceleration method for transformer-based Diffusion Policy. BAC minimizes the caching error by adaptively scheduling cache updates through the Adaptive Caching Scheduler. Moreover, we conduct theoretical and empirical analysis on the error surge phenomenon due to inter-block error propagation, and propose the Bubbling Union Algorithm to truncate the propagation. Extensive experiments demonstrate that BAC achieves substantial speedups without performance degradation, typically exceeding 3× compared to full computation.

## 6 ETHICS STATEMENT

In this work, all experiments are based on publicly available robotic simulation datasets, without involving new human subjects, personal data, or sensitive information. Our method focuses solely on improving computational efficiency and raises no concerns of privacy infringement, discrimination, bias, or legal non-compliance.

## 7 REPRODUCIBILITY STATEMENT

We have made extensive efforts to ensure reproducibility of our results. All datasets used in this work are publicly available robotic simulation benchmarks (RoboMimic, Push-T, Kitchen, Block-Pushing, ManiSkill), with details of preprocessing and task setups provided in Appendix A.7. The proposed Block-wise Adaptive Caching (BAC) algorithm is fully described in Section 3, with implementation details and hyperparameters reported in Section 4.1. Complete theoretical assumptions and proofs, including Proposition 3.1, are presented in Appendix A.4. Additional ablation studies, parameter sensitivity analyses, and visualizations are included in the supplemental material (Appendix A.7, A.8, A.9, A.12, A.13) to further support reproducibility. The full implementation and instructions will be released on an open-source GitHub repository to enable replication of all experiments.

## 8 ACKNOWLEDGEMENT

This work is supported by the National Natural Science Foundation of China (Grant No. 92467204, 62472249, and 62402264), and the Shenzhen Science and Technology Program (Grant No. JCYJ20220818101014030 and KJZD20240903102300001).

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

# A APPENDIX / SUPPLEMENTAL MATERIAL

In this supplemental material, we provide declaration of LLMs uses in Appendix A.1, detailed background on diffusion-based VLA models in Appendix A.2, limitation of our work in Appendix A.3, complete proofs of Proposition 3.1 in Appendix A.4, discussion on cause of error surge phenomenon in Appendix A.5, more details on the benchmark in Appendix A.6, along with additional experiments examining the choice of parameter $n$ in Appendix A.7. We further include ablation studies analyzing different metrics in Appendix A.8, and present more temporal similarity figures in Appendix A.9. Additionally, Appendix A.10 contains illustrations demonstrating episode homogeneity within individual tasks, while Appendix A.11 offers supplementary visualizations supporting Figure 3. In Appendix A.12 and Appendix A.13, we present the update steps obtained under $S = 10$, $n = 5$, while using cosine as metric via BAC, showing the update steps obtained after employing ACS and BUA, respectively. In Appendix A.14, we provide detailed visualizations and analysis of the cache behavior. We provide large-scale evidence showing that episodes within the same task exhibit strong homogeneity in Appendix A.15. Finally, we demonstrate that BAC provides additional speedup even when applied on top of the DDIM baseline in Appendix A.16.

## A.1 THE USE OF LARGE LANGUAGE MODELS (LLMS)

In this work, Large Language Models (LLMs) were used solely to polish the language for clarity and readability. No LLMs were employed for idea generation, experimental design, data analysis, or any other part of the research process.

## A.2 BACKGROUND ON DIFFUSION-BASED VLA MODELS

Diffusion models (Esser et al., 2024; Bar-Tal et al., 2024) were originally proposed for image generation (Sohl-Dickstein et al., 2015; Ho et al., 2020) and have been adapted for robot policy learning (Martinez-Cantin et al., 2007). Traditionally, diffusion-based vision–language–action (VLA) (Ma et al., 2024c; Brohan et al., 2023; Kim et al., 2024) methods have depended on U-Net (Ronneberger et al., 2015) based denoising backbones borrowed directly from image generation pipelines to model multimodal action distributions and ensure stable training. Within the Diffusion Policy framework (Chi et al., 2023), both U-Net and Diffusion Transformer (DiT) (Peebles & Xie, 2023; Yuan et al., 2024; Zhao et al., 2024) denoisers are supported, enabling exploration of hybrid backbone designs. More recent work has begun to replace U-Net with DiT architectures to improve scalability and expressive power. Diffusion Transformer Policy (Hou et al., 2024) is itself a DiT variant within the broader Diffusion Policy framework and uses a large-scale Transformer (Vaswani et al., 2017) as the denoiser in continuous action spaces, conditioned on visual observations and language instructions. The Diffusion-VLA framework (Wen et al., 2024) unifies autoregressive next-token reasoning with diffusion-based action generation into a single, scalable framework for fast, interpretable, and generalizable visuomotor robot policies. DexVLA (Wen et al., 2025) introduces plug-in diffusion expert modules that decouple action generation from the core VLA backbone. However, the iterative denoising steps inherent in diffusion models introduce substantial inference latency that poses challenges for high-frequency VLA tasks requiring real-time responsiveness. Consequently, accelerating the inference procedure of diffusion-based policies through techniques such as caching (Xu et al., 2025; Song et al., 2025) is critical for deploying responsive VLA-driven agents (Chiang et al., 2024; Xiang et al., 2025; Li et al., 2025).

## A.3 LIMITATION

The primary limitation of our work arises when the base model's accuracy on a given task is very low, as our caching strategy may inadvertently amplify this inaccuracy.

## A.4 PROOF FOR PROPOSITION 3.1

**Assumption A.1.**

*1. Activation function $\phi$ is twice continuously differentiable with bounded second derivative*

*2. LayerNorm variance $\sigma(X) \geq \sigma_{\min} > 0$ for all valid inputs $X$*

3. Weight matrices satisfy $\|W_1\|_2 \leq C_1$, $\|W_2\|_2 \leq C_2$ for fixed constants $C_1, C_2$

**Proposition A.1.** *Under Assumption 1, for input error $\delta$ with $\|\delta\| \leq \epsilon$, the FFN block output error admits the first-order approximation:*

$$\text{FFN}(X + \delta) - \text{FFN}(X) = f(\delta) + O(\|\delta\|^2) \tag{15}$$

*where the linear response operator $f(\delta)$ is given by:*

$$f(\delta) = W_2 \operatorname{diag}(\phi'(U)) W_1 (A - B) \delta \tag{16}$$

*with $U = W_1 \text{LN}(X) + b_1$ and operators:*

$$A = \frac{\operatorname{diag}(\gamma) \cdot (I - \frac{1}{d}\mathbf{1}\mathbf{1}^\top)}{\sigma(X)} \tag{17}$$

$$B = \frac{\operatorname{diag}(\gamma) \cdot (X - \mu(X)\mathbf{1})(X - \mu(X)\mathbf{1})^\top}{d\sigma(X)^3} \tag{18}$$

*Proof.* We analyze the propagation of input error $\delta$ :

Let $\Delta\widetilde{X} = \text{LN}(X + \delta) - \text{LN}(X)$. Define:

$$\mu \triangleq \mu(X), \quad \sigma \triangleq \sigma(X)$$
$$\mu_\delta \triangleq \mu(\delta), \quad \sigma_\delta^2 \triangleq \sigma^2(X + \delta)$$

The mean of $X + \delta$ is given by:

$$\Delta\mu = \mu(X + \delta) - \mu = \frac{1}{d}\mathbf{1}^\top\delta = \mu_\delta \tag{19}$$

The variance of $X + \delta$ is given by:

$$\sigma_\delta^2 = \frac{1}{d}\|X + \delta - (\mu + \mu_\delta)\mathbf{1}\|^2$$
$$= \sigma^2 + \frac{2}{d}(X - \mu\mathbf{1})^\top(\delta - \mu_\delta\mathbf{1}) + O(\|\delta\|^2) \tag{20}$$

Taking the square root and expanding:

$$\Delta\sigma = \frac{(X - \mu\mathbf{1})^\top\delta}{d\sigma} + O(\|\delta\|^2) \tag{21}$$

For each dimension $i$:

$$\Delta\widetilde{X}_i = \gamma\Big(\frac{X_i + \delta_i - \mu - \mu_\delta}{\sigma + \Delta\sigma} - \frac{X_i - \mu}{\sigma}\Big)$$
$$\approx \gamma\Big(\frac{\delta_i - \mu_\delta}{\sigma} - \frac{(X_i - \mu)}{\sigma^2}\Delta\sigma\Big) \tag{22}$$

Substituting $\Delta\sigma$ from Eq. 21 yields

$$\Delta\widetilde{X}_i = \gamma\Big(\frac{\delta_i - \mu_\delta}{\sigma} - \frac{(X_i - \mu)}{d\,\sigma^3}\sum_{k=1}^{d}(X_k - \mu)\,\delta_k\Big)$$
$$= \sum_{j=1}^{d}\Big[\frac{\gamma(\delta_{ij} - \frac{1}{d})}{\sigma} - \frac{\gamma\,(X_i - \mu)(X_j - \mu)}{d\,\sigma^3}\Big]\delta_j. \tag{23}$$

Eq. 23 implies:

$$\Delta\widetilde{X} = \underbrace{\frac{\mathrm{diag}(\gamma) \cdot \left(I - \frac{1}{d}\mathbf{1}\mathbf{1}^\top\right)}{\sigma}}_{A} \delta \; - \; \underbrace{\frac{\mathrm{diag}(\gamma) \cdot (X - \mu\mathbf{1})(X - \mu\mathbf{1})^\top}{d\,\sigma^3}}_{B} \delta \; + \; O(\|\delta\|^2). \tag{24}$$

The error propagates through the first linear layer:

$$\Delta U = W_1 \Delta\widetilde{X} = W_1(A - B)\delta + O(\|\delta\|^2) \tag{25}$$

Using Taylor expansion of $\phi$ at $U$:

$$\begin{aligned}
\phi(U + \Delta U) - \phi(U) &= \mathrm{diag}(\phi'(U))\Delta U + O(\|\Delta U\|^2) \\
&= \mathrm{diag}(\phi'(U))W_1(A - B)\delta + O(\|\delta\|^2)
\end{aligned} \tag{26}$$

Finally, projecting through $W_2$:

$$f(\delta) = W_2 \,\mathrm{diag}(\phi'(U))\, W_1 \, (A - B)\, \delta. \tag{27}$$

The proof is complete.

$\square$

## A.5  DISCUSSION ON CAUSE OF ERROR SURGE PHENOMENON

Based on Eq. 10, we conducted additional experiments to examine the Frobenius norm distributions of weights across different blocks. In FFN blocks, errors propagate sequentially through a layer normalization, the first linear transformation ($W_1$), a GELU activation function, and the second linear transformation ($W_2$). In contrast, error propagation in attention blocks involves layer normalization, the query ($W_Q$), key ($W_K$), and value ($W_V$) projection matrices, and an output projection matrix that aggregates the multi-head attention outputs. As illustrated in Fig. 8, we found that the magnitudes of $W_1$ and $W_2$ are approximately three times larger than those of $W_V$. This substantial difference in weight magnitudes likely contributes to the observed error surges within FFN blocks. Furthermore, the absence of intermediate normalization steps between the two linear layers in FFN blocks may exacerbate error amplification.

## A.6  MORE DETAILS ON BENCHMARKS

### A.6.1  DATASETS

**Pusht-T.** This dataset is based on the implicit behavioral cloning benchmark introduced by Florence et al. (Florence et al., 2022), comprising human-collected demonstrations of T-shaped block pushing with top-down RGB observations and 2D end-effector velocity control. Variation is added by random initial conditions for T block and end-effector. The task requires exploiting complex and contact-rich object dynamics to push the T block precisely, using point contacts. There are two variants: one with RGB image observations and another with 9 2D keypoints obtained from the groundtruth pose of the T block, both with proprioception for endeffector location (Chi et al., 2023).

**Block Pushing.** This dataset consists of scripted trajectories first presented in Behavior Transformers by Shafiullah et al. (Shafiullah et al., 2022). This task tests the policy's ability to model multimodal action distributions by pushing two blocks into two squares in any order. The demonstration data is generated by a scripted oracle with access to groundtruth state info (Chi et al., 2023).

**Franka Kitchen.** This dataset originates from the Relay Policy Learning framework proposed by Gupta et al. (Gupta et al., 2019), featuring 566 VR tele-operated demonstrations of multi-step manipulation tasks in a simulated kitchen using a 9-DoF Franka Panda arm. The goal is to execute

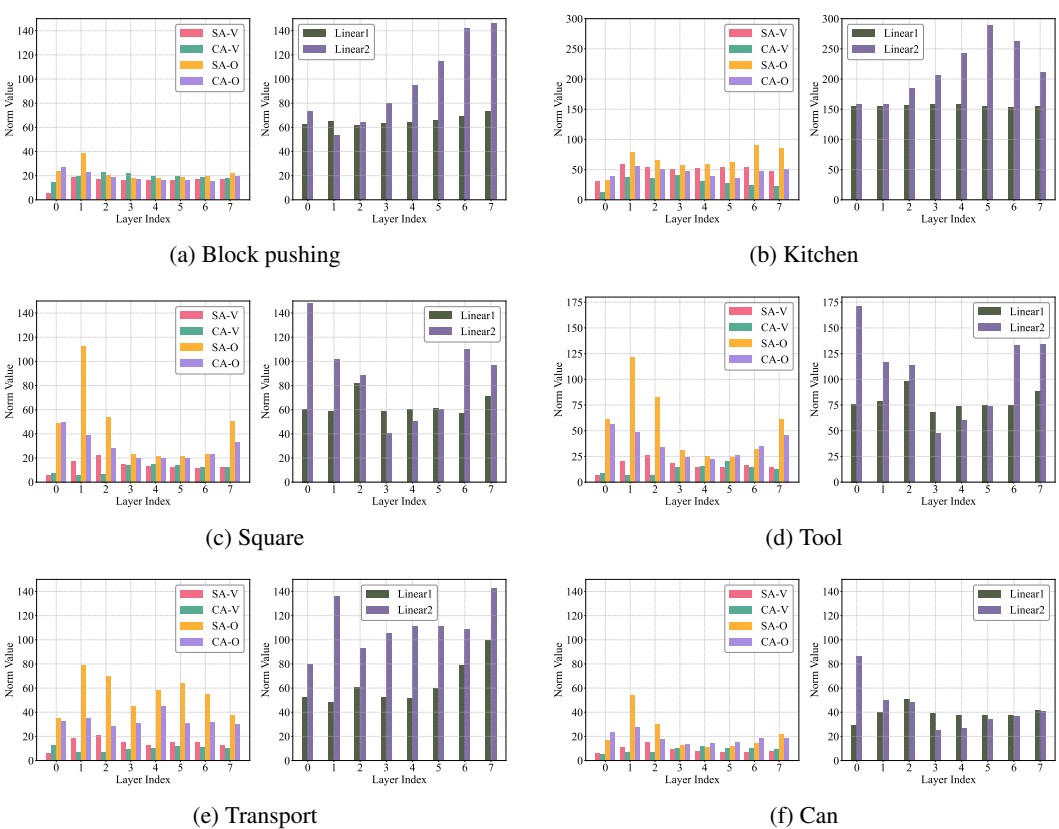

Figure 8: Weight Norm

as many demonstrated tasks as possible, regardless of order, showcasing both short-horizon and long-horizon multimodality (Chi et al., 2023).

**RoboMimic.** This dataset, introduced by Mandlekar et al. (Mandlekar et al., 2021), covers five manipulation tasks. Each task includes a Proficient-Human (PH) teleoperated demonstration set, and four of the tasks additionally offer Mixed-Human (MH) sets combining proficient and non-proficient operators (9 variants in total). The PH data were recorded by a single operator via the RoboTurk platform, whereas the MH sets were collected from six different operators using the same system.

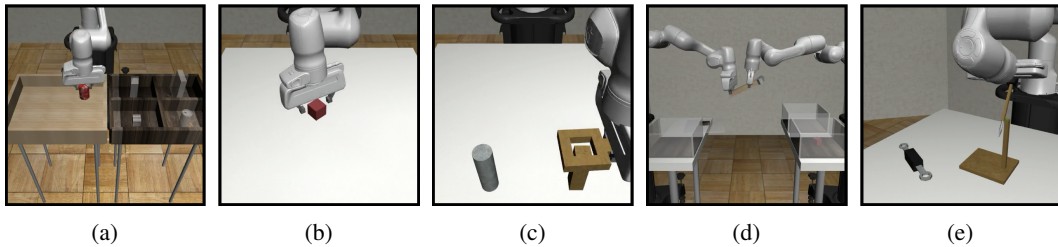

|     (a)     |     (b)     |     (c)     |     (d)     |     (e)     |

Figure 9: Visualizations of different tasks. (a) Can (b) Lift (c) Square (d) Transport (e) Tool_hang.

Fig. 9 illustrates the five subtasks in the RoboMimic Image dataset. Below we describe each subtask.

- **Can** (Fig. 9a): The robot must grasp a cylinder-shaped object and placing it into a bin. This subtask tests precise grasp planning and fingertip control under varying object poses (Mandlekar et al., 2021).

- **Lift** (Fig. 9b): The manipulator picks up a heavier, irregularly shaped object (e.g. a small box) and raises it to a designated height. It evaluates the policy's ability to modulate grip force and maintain stable trajectories (Mandlekar et al., 2021).

- **Square** (Fig. 9c): The agent must push or slide an object so that its center follows a square-shaped path on the table. This challenges both straight-line control and precise cornering maneuvers (Mandlekar et al., 2021).

- **Transport** (Fig. 9d): The agent must learn bimanual maneuvers to transfer a hammer from a closed container on a shelf to the target bin on another shelf. It tests coordinated lifting and translational motion under variable loads (Mandlekar et al., 2021).

- **Tool_Hang** (Fig. 9e): The robot arm must learn high-precision manipulation behaviors to assemble a frame by inserting a hook into a narrow base. This requires fine-tuned wrist orientation and insertion accuracy (Mandlekar et al., 2021).

### A.6.2 PRE-TRAINED CHECKPOINTS

We use the pre-trained checkpoints of `diffusion_policy_transformer` model provided by Diffusion Policy (Chi et al., 2023), where checkpoints of image-based tasks are stored under link[1] and those of multi-stage tasks are stored under link[2]. Following Diffusion Policy (Chi et al., 2023), we evaluate the success rates of two types of checkpoints: The checkpoints that achieve the maximum performance during training and those stored in the last 10 epochs. For robustness, these checkpoints are collected in three training seeds.

### A.7 ADDITIONAL EXPERIMENTS ON $n$

In this experiment, we aim to answer two questions: (1) how $n$ influences the effectiveness of BUA in mitigating update-induced error? (2) Does the effectiveness of $n$ relate to $S$?

To answer question 1, we evaluate all tasks with $n = 3$ and cache number $S = 10$. As shown in Table 2, when $n = 3$, the average success rates for the three task categories were 0.71, 0.77, and 0.97, respectively, which are slightly lower than those for $n = 5$ (0.79, 0.77, 0.98) but still significantly

---

[1]https://diffusion-policy.cs.columbia.edu/data/experiments/image/
[2]https://diffusion-policy.cs.columbia.edu/data/experiments/low_dim/

better than the *Uniform* baseline. Increasing $n$ can further improve performance at the cost of higher computational overhead. Therefore, we choose $n = 5$ as the default setting, as it achieves lossless performance while maximizing acceleration.

To answer the second question, we investigate the relationship between the hyperparameter $n$ (number of selected upstream blocks) and $S$ (cache number) in the context of the effectiveness of BUA in mitigating update-induced errors. The experimental results in Table 3 demonstrate that the effectiveness of $n$ is closely tied to $S$.

$n$ plays a more important role in mitigating update-induced errors. For certain difficult tasks, such as Tool_hang$_{ph}$ and Transport$_{mh}$, the results for $n = 3, S = 20$ (e.g., 0.32/0.53 and 0.29/0.43, respectively) are outperformed by $n = 5, S = 10$ (e.g., 0.49/0.55 and 0.30/0.46, respectively). This indicates that an appropriately tuned $n$ plays a dominant role in optimizing the effectiveness of BUA for challenging tasks.

Notably, the effectiveness of $n$ depends on an appropriate $S$. for a fixed $n = 3$, increasing $S$ from 5 to 20 significantly improves the average success rate across tasks with performance collapse, from 0.07 at $S = 5$ to 0.80 at $S = 20$, suggesting that a larger number of cache update steps helps to mitigate update-induced errors. Thus, the interplay between $n$ and $S$ suggests that a balanced combination, such as $n = 5, S = 10$, achieves robust performance across diverse tasks.

Table 2: Effect of the hyperparameter $n$ on the mitigation of update-induced error across PH, MH and multi-stage settings by BUA ($S = 10$).

Benchmark on Proficient Human (PH) demonstration data.

| Method | Success Rate ↑ | | | | | | AVG | FLOPs | Speed × |
|---|---|---|---|---|---|---|---|---|---|
| | Lift | Can | Square | Transport | Tool | Push–T | | | |
| BAC ($n = 3$) | 1.00/1.00 | 0.93/0.96 | 0.83/0.91 | 0.81/0.79 | 0.03/0.07 | 0.59/0.60 | 0.71 | 2.39G | 3.21 |
| BAC ($n = 5$) | 1.00/1.00 | 0.94/0.97 | 0.82/0.89 | 0.77/0.82 | 0.49/0.55 | 0.59/0.62 | 0.79 | 2.66G | 3.40 |

Benchmark on Mixed Human (MH) demonstration data.

| Method | Success Rate ↑ | | | | AVG | FLOPs | Speed × |
|---|---|---|---|---|---|---|---|
| | Lift | Can | Square | Transport | | | |
| BAC ($n = 3$) | 0.97/0.99 | 0.93/0.97 | 0.79/0.77 | 0.21/0.53 | 0.77 | 2.48G | 3.21 |
| BAC ($n = 5$) | 0.99/0.98 | 0.95/0.97 | 0.77/0.79 | 0.30/0.46 | 0.77 | 2.64G | 3.41 |

Benchmark on multi-stage tasks. For Block-Pushing task, $p_x$ is the frequency of pushing x blocks into the targets. For Kitchen task, $p_x$ is the frequency of interacting with x or more objects (e.g., the bottom burner).

| Method | Success Rate ↑ | | | | | | AVG | FLOPs | Speed × |
|---|---|---|---|---|---|---|---|---|---|
| | BP$_{p1}$ | BP$_{p2}$ | Kit$_{p1}$ | Kit$_{p2}$ | Kit$_{p3}$ | Kit$_{p4}$ | | | |
| BAC ($n = 3$) | 0.99/0.98 | 0.94/0.93 | 0.99/1.00 | 0.97/0.99 | 0.95/0.99 | 0.89/0.97 | 0.97 | 2.23G | 3.38 |
| BAC ($n = 5$) | 1.00/0.99 | 0.97/0.95 | 1.00/0.99 | 1.00/0.99 | 1.00/0.99 | 0.94/0.97 | 0.98 | 2.44G | 3.60 |

## A.8 Additional Experiments on Different Similarity Metrics

In this experiment, we evaluate the performance of BAC across all tasks using four similarity metrics: Mean Squared Error (MSE), L1-Norm distance (L1), Wasserstein-1 distance (Wa), and Cosine similarity (Cosine), as presented in Table 4. BAC with metric Cosine achieves average success rates of 0.79, 0.77, and 0.98 for PH, MH, and multi-stage tasks, outperforming BAC with metric MSE (0.78, 0.70, 0.98), BAC with metric L1 (0.72, 0.67, 0.98), and BAC with metric Wa (0.70, 0.56, 0.81). BAC with metric Cosine excels particularly in MH settings while matching MSE and L1 in multi-stage tasks. BAC with metric Wa struggles with complex tasks (e.g., Kit$_{p4}$: 0.39/0.67). All metrics exhibit similar computational costs, with FLOPs ranging from 2.40G to 2.73G and speed

Table 3: Benchmark Results across tasks that have performance collapse (Tool_hang$_{ph}$, Transport$_{mh}$, Kitchen), all results set $n = 3$ and $S \in \{5, 7, 20\}$.

| Method | Success Rate ↑ | | | | | | AVG | FLOPs | Speed × |
|---|---|---|---|---|---|---|---|---|---|
| | Tool$_{ph}$ | Trans$_{mh}$ | Kit$_{p1}$ | Kit$_{p2}$ | Kit$_{p3}$ | Kit$_{p4}$ | | | |
| BAC ($S = 5$) | 0.00/0.00 | 0.26/0.50 | 0.04/0.07 | 0.00/0.00 | 0.00/0.00 | 0.00/0.00 | 0.07 | 1.49G | 4.02 |
| BAC ($S = 7$) | 0.00/0.01 | 0.19/0.32 | 0.82/0.93 | 0.64/0.79 | 0.55/0.70 | 0.33/0.52 | 0.49 | 1.90G | 3.66 |
| BAC ($S = 10$) | 0.03/0.07 | 0.21/0.53 | 0.99/1.00 | 0.97/0.99 | 0.95/0.99 | 0.89/0.97 | 0.72 | 2.43G | 3.34 |
| BAC ($S = 20$) | 0.32/0.53 | 0.29/0.43 | 1.00/1.00 | 1.00/1.00 | 1.00/1.00 | 0.99/0.97 | 0.80 | 4.16G | 2.53 |

from $3.07\times$ to $3.42\times$. Thus, BAC with metric Cosine and BAC with metric MSE demonstrate the most robust performance, with Cosine preferred for noisy data and complex tasks.

Table 4: Benchmark Results across Proficient Human (PH), Mixed Human (MH), and Multi-stage Demonstrations ($n = 5$, $S = 10$) using MSE, L1, and Wa Metrics.

Benchmark on Proficient Human (PH) demonstration data.

| Method | Success Rate ↑ | | | | | | AVG | FLOPs | Speed × |
|---|---|---|---|---|---|---|---|---|---|
| | Lift | Can | Square | Transport | Tool | Push–T | | | |
| BAC (MSE) | 1.00/1.00 | 0.79/0.79 | 0.71/0.83 | 0.75/0.77 | 0.40/0.55 | 0.58/0.66 | 0.78 | 2.73G | 3.21 |
| BAC (L1) | 1.00/1.00 | 0.89/0.95 | 0.74/0.85 | 0.71/0.73 | 0.19/0.33 | 0.56/0.65 | 0.72 | 2.60G | 3.19 |
| BAC (Wa) | 1.00/1.00 | 0.37/0.75 | 0.79/0.87 | 0.80/0.79 | 0.37/0.50 | 0.52/0.61 | 0.70 | 2.73G | 3.28 |
| BAC (Cosine) | 1.00/1.00 | 0.94/0.97 | 0.82/0.89 | 0.77/0.82 | 0.49/0.55 | 0.59/0.62 | 0.79 | 2.66G | 3.40 |

Benchmark on Mixed Human (MH) demonstration data.

| Method | Success Rate ↑ | | | | AVG | FLOPs | Speed × |
|---|---|---|---|---|---|---|---|
| | Lift | Can | Square | Transport | | | |
| BAC (MSE) | 1.00/0.99 | 0.41/0.86 | 0.75/0.77 | 0.26/0.50 | 0.70 | 2.69G | 3.21 |
| BAC (L1) | 0.97/1.00 | 0.44/0.87 | 0.71/0.74 | 0.23/0.43 | 0.67 | 2.63G | 3.18 |
| BAC (Wa) | 0.98/0.99 | 0.83/0.90 | 0.09/0.05 | 0.19/0.45 | 0.56 | 2.71G | 3.07 |
| BAC (Cosine) | 0.99/0.98 | 0.95/0.97 | 0.77/0.79 | 0.30/0.46 | 0.77 | 2.64G | 3.41 |

Benchmark on multi-stage tasks. For Block-Pushing task, $p_x$ is the frequency of pushing x blocks into the targets. For Kitchen task, $p_x$ is the frequency of interacting with x or more objects (e.g., the bottom burner).

| Method | Success Rate ↑ | | | | | | AVG | FLOPs | Speed × |
|---|---|---|---|---|---|---|---|---|---|
| | BP$_{p1}$ | BP$_{p2}$ | Kit$_{p1}$ | Kit$_{p2}$ | Kit$_{p3}$ | Kit$_{p4}$ | | | |
| BAC (MSE) | 0.99/0.99 | 0.95/0.93 | 1.00/1.00 | 1.00/1.00 | 0.99/0.99 | 0.97/0.93 | 0.98 | 2.49G | 3.39 |
| BAC (L1) | 0.99/0.99 | 0.94/0.94 | 1.00/1.00 | 1.00/1.00 | 1.00/1.00 | 0.93/0.95 | 0.98 | 2.40G | 3.42 |
| BAC (Wa) | 0.99/1.00 | 0.95/0.95 | 0.83/0.95 | 0.66/0.90 | 0.55/0.85 | 0.39/0.67 | 0.81 | 2.61G | 3.08 |
| BAC (Cosine) | 1.00/0.99 | 0.97/0.95 | 1.00/0.99 | 1.00/0.99 | 1.00/0.99 | 0.94/0.97 | 0.98 | 2.44G | 3.60 |

## A.9 More details on Temporal Similarities

We compute cosine similarity between intermediate features of different timesteps to analyze temporal similarity patterns, using the Square$_{ph}$ task as a case study. Fig. 10 provides more details on the temporal similarity patterns of different decoder blocks. The observation that the feature similarity between consecutive timesteps varies non-uniformly over time exists in all the decoder blocks. This suggests the necessity of ACS method.

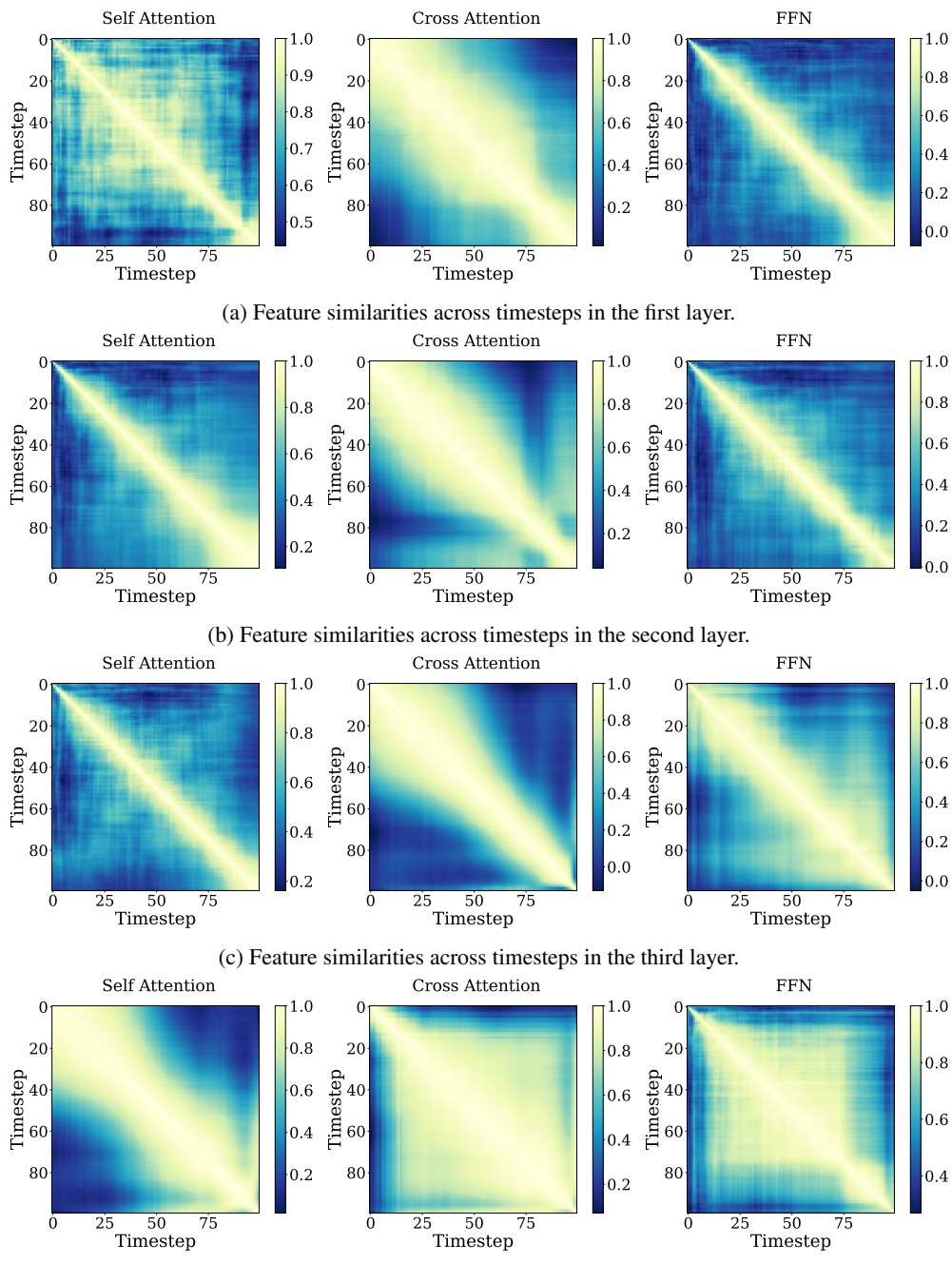

(a) Feature similarities across timesteps in the first layer.

(b) Feature similarities across timesteps in the second layer.

(c) Feature similarities across timesteps in the third layer.

(d) Feature similarities across timesteps in the fourth layer.

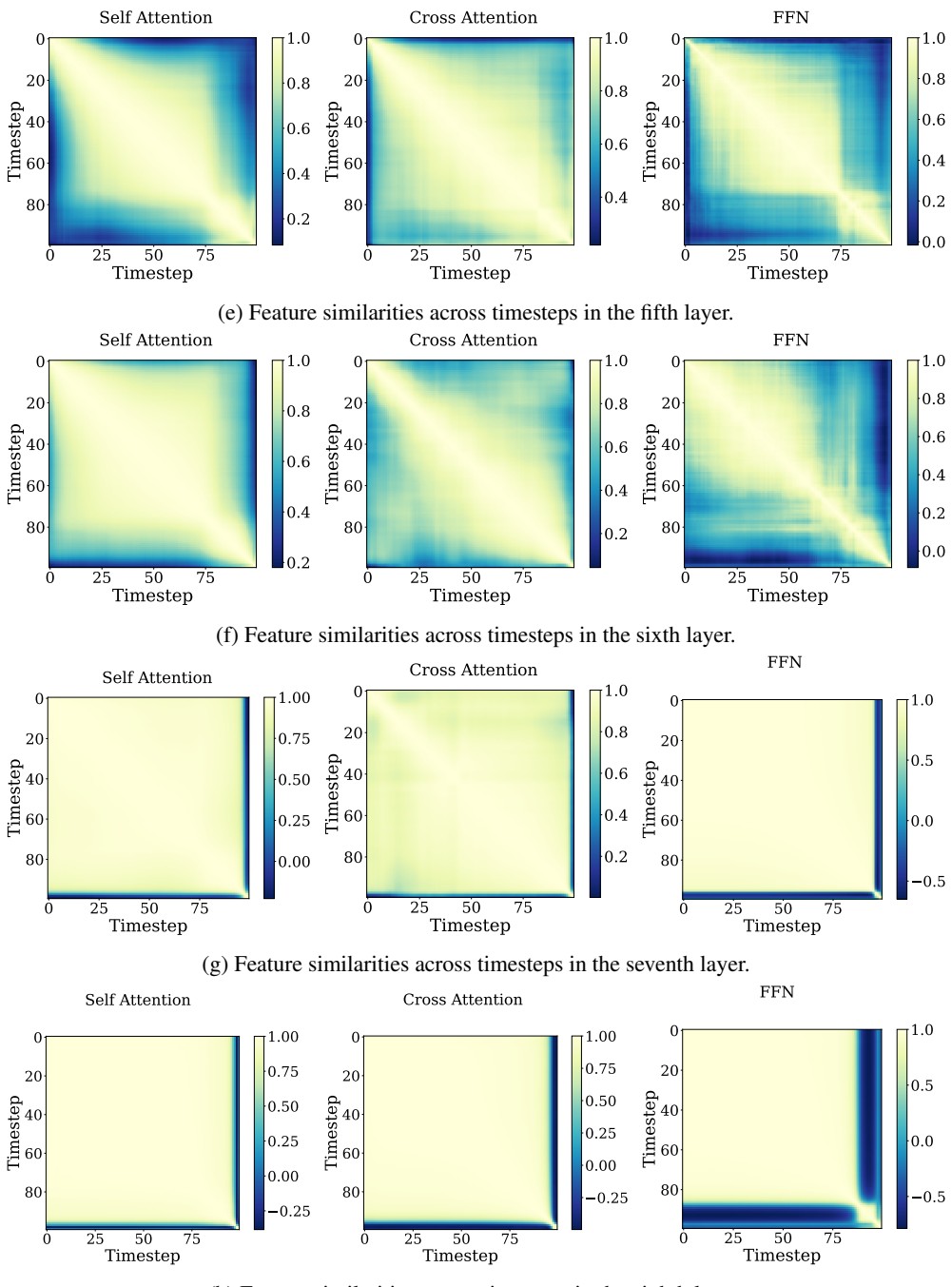

(e) Feature similarities across timesteps in the fifth layer.

(f) Feature similarities across timesteps in the sixth layer.

(g) Feature similarities across timesteps in the seventh layer.

(h) Feature similarities across timesteps in the eighth layer.

Figure 10: Feature similarities across timesteps for different blocks in each decoder layer.

To provide more details on the block-wise temporal similarity pattern, we compute cosine similarities under different intervals of consecutive steps in timestep $t$ and earlier timesteps $t-k$ for various values of $k$ (1, 5, 10, 15, 20). As shown in Fig. 11, different blocks exhibit distinct temporal similarity patterns. Some blocks maintain high similarity across long horizons, indicating few updates, while others show rapid drops in similarity even at short intervals, suggesting more update steps. This suggests the necessity of a block-wise schedule.

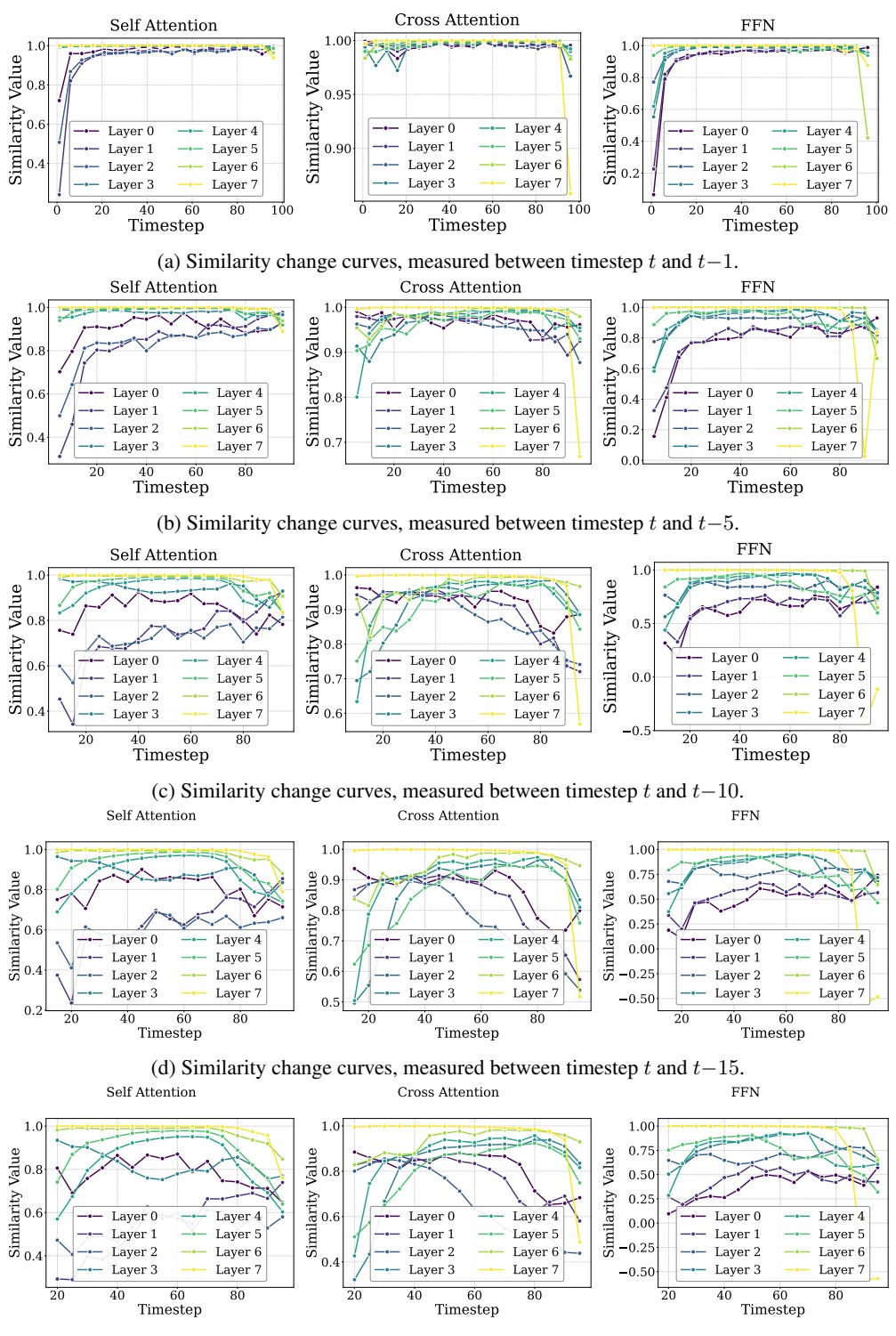

Figure 11: Block-wise feature similarity between consecutive steps. Each subfigure shows the similarity between features at timestep $t$ and $(t-k$, where $k = 1, 5, 10, 15, 20)$ for different blocks.

### A.10 MORE DETAILS ON THE HIGH EPISODE HOMOGENEITY WITHIN INDIVIDUAL TASKS

In this section, we present evidence for the high episode homogeneity of an embodied task by highlighting the distinct similarity patterns in action generation tasks versus image generation tasks.

In action generation tasks, as shown in Fig. 12, we visualize the feature similarity matrices from the same layer of Diffusion Policy, under the same task (Square$_{ph}$), across two different scene demos (demo id 11001 vs. demo id 20000). Despite changes in scene settings, the similarity matrices remain strikingly consistent, suggesting a high degree of representational homogeneity across episodes.

In contrast, in image generation tasks, as shown in Fig. 13, we visualize the feature similarity matrices from the same layer of DiT-XL/2 (Peebles & Xie, 2023), across two different classes in ImageNet (class label 15:"robin, American robin, Turdus migratorius" vs. class label 800: "slot, one-armed bandit"). The results from both the self-attention and MLP blocks reveal clear differences in feature patterns.

While an image generation task shows obvious differences in feature patterns across different classes, an action generation task shows almost no difference across different scene demos within the same task. These observations support the efficiency of our method, specifically in embodied episodes. Benefiting from the high episode homogeneity within individual tasks, our method can be run only once for a given task before inference, incurring virtually no additional cost.

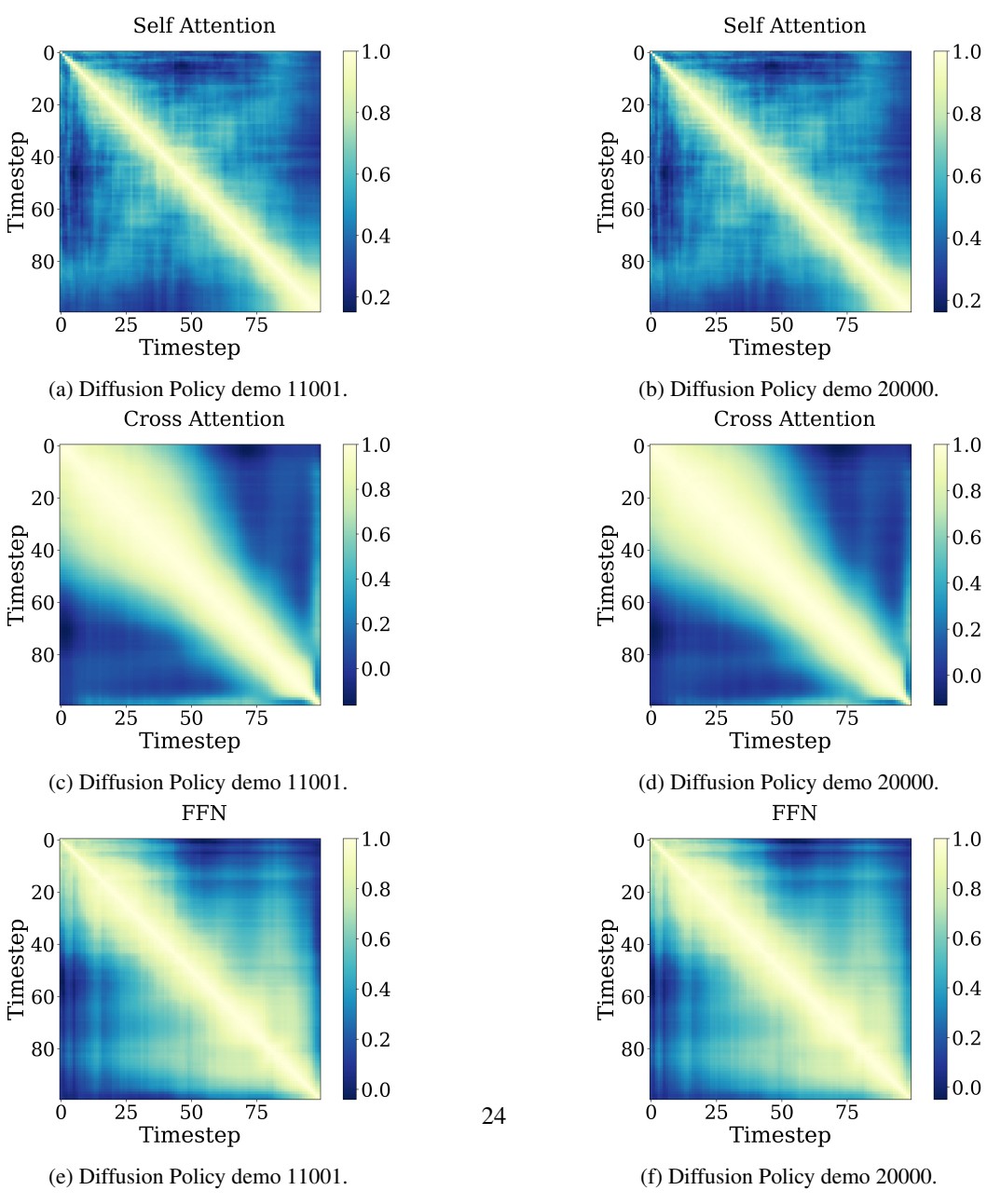

(a) Diffusion Policy demo 11001.    (b) Diffusion Policy demo 20000.

(c) Diffusion Policy demo 11001.    (d) Diffusion Policy demo 20000.

24

(e) Diffusion Policy demo 11001.    (f) Diffusion Policy demo 20000.

Figure 12: Feature similarities across timesteps in action generation tasks. We visualize similarity matrices of different blocks in the third layer across different scene demos (e.g., demo id 11001 vs. demo id 20000).

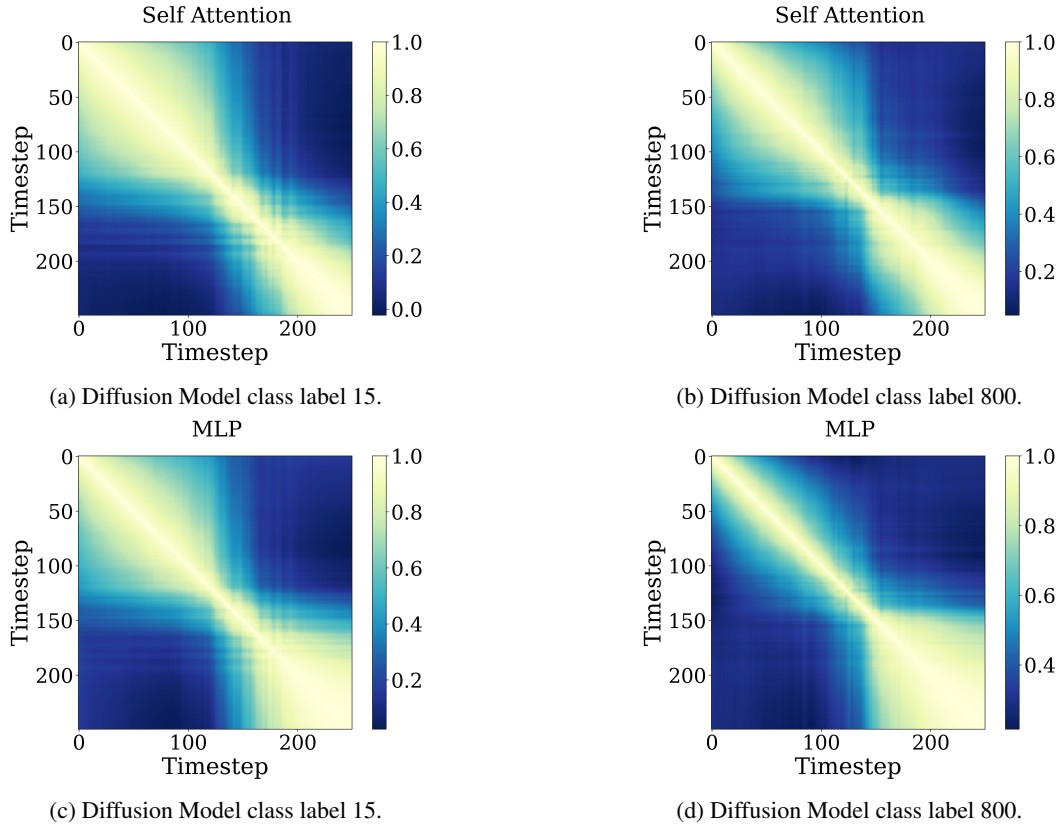

(a) Diffusion Model class label 15.

(b) Diffusion Model class label 800.

(c) Diffusion Model class label 15.

(d) Diffusion Model class label 800.

Figure 13: Feature similarities across timesteps in image generation tasks. We visualize similarity matrices of different blocks in the fourteenth layer across different classes (e.g., class label 15 vs. class label 800).

## A.11   MORE DETAILS FOR FIG.3

To support findings of the error surge phenomenon, we present comparisons of caching errors across different FFN blocks using a block-wise schedule versus a unified schedule. As shown in Fig. 14, the block-wise schedule leads to error surges in nearly all FFN blocks except the first one, in contrast to the unified schedule. This observation further suggests that caching errors can propagate through downstream FFN blocks.

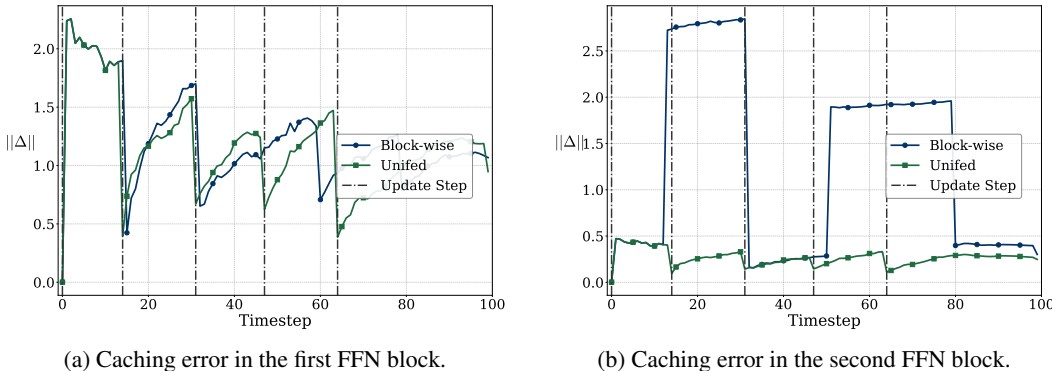

(a) Caching error in the first FFN block.

(b) Caching error in the second FFN block.

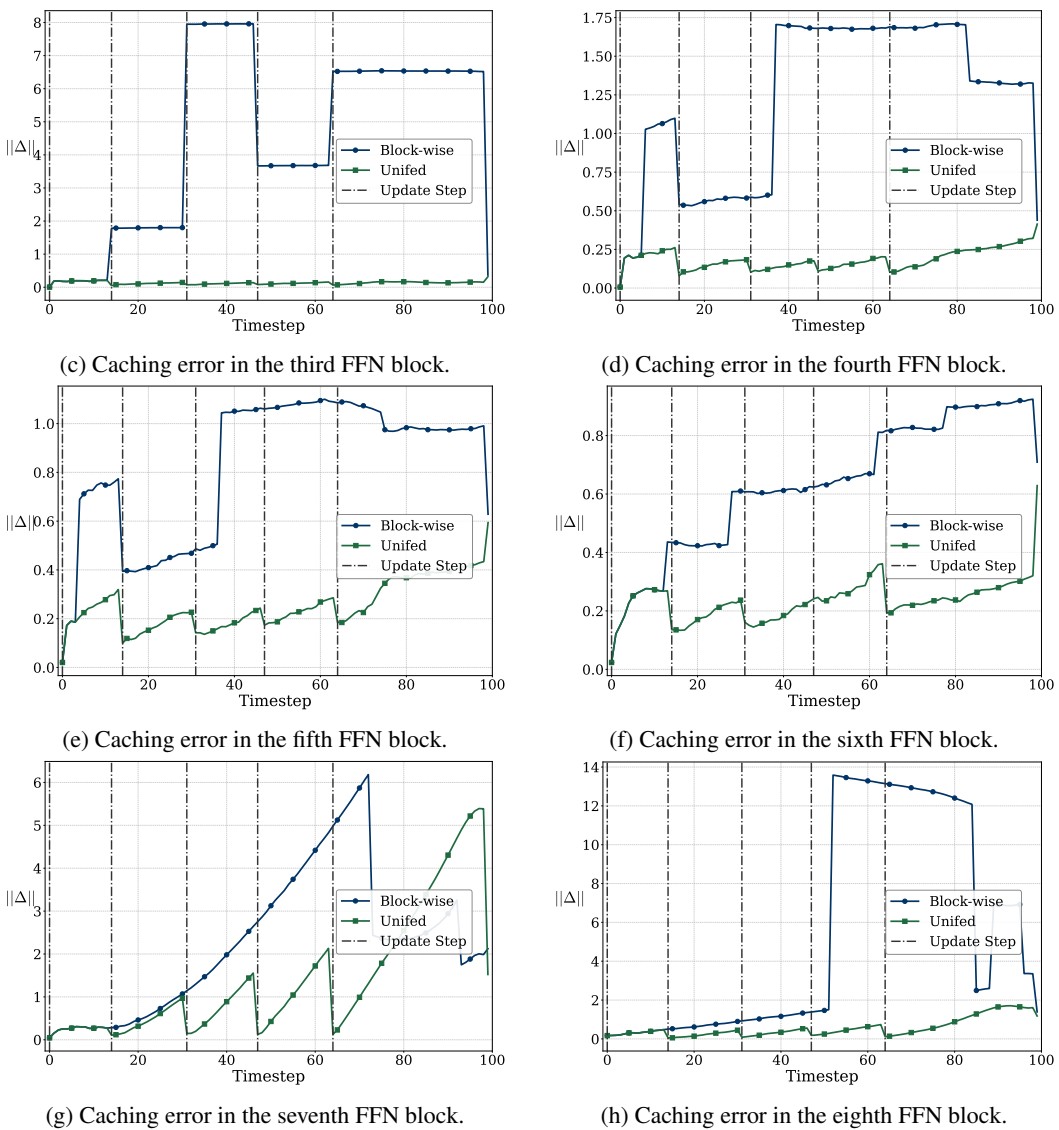

(c) Caching error in the third FFN block.

(d) Caching error in the fourth FFN block.

(e) Caching error in the fifth FFN block.

(f) Caching error in the sixth FFN block.

(g) Caching error in the seventh FFN block.

(h) Caching error in the eighth FFN block.

Figure 14: Caching error across different FFN blocks using a block-wise schedule versus a unified one.

Fig. 15 presents block-wise heatmaps of caching error across all blocks throughout the diffusion process, covering a diverse set of tasks and two demonstration settings: Proficient Human (PH) and Mixed Human (MH). Each subfigure corresponds to a different task, with color intensity representing the magnitude of caching error at each block and timestep, and white dots indicating cache update steps.

These visualizations support our analysis of inter-block error propagation by consistently exhibiting the following key phenomenon: in multiple tasks, blocks occasionally update at steps where their upstream blocks with large caching errors have not yet been updated. This mismatch leads to sudden, sharp increases in error (seen as abrupt darkening) in the downstream block, with no smooth transition. Importantly, even after upstream blocks are later updated, the downstream surge in error remains, indicating that the update failed to recover from the propagated upstream error.

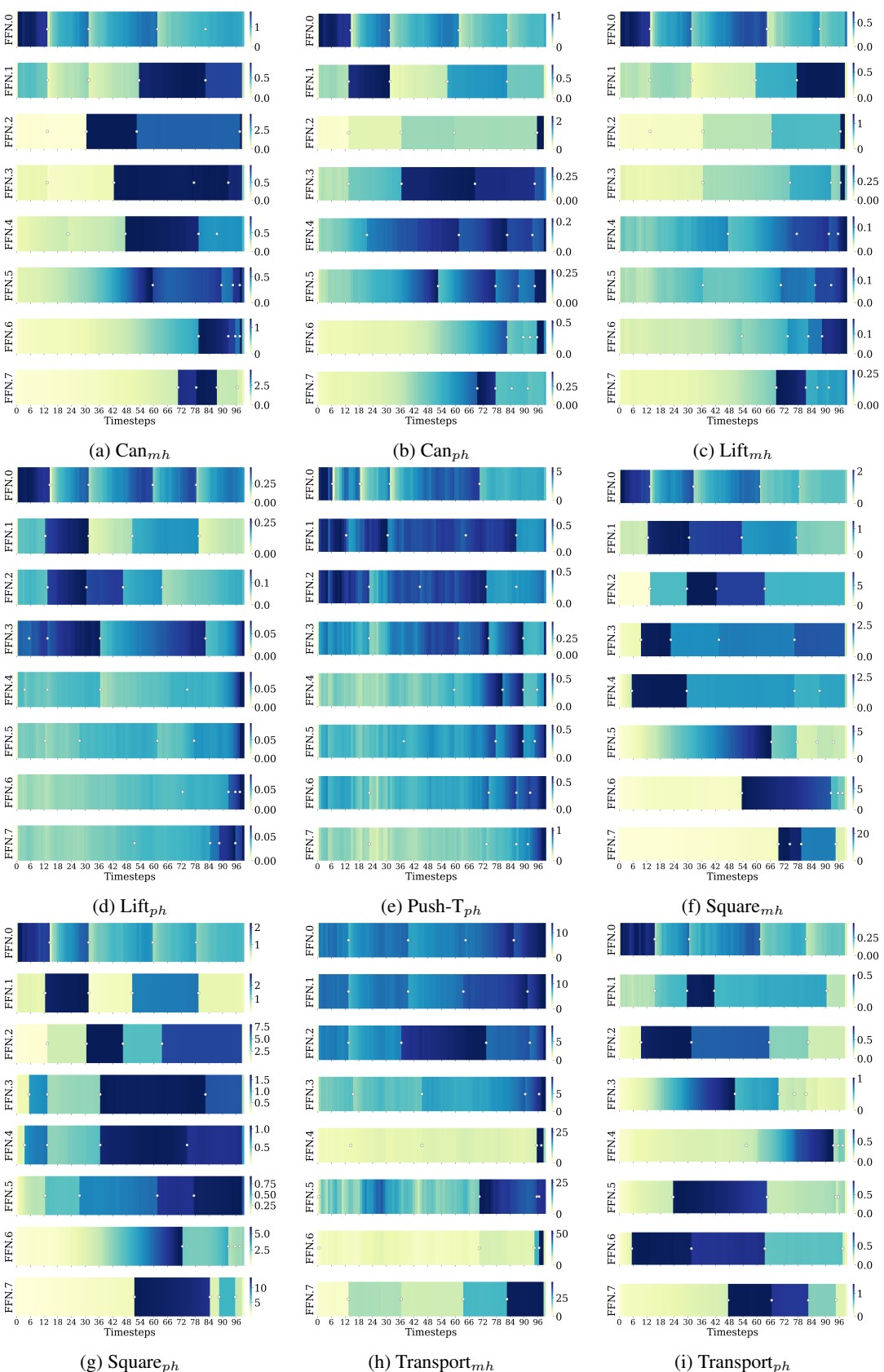

(a) Can$_{mh}$

(b) Can$_{ph}$

(c) Lift$_{mh}$

(d) Lift$_{ph}$

(e) Push-T$_{ph}$

(f) Square$_{mh}$

(g) Square$_{ph}$

(h) Transport$_{mh}$

(i) Transport$_{ph}$

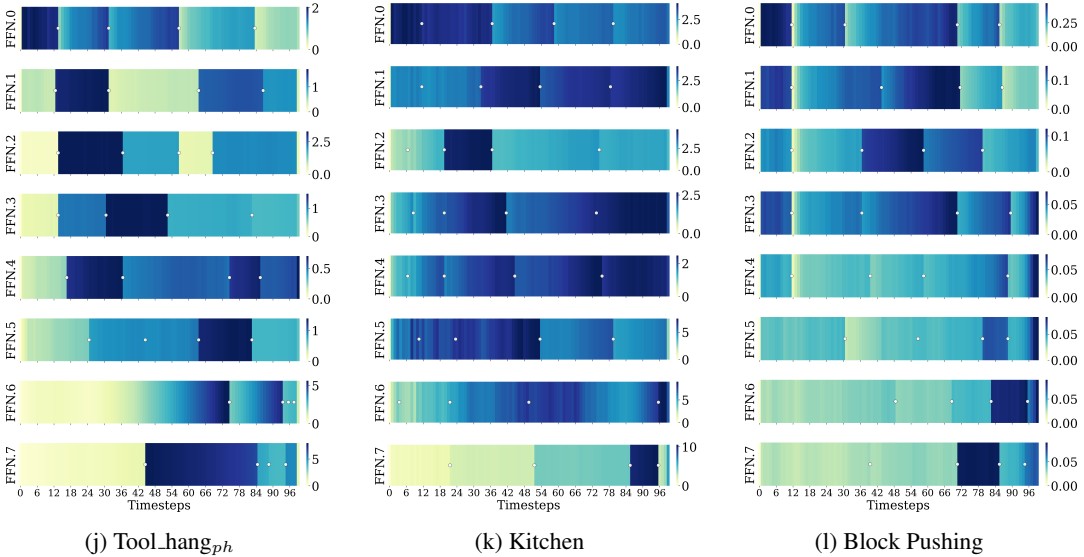

(j) Tool_hang$_{ph}$         (k) Kitchen         (l) Block Pushing

Figure 15: Caching error across all blocks throughout the diffusion process, with white dots indicating update steps.

## A.12 DETAILS ON UPDATE STEPS COMPUTED BY ACS

Our algorithm employs a two-stage paradigm where we apply ACS followed by BUA to determine the optimal update steps for different blocks in the offline stage, and then accelerate Diffusion Policy by updating and reusing the cached features based on the prepared update steps in the online stage.

In Tables 5, 6, 7, 8, 9, 10, 11, 12, 13, 14, 15 and 16, we report the update steps after employing ACS for all blocks across all tasks at $S = 10$.

Table 5: Update steps for $\text{Can}_{ph}$ computed by ACS.

| Block | Steps |
|---|---|
| layers.0.SA | 0, 2, 9, 18, 30, 49, 62, 69, 82, 91 |
| layers.0.CA | 0, 18, 33, 44, 57, 71, 80, 84, 89, 94 |
| layers.0.FFN | 0, 4, 10, 19, 31, 40, 53, 65, 79, 88 |
| layers.1.SA | 0, 4, 10, 21, 32, 44, 54, 65, 79, 88 |
| layers.1.CA | 0, 14, 25, 36, 48, 60, 73, 82, 87, 93 |
| layers.1.FFN | 0, 8, 16, 28, 38, 51, 60, 68, 80, 93 |
| layers.2.SA | 0, 4, 8, 13, 28, 38, 54, 68, 80, 90 |
| layers.2.CA | 0, 9, 18, 28, 38, 51, 67, 80, 86, 95 |
| layers.2.FFN | 0, 14, 27, 37, 51, 62, 71, 80, 85, 95 |
| layers.3.SA | 0, 19, 30, 40, 51, 62, 72, 82, 90, 98 |
| layers.3.CA | 0, 6, 14, 25, 37, 49, 62, 75, 90, 98 |
| layers.3.FFN | 0, 4, 14, 23, 37, 48, 60, 78, 89, 95 |
| layers.4.SA | 0, 13, 25, 37, 53, 66, 81, 90, 95, 98 |
| layers.4.CA | 0, 3, 9, 22, 39, 57, 80, 86, 94, 98 |
| layers.4.FFN | 0, 6, 18, 39, 62, 75, 80, 86, 93, 96 |
| layers.5.SA | 0, 6, 22, 42, 63, 80, 88, 93, 96, 98 |
| layers.5.CA | 0, 4, 11, 29, 43, 61, 81, 90, 95, 98 |
| layers.5.FFN | 0, 37, 60, 79, 88, 92, 94, 96, 98, 99 |
| layers.6.SA | 0, 37, 67, 83, 89, 92, 94, 96, 97, 98 |
| layers.6.CA | 0, 13, 28, 51, 74, 88, 93, 95, 96, 98 |
| layers.6.FFN | 0, 28, 62, 80, 89, 93, 95, 96, 97, 99 |
| layers.7.SA | 0, 29, 64, 79, 86, 93, 95, 96, 97, 98 |
| layers.7.CA | 0, 12, 26, 49, 68, 83, 89, 93, 95, 97 |
| layers.7.FFN | 0, 47, 69, 74, 77, 81, 86, 95, 97, 99 |

Table 6: Update steps for $\text{Lift}_{ph}$ computed by ACS.

| Block | Steps |
|---|---|
| layers.0.SA | 0, 1, 6, 51, 67, 76, 82, 87, 92, 95 |
| layers.0.CA | 0, 22, 44, 60, 70, 75, 80, 85, 93, 96 |
| layers.0.FFN | 0, 4, 8, 16, 27, 37, 49, 62, 74, 88 |
| layers.1.SA | 0, 4, 10, 19, 28, 37, 53, 65, 78, 88 |
| layers.1.CA | 0, 15, 26, 34, 39, 48, 60, 69, 78, 92 |
| layers.1.FFN | 0, 4, 10, 18, 28, 37, 54, 65, 79, 88 |
| layers.2.SA | 0, 2, 7, 19, 31, 40, 51, 63, 75, 87 |
| layers.2.CA | 0, 6, 22, 33, 39, 48, 68, 79, 90, 96 |
| layers.2.FFN | 0, 4, 10, 17, 27, 37, 51, 65, 78, 88 |
| layers.3.SA | 0, 6, 19, 40, 65, 74, 81, 86, 94, 98 |
| layers.3.CA | 0, 6, 15, 22, 38, 53, 66, 78, 88, 97 |
| layers.3.FFN | 0, 4, 14, 31, 49, 68, 79, 88, 93, 97 |
| layers.4.SA | 0, 6, 19, 40, 65, 75, 85, 92, 95, 98 |
| layers.4.CA | 0, 10, 21, 32, 44, 57, 70, 81, 90, 97 |
| layers.4.FFN | 0, 15, 37, 68, 79, 88, 93, 96, 98, 99 |
| layers.5.SA | 0, 19, 40, 65, 75, 81, 85, 89, 93, 96 |
| layers.5.CA | 0, 9, 18, 27, 37, 56, 62, 69, 76, 86 |
| layers.5.FFN | 0, 4, 15, 37, 48, 74, 88, 93, 96, 98 |
| layers.6.SA | 0, 7, 19, 40, 58, 67, 74, 84, 90, 96 |
| layers.6.CA | 0, 8, 15, 23, 34, 44, 54, 71, 87, 95 |
| layers.6.FFN | 0, 15, 31, 44, 57, 69, 79, 88, 93, 97 |
| layers.7.SA | 0, 7, 19, 39, 53, 67, 75, 84, 89, 96 |
| layers.7.CA | 0, 8, 14, 23, 35, 45, 57, 71, 81, 95 |
| layers.7.FFN | 0, 15, 29, 40, 54, 69, 79, 86, 91, 95 |

Table 7: Update steps for $\text{Square}_{ph}$ computed by ACS.

| Block | Steps |
|---|---|
| layers.0.SA | $0, 3, 9, 17, 30, 49, 62, 69, 80, 89$ |
| layers.0.CA | $0, 10, 20, 32, 43, 53, 68, 77, 83, 91$ |
| layers.0.FFN | $0, 4, 10, 21, 31, 40, 51, 62, 75, 84$ |
| layers.1.SA | $0, 4, 10, 19, 28, 38, 51, 61, 75, 88$ |
| layers.1.CA | $0, 8, 16, 29, 43, 59, 73, 80, 88, 93$ |
| layers.1.FFN | $0, 8, 15, 23, 37, 51, 64, 75, 87, 93$ |
| layers.2.SA | $0, 7, 16, 27, 37, 49, 60, 68, 78, 88$ |
| layers.2.CA | $0, 10, 26, 40, 53, 63, 72, 81, 90, 97$ |
| layers.2.FFN | $0, 7, 14, 31, 44, 57, 67, 79, 87, 93$ |
| layers.3.SA | $0, 14, 24, 34, 44, 55, 66, 76, 84, 90$ |
| layers.3.CA | $0, 6, 11, 25, 36, 48, 59, 69, 79, 91$ |
| layers.3.FFN | $0, 4, 8, 15, 24, 37, 64, 76, 87, 95$ |
| layers.4.SA | $0, 6, 13, 24, 38, 57, 76, 82, 88, 93$ |
| layers.4.CA | $0, 4, 11, 21, 35, 56, 70, 84, 91, 97$ |
| layers.4.FFN | $0, 4, 8, 14, 23, 36, 48, 69, 78, 90$ |
| layers.5.SA | $0, 5, 11, 22, 37, 64, 78, 86, 93, 97$ |
| layers.5.CA | $0, 5, 8, 12, 18, 25, 36, 52, 68, 89$ |
| layers.5.FFN | $0, 4, 15, 24, 31, 44, 53, 64, 78, 93$ |
| layers.6.SA | $0, 9, 30, 50, 69, 78, 87, 95, 97, 98$ |
| layers.6.CA | $0, 2, 9, 16, 24, 34, 53, 71, 86, 97$ |
| layers.6.FFN | $0, 44, 61, 74, 83, 90, 93, 95, 96, 98$ |
| layers.7.SA | $0, 43, 65, 82, 88, 91, 93, 95, 96, 98$ |
| layers.7.CA | $0, 7, 53, 69, 80, 87, 93, 95, 97, 98$ |
| layers.7.FFN | $0, 10, 52, 78, 83, 87, 89, 91, 95, 97$ |

## A.13 DETAILS ON UPDATE STEPS COMPUTED BY BUA

In Tables 17, 18, 19, 20, 21, 22, 23, 24, 25, 26, 27 and 28, we report the steps added after employing BUA across all tasks at $S = 10$ and $k = 5$.

Table 8: Update steps for Transport$_{ph}$ computed by ACS.

| Block | Steps |
|-------|-------|
| layers.0.SA | $0, 2, 10, 24, 47, 58, 70, 76, 86, 94$ |
| layers.0.CA | $0, 7, 15, 23, 34, 52, 65, 76, 86, 93$ |
| layers.0.FFN | $0, 3, 10, 21, 31, 44, 56, 65, 76, 86$ |
| layers.1.SA | $0, 2, 10, 19, 30, 41, 48, 54, 70, 98$ |
| layers.1.CA | $0, 13, 21, 31, 48, 56, 65, 71, 76, 88$ |
| layers.1.FFN | $0, 1, 3, 7, 19, 30, 43, 52, 75, 90$ |
| layers.2.SA | $0, 3, 11, 18, 25, 35, 45, 72, 93, 97$ |
| layers.2.CA | $0, 14, 24, 40, 52, 66, 75, 83, 88, 93$ |
| layers.2.FFN | $0, 12, 34, 52, 62, 69, 74, 78, 84, 91$ |
| layers.3.SA | $0, 17, 42, 68, 83, 91, 94, 95, 97, 99$ |
| layers.3.CA | $0, 6, 13, 29, 45, 71, 77, 83, 91, 96$ |
| layers.3.FFN | $0, 43, 74, 82, 83, 85, 87, 91, 95, 98$ |
| layers.4.SA | $0, 16, 27, 43, 58, 82, 88, 93, 96, 99$ |
| layers.4.CA | $0, 6, 13, 30, 49, 61, 82, 91, 96, 99$ |
| layers.4.FFN | $0, 16, 34, 57, 78, 91, 92, 93, 94, 97$ |
| layers.5.SA | $0, 13, 22, 34, 47, 59, 78, 91, 94, 97$ |
| layers.5.CA | $0, 7, 24, 41, 52, 61, 82, 91, 95, 97$ |
| layers.5.FFN | $0, 14, 29, 49, 63, 70, 88, 93, 95, 97$ |
| layers.6.SA | $0, 14, 26, 41, 56, 75, 85, 93, 95, 97$ |
| layers.6.CA | $0, 7, 22, 48, 54, 61, 82, 91, 94, 97$ |
| layers.6.FFN | $0, 28, 49, 60, 67, 79, 88, 91, 96, 97$ |
| layers.7.SA | $0, 15, 26, 40, 71, 78, 82, 95, 97, 99$ |
| layers.7.CA | $0, 20, 51, 62, 68, 73, 82, 89, 98, 99$ |
| layers.7.FFN | $0, 36, 62, 67, 75, 82, 85, 91, 95, 98$ |

Table 9: Update steps for Tool_hang$_{ph}$ computed by ACS.

| Block | Steps |
|-------|-------|
| layers.0.SA | $0, 4, 9, 17, 29, 44, 58, 70, 77, 91$ |
| layers.0.CA | $0, 12, 22, 32, 44, 56, 68, 76, 84, 92$ |
| layers.0.FFN | $0, 2, 5, 10, 18, 30, 40, 54, 72, 84$ |
| layers.1.SA | $0, 2, 8, 15, 28, 40, 53, 66, 77, 89$ |
| layers.1.CA | $0, 11, 20, 31, 43, 56, 68, 78, 84, 92$ |
| layers.1.FFN | $0, 2, 6, 12, 24, 40, 50, 66, 77, 86$ |
| layers.2.SA | $0, 1, 8, 24, 31, 40, 49, 71, 80, 91$ |
| layers.2.CA | $0, 6, 13, 24, 40, 55, 67, 78, 86, 93$ |
| layers.2.FFN | $0, 3, 10, 24, 31, 40, 49, 61, 71, 85$ |
| layers.3.SA | $0, 9, 23, 31, 40, 49, 60, 71, 79, 88$ |
| layers.3.CA | $0, 2, 12, 22, 31, 44, 54, 69, 77, 87$ |
| layers.3.FFN | $0, 3, 10, 16, 24, 40, 63, 77, 84, 93$ |
| layers.4.SA | $0, 3, 13, 24, 40, 61, 76, 83, 90, 97$ |
| layers.4.CA | $0, 6, 14, 20, 32, 46, 62, 79, 88, 97$ |
| layers.4.FFN | $0, 2, 5, 10, 23, 44, 66, 78, 84, 93$ |
| layers.5.SA | $0, 3, 10, 18, 25, 39, 55, 76, 84, 97$ |
| layers.5.CA | $0, 2, 7, 16, 27, 37, 55, 75, 82, 89$ |
| layers.5.FFN | $0, 48, 71, 80, 85, 88, 91, 94, 96, 98$ |
| layers.6.SA | $0, 54, 74, 83, 89, 92, 94, 95, 96, 98$ |
| layers.6.CA | $0, 13, 27, 48, 66, 82, 90, 94, 96, 98$ |
| layers.6.FFN | $0, 11, 40, 53, 71, 80, 91, 95, 97, 99$ |
| layers.7.SA | $0, 24, 74, 82, 87, 90, 95, 96, 97, 98$ |
| layers.7.CA | $0, 6, 43, 66, 81, 90, 94, 96, 97, 98$ |
| layers.7.FFN | $0, 60, 74, 79, 81, 83, 87, 95, 97, 99$ |

Table 10: Update steps for Pusht-T computed by ACS.

| Block | Steps |
|-------|-------|
| layers.0.SA | $0, 3, 7, 15, 22, 31, 45, 63, 74, 87$ |
| layers.0.CA | $0, 7, 23, 30, 36, 45, 58, 78, 88, 95$ |
| layers.0.FFN | $0, 2, 7, 17, 25, 32, 45, 62, 74, 87$ |
| layers.1.SA | $0, 7, 22, 31, 39, 45, 54, 64, 75, 87$ |
| layers.1.CA | $0, 13, 21, 31, 44, 54, 63, 75, 80, 89$ |
| layers.1.FFN | $0, 3, 7, 17, 23, 33, 47, 59, 74, 87$ |
| layers.2.SA | $0, 6, 25, 40, 55, 64, 75, 83, 89, 96$ |
| layers.2.CA | $0, 7, 18, 25, 32, 40, 45, 51, 74, 91$ |
| layers.2.FFN | $0, 7, 19, 31, 45, 54, 64, 74, 82, 90$ |
| layers.3.SA | $0, 6, 26, 40, 54, 64, 72, 82, 89, 96$ |
| layers.3.CA | $0, 11, 20, 25, 31, 42, 49, 59, 82, 93$ |
| layers.3.FFN | $0, 7, 21, 31, 45, 54, 65, 74, 82, 90$ |
| layers.4.SA | $0, 6, 32, 47, 59, 71, 81, 89, 94, 97$ |
| layers.4.CA | $0, 10, 19, 27, 39, 49, 65, 81, 90, 95$ |
| layers.4.FFN | $0, 6, 25, 39, 51, 65, 74, 85, 91, 96$ |
| layers.5.SA | $0, 6, 31, 54, 74, 87, 91, 94, 96, 98$ |
| layers.5.CA | $0, 14, 21, 38, 46, 51, 58, 66, 76, 86$ |
| layers.5.FFN | $0, 7, 17, 23, 45, 68, 79, 85, 91, 96$ |
| layers.6.SA | $0, 22, 54, 71, 81, 87, 91, 94, 96, 98$ |
| layers.6.CA | $0, 5, 10, 21, 32, 44, 51, 59, 73, 92$ |
| layers.6.FFN | $0, 7, 17, 23, 45, 64, 74, 84, 91, 96$ |
| layers.7.SA | $0, 21, 52, 72, 82, 87, 90, 93, 95, 97$ |
| layers.7.CA | $0, 14, 22, 39, 47, 53, 58, 79, 89, 96$ |
| layers.7.FFN | $0, 7, 19, 31, 45, 66, 86, 91, 94, 97$ |

Table 11: Update steps for $\text{Can}_{mh}$ computed by ACS.

| Block | Steps |
|-------|-------|
| layers.0.SA | $0, 7, 16, 26, 41, 52, 62, 75, 83, 92$ |
| layers.0.CA | $0, 14, 28, 43, 56, 70, 78, 80, 86, 93$ |
| layers.0.FFN | $0, 4, 11, 19, 29, 43, 54, 70, 81, 90$ |
| layers.1.SA | $0, 4, 10, 20, 29, 42, 54, 70, 81, 89$ |
| layers.1.CA | $0, 13, 26, 42, 55, 68, 78, 84, 90, 95$ |
| layers.1.FFN | $0, 5, 11, 19, 30, 43, 59, 69, 81, 92$ |
| layers.2.SA | $0, 5, 13, 29, 43, 59, 70, 78, 85, 92$ |
| layers.2.CA | $0, 9, 17, 26, 39, 54, 67, 77, 86, 96$ |
| layers.2.FFN | $0, 3, 11, 18, 24, 33, 43, 60, 80, 92$ |
| layers.3.SA | $0, 12, 25, 38, 49, 59, 69, 80, 86, 96$ |
| layers.3.CA | $0, 8, 16, 25, 37, 52, 63, 71, 87, 96$ |
| layers.3.FFN | $0, 3, 11, 19, 28, 43, 57, 80, 88, 96$ |
| layers.4.SA | $0, 9, 19, 30, 45, 63, 80, 87, 95, 98$ |
| layers.4.CA | $0, 3, 6, 13, 23, 38, 52, 63, 71, 84$ |
| layers.4.FFN | $0, 2, 11, 38, 62, 78, 85, 92, 97, 99$ |
| layers.5.SA | $0, 11, 34, 60, 80, 87, 92, 95, 97, 98$ |
| layers.5.CA | $0, 6, 19, 31, 44, 57, 74, 89, 95, 97$ |
| layers.5.FFN | $0, 54, 69, 81, 88, 91, 93, 95, 97, 99$ |
| layers.6.SA | $0, 60, 80, 89, 92, 94, 95, 96, 97, 98$ |
| layers.6.CA | $0, 6, 24, 46, 70, 89, 93, 95, 96, 98$ |
| layers.6.FFN | $0, 43, 59, 79, 87, 91, 94, 96, 97, 98$ |
| layers.7.SA | $0, 24, 59, 72, 81, 88, 91, 94, 96, 98$ |
| layers.7.CA | $0, 6, 28, 54, 69, 83, 90, 95, 97, 98$ |
| layers.7.FFN | $0, 60, 70, 73, 76, 80, 86, 95, 97, 99$ |

Table 12: Update steps for $\text{Lift}_{mh}$ computed by ACS.

| Block | Steps |
|---|---|
| layers.0.SA | $0, 3, 8, 21, 33, 49, 62, 69, 81, 91$ |
| layers.0.CA | $0, 14, 24, 36, 49, 60, 71, 80, 87, 94$ |
| layers.0.FFN | $0, 4, 8, 16, 28, 37, 53, 65, 79, 88$ |
| layers.1.SA | $0, 4, 8, 16, 27, 38, 51, 62, 79, 87$ |
| layers.1.CA | $0, 7, 16, 29, 41, 54, 67, 79, 88, 96$ |
| layers.1.FFN | $0, 4, 8, 14, 30, 38, 52, 65, 79, 88$ |
| layers.2.SA | $0, 4, 7, 12, 25, 33, 43, 60, 78, 90$ |
| layers.2.CA | $0, 9, 19, 30, 40, 49, 58, 69, 80, 90$ |
| layers.2.FFN | $0, 15, 23, 32, 43, 60, 68, 83, 93, 98$ |
| layers.3.SA | $0, 19, 34, 44, 54, 64, 78, 86, 94, 98$ |
| layers.3.CA | $0, 5, 12, 20, 28, 36, 47, 64, 76, 96$ |
| layers.3.FFN | $0, 4, 22, 40, 54, 68, 78, 83, 90, 96$ |
| layers.4.SA | $0, 5, 18, 32, 45, 62, 79, 89, 96, 99$ |
| layers.4.CA | $0, 5, 14, 25, 36, 47, 61, 79, 88, 95$ |
| layers.4.FFN | $0, 13, 40, 57, 68, 79, 89, 93, 96, 98$ |
| layers.5.SA | $0, 32, 56, 72, 83, 90, 94, 96, 98, 99$ |
| layers.5.CA | $0, 14, 31, 50, 66, 79, 87, 91, 95, 98$ |
| layers.5.FFN | $0, 33, 60, 75, 85, 90, 93, 95, 97, 99$ |
| layers.6.SA | $0, 34, 65, 80, 87, 91, 93, 95, 96, 98$ |
| layers.6.CA | $0, 9, 22, 38, 57, 78, 89, 93, 95, 97$ |
| layers.6.FFN | $0, 27, 51, 65, 81, 89, 92, 94, 96, 98$ |
| layers.7.SA | $0, 15, 32, 63, 80, 87, 93, 95, 96, 98$ |
| layers.7.CA | $0, 4, 21, 39, 55, 70, 84, 92, 95, 97$ |
| layers.7.FFN | $0, 43, 63, 68, 71, 75, 82, 92, 96, 98$ |

Table 13: Update steps for $\text{Square}_{mh}$ computed by ACS.

| Block | Steps |
|---|---|
| layers.0.SA | $0, 1, 3, 9, 18, 31, 51, 61, 76, 85$ |
| layers.0.CA | $0, 13, 26, 41, 54, 66, 77, 81, 86, 92$ |
| layers.0.FFN | $0, 3, 8, 18, 29, 40, 53, 66, 77, 88$ |
| layers.1.SA | $0, 1, 6, 12, 24, 31, 50, 66, 77, 87$ |
| layers.1.CA | $0, 7, 19, 34, 47, 58, 68, 78, 86, 92$ |
| layers.1.FFN | $0, 1, 7, 15, 24, 40, 50, 66, 77, 89$ |
| layers.2.SA | $0, 9, 24, 32, 49, 61, 71, 77, 84, 91$ |
| layers.2.CA | $0, 8, 21, 33, 46, 60, 72, 82, 90, 96$ |
| layers.2.FFN | $0, 3, 9, 23, 34, 44, 58, 66, 78, 93$ |
| layers.3.SA | $0, 9, 23, 33, 44, 55, 68, 78, 84, 96$ |
| layers.3.CA | $0, 8, 15, 24, 34, 47, 61, 77, 90, 96$ |
| layers.3.FFN | $0, 3, 10, 24, 44, 68, 78, 86, 92, 97$ |
| layers.4.SA | $0, 4, 14, 25, 41, 62, 78, 84, 92, 96$ |
| layers.4.CA | $0, 9, 15, 23, 34, 54, 76, 88, 93, 96$ |
| layers.4.FFN | $0, 3, 9, 20, 49, 69, 78, 84, 91, 97$ |
| layers.5.SA | $0, 12, 28, 48, 77, 85, 91, 94, 96, 98$ |
| layers.5.CA | $0, 6, 15, 26, 35, 56, 74, 89, 95, 97$ |
| layers.5.FFN | $0, 41, 76, 82, 86, 89, 91, 95, 97, 99$ |
| layers.6.SA | $0, 77, 87, 91, 93, 94, 95, 97, 98, 99$ |
| layers.6.CA | $0, 12, 23, 58, 84, 90, 93, 95, 97, 98$ |
| layers.6.FFN | $0, 6, 28, 58, 85, 93, 95, 96, 97, 98$ |
| layers.7.SA | $0, 20, 55, 73, 84, 90, 93, 96, 97, 98$ |
| layers.7.CA | $0, 3, 13, 30, 46, 83, 91, 94, 96, 98$ |
| layers.7.FFN | $0, 21, 58, 72, 77, 81, 85, 92, 95, 98$ |

Table 14: Update steps for Transport$_{mh}$ computed by ACS.

| Block | Steps |
|---|---|
| layers.0.SA | $0, 2, 7, 18, 30, 43, 58, 70, 80, 91$ |
| layers.0.CA | $0, 3, 5, 10, 24, 35, 62, 79, 93, 97$ |
| layers.0.FFN | $0, 1, 7, 18, 29, 51, 61, 71, 81, 93$ |
| layers.1.SA | $0, 4, 13, 25, 38, 51, 62, 76, 85, 93$ |
| layers.1.CA | $0, 6, 17, 29, 41, 53, 63, 76, 89, 95$ |
| layers.1.FFN | $0, 7, 18, 29, 40, 52, 62, 74, 86, 94$ |
| layers.2.SA | $0, 5, 18, 28, 46, 58, 70, 78, 90, 96$ |
| layers.2.CA | $0, 2, 10, 24, 37, 49, 59, 68, 77, 90$ |
| layers.2.FFN | $0, 3, 10, 24, 41, 52, 63, 78, 88, 95$ |
| layers.3.SA | $0, 5, 18, 29, 42, 58, 71, 80, 90, 94$ |
| layers.3.CA | $0, 5, 13, 22, 32, 44, 53, 64, 77, 91$ |
| layers.3.FFN | $0, 1, 6, 18, 30, 41, 52, 63, 78, 91$ |
| layers.4.SA | $0, 3, 10, 18, 30, 41, 55, 74, 81, 93$ |
| layers.4.CA | $0, 5, 13, 23, 31, 41, 53, 68, 78, 93$ |
| layers.4.FFN | $0, 1, 7, 18, 30, 41, 52, 68, 80, 91$ |
| layers.5.SA | $0, 1, 3, 10, 19, 31, 41, 54, 76, 91$ |
| layers.5.CA | $0, 6, 15, 25, 31, 36, 42, 52, 63, 78$ |
| layers.5.FFN | $0, 5, 19, 42, 66, 88, 92, 96, 98, 99$ |
| layers.6.SA | $0, 1, 8, 20, 41, 63, 75, 87, 97, 99$ |
| layers.6.CA | $0, 17, 36, 56, 77, 88, 93, 96, 98, 99$ |
| layers.6.FFN | $0, 1, 3, 10, 47, 63, 76, 90, 97, 98$ |
| layers.7.SA | $0, 1, 10, 46, 86, 90, 92, 94, 96, 98$ |
| layers.7.CA | $0, 1, 3, 10, 18, 42, 52, 75, 84, 97$ |
| layers.7.FFN | $0, 3, 10, 17, 35, 46, 63, 76, 90, 97$ |

Table 15: Update steps for Block Pushing computed by ACS.

| Block | Steps |
|---|---|
| layers.0.SA | $0, 1, 2, 4, 14, 25, 32, 40, 62, 74$ |
| layers.0.CA | $0, 3, 13, 26, 41, 57, 73, 83, 91, 97$ |
| layers.0.FFN | $0, 1, 3, 7, 13, 18, 26, 59, 74, 82$ |
| layers.1.SA | $0, 1, 7, 13, 18, 25, 40, 59, 74, 90$ |
| layers.1.CA | $0, 4, 12, 26, 42, 58, 72, 83, 92, 97$ |
| layers.1.FFN | $0, 3, 7, 16, 26, 40, 59, 76, 83, 92$ |
| layers.2.SA | $0, 3, 7, 16, 25, 40, 57, 74, 83, 96$ |
| layers.2.CA | $0, 4, 7, 12, 19, 32, 48, 66, 84, 95$ |
| layers.2.FFN | $0, 3, 7, 17, 26, 40, 59, 74, 86, 96$ |
| layers.3.SA | $0, 5, 13, 26, 41, 52, 70, 81, 86, 96$ |
| layers.3.CA | $0, 3, 6, 9, 10, 12, 22, 56, 81, 98$ |
| layers.3.FFN | $0, 5, 10, 18, 26, 52, 71, 83, 92, 97$ |
| layers.4.SA | $0, 4, 9, 16, 26, 41, 53, 71, 83, 95$ |
| layers.4.CA | $0, 3, 9, 16, 25, 36, 51, 68, 82, 93$ |
| layers.4.FFN | $0, 7, 16, 30, 41, 53, 70, 83, 92, 97$ |
| layers.5.SA | $0, 5, 13, 25, 40, 53, 71, 83, 92, 97$ |
| layers.5.CA | $0, 5, 11, 20, 28, 42, 53, 71, 81, 90$ |
| layers.5.FFN | $0, 5, 15, 27, 46, 58, 71, 83, 92, 97$ |
| layers.6.SA | $0, 11, 25, 38, 53, 70, 81, 88, 93, 97$ |
| layers.6.CA | $0, 3, 6, 10, 27, 42, 64, 75, 84, 93$ |
| layers.6.FFN | $0, 17, 38, 53, 64, 74, 83, 89, 93, 97$ |
| layers.7.SA | $0, 25, 51, 66, 75, 81, 86, 90, 93, 97$ |
| layers.7.CA | $0, 7, 14, 26, 49, 71, 83, 91, 95, 97$ |
| layers.7.FFN | $0, 16, 40, 64, 76, 83, 89, 93, 96, 98$ |

Table 16: Update steps for Kitchen computed by ACS.

| Block | Steps |
|---|---|
| layers.0.SA | 0, 2, 5, 10, 19, 29, 43, 54, 70, 83 |
| layers.0.CA | 0, 1, 6, 18, 29, 41, 54, 68, 85, 95 |
| layers.0.FFN | 0, 3, 7, 14, 25, 38, 46, 59, 70, 84 |
| layers.1.SA | 0, 4, 10, 19, 28, 40, 49, 59, 75, 87 |
| layers.1.CA | 0, 1, 4, 8, 11, 19, 30, 40, 49, 61 |
| layers.1.FFN | 0, 5, 10, 18, 29, 43, 54, 68, 78, 91 |
| layers.2.SA | 0, 4, 10, 19, 30, 41, 54, 72, 84, 92 |
| layers.2.CA | 0, 2, 5, 8, 13, 19, 30, 41, 54, 70 |
| layers.2.FFN | 0, 1, 5, 10, 18, 27, 41, 55, 70, 87 |
| layers.3.SA | 0, 1, 7, 18, 29, 41, 55, 68, 82, 92 |
| layers.3.CA | 0, 3, 6, 10, 13, 19, 31, 41, 55, 73 |
| layers.3.FFN | 0, 2, 5, 8, 11, 18, 27, 41, 59, 79 |
| layers.4.SA | 0, 4, 10, 18, 31, 41, 59, 72, 84, 92 |
| layers.4.CA | 0, 1, 7, 15, 27, 38, 49, 59, 70, 82 |
| layers.4.FFN | 0, 2, 5, 11, 18, 27, 41, 54, 70, 88 |
| layers.5.SA | 0, 3, 8, 15, 25, 37, 46, 61, 78, 91 |
| layers.5.CA | 0, 3, 13, 22, 32, 43, 54, 65, 76, 92 |
| layers.5.FFN | 0, 1, 5, 11, 18, 27, 43, 57, 72, 88 |
| layers.6.SA | 0, 12, 28, 41, 52, 62, 72, 84, 92, 98 |
| layers.6.CA | 0, 1, 6, 13, 21, 28, 40, 54, 69, 92 |
| layers.6.FFN | 0, 1, 5, 18, 29, 47, 59, 92, 96, 98 |
| layers.7.SA | 0, 2, 41, 62, 80, 89, 93, 95, 97, 98 |
| layers.7.CA | 0, 7, 16, 40, 53, 60, 76, 88, 92, 98 |
| layers.7.FFN | 0, 1, 6, 11, 25, 38, 60, 68, 88, 95 |

Table 17: Update steps added for $\text{Can}_{ph}$ after BUA.

| Block | Added Steps |
|---|---|
| layers.0.FFN | 6, 8, 14, 16, 18, 23, 27, 28, 37, 38, 39, 47, 48, 51, 60, 62, 68, 69, 71, 74, 75, 77, 78, 80, 81, 85, 86, 89, 92, 93, 94, 95, 96, 97, 98, 99 |
| layers.5.FFN | 28, 47, 62, 69, 74, 77, 80, 81, 86, 89, 93, 95, 97 |
| layers.6.SA | 28, 47, 62, 69, 74, 77 |
| layers.6.FFN | 47, 69, 74, 77, 81, 86 |

Table 18: Update steps added for $\text{Lift}_{ph}$ after BUA.

| Block | Added Steps |
|---|---|
| layers.0.FFN | 10, 14, 15, 17, 18, 28, 29, 31, 40, 44, 48, 51, 54, 57, 65, 68, 69, 78, 79, 86, 91, 93, 95, 96, 97, 98, 99 |
| layers.1.SA | 14, 15, 17, 18, 27, 29, 31, 40, 44, 48, 49, 51, 54, 57, 68, 69, 74, 79, 86, 91, 93, 95, 96, 97, 98, 99 |
| layers.1.FFN | 14, 15, 17, 27, 29, 31, 40, 44, 48, 49, 51, 57, 68, 69, 74, 78, 86, 91, 93, 95, 96, 97, 98, 99 |
| layers.2.FFN | 14, 15, 29, 31, 40, 44, 48, 49, 54, 57, 68, 69, 74, 79, 86, 91, 93, 95, 96, 97, 98, 99 |
| layers.3.FFN | 15, 29, 37, 40, 44, 48, 54, 57, 69, 74, 86, 91, 95, 96, 98, 99 |

Table 19: Update steps added for Square$_{ph}$ after BUA.

| Block | Added Steps |
| --- | --- |
| layers.0.FFN | 7, 8, 14, 15, 23, 24, 36, 37, 44, 48, 52, 53, 57, 61, 64, 67, 69, 74, 76, 78, 79, 83, 87, 89, 90, 91, 93, 95, 96, 97, 98 |
| layers.6.FFN | 10, 52, 78, 87, 89, 91, 97 |
| layers.7.SA | 10, 52, 78, 83, 87, 89, 97 |
| layers.7.CA | 10, 52, 78, 83, 89, 91 |

Table 20: Update steps added for Transport$_{ph}$ after BUA.

| Block | Added Steps |
| --- | --- |
| layers.0.FFN | 1, 7, 12, 14, 16, 19, 28, 29, 30, 34, 36, 43, 49, 52, 57, 60, 62, 63, 67, 69, 70, 74, 75, 78, 79, 82, 83, 84, 85, 87, 88, 90, 91, 92, 93, 94, 95, 96, 97, 98 |
| layers.3.SA | 14, 16, 28, 29, 34, 36, 43, 49, 57, 60, 62, 63, 67, 70, 74, 75, 78, 79, 82, 85, 87, 88, 92, 93, 96, 98 |
| layers.3.CA | 14, 16, 28, 34, 36, 43, 49, 57, 60, 62, 63, 67, 70, 74, 75, 78, 79, 82, 85, 87, 88, 92, 93, 94, 95, 97, 98 |
| layers.3.FFN | 14, 16, 28, 29, 34, 36, 49, 57, 60, 62, 63, 67, 70, 75, 78, 79, 88, 92, 93, 94, 96, 97 |
| layers.4.FFN | 14, 28, 29, 36, 49, 60, 62, 63, 67, 70, 75, 79, 82, 85, 88, 95, 96, 98 |

Table 21: Update steps added for Tool$_{ph}$ after BUA.

| Block | Added Steps |
| --- | --- |
| layers.0.FFN | 3, 6, 11, 12, 16, 23, 24, 31, 44, 48, 49, 50, 53, 60, 61, 63, 66, 71, 74, 77, 78, 79, 80, 81, 83, 85, 86, 87, 88, 91, 93, 94, 95, 96, 97, 98, 99 |
| layers.1.FFN | 3, 6, 11, 12, 16, 23, 24, 31, 44, 48, 49, 50, 53, 60, 61, 63, 66, 71, 74, 77, 78, 79, 80, 81, 83, 85, 86, 87, 88, 91, 93, 94, 95, 96, 97, 98, 99 |
| layers.5.FFN | 11, 40, 53, 60, 74, 79, 81, 83, 87, 95, 97, 99 |
| layers.6.SA | 11, 40, 53, 60, 71, 79, 80, 81, 87, 91, 97, 99 |
| layers.6.CA | 11, 40, 53, 60, 71, 74, 79, 80, 81, 83, 87, 91, 95, 97, 99 |

Table 22: Update steps added for Pusht-T after BUA.

| Block | Added Steps |
| --- | --- |
| layers.0.FFN | 3, 6, 19, 21, 23, 31, 33, 39, 47, 51, 54, 59, 64, 65, 66, 68, 79, 82, 84, 85, 86, 90, 91, 94, 96, 97 |
| layers.4.FFN | 7, 17, 19, 23, 31, 45, 64, 66, 68, 79, 84, 86, 94, 97 |
| layers.5.FFN | 19, 31, 64, 66, 74, 84, 86, 94, 97 |
| layers.6.FFN | 19, 31, 66, 86, 94, 97 |

Table 23: Update steps added for $\text{Can}_{mh}$ after BUA.

| Block | Added Steps |
|---|---|
| layers.0.FFN | 2, 3, 5, 18, 24, 28, 30, 33, 38, 57, 59, 60, 62, 69, 73, 76, 78, 79, 80, 85, 86, 87, 88, 91, 92, 93, 94, 95, 96, 97, 98, 99 |
| layers.5.FFN | 43, 59, 60, 70, 73, 76, 79, 80, 86, 87, 94, 96, 98 |
| layers.6.SA | 43, 59, 70, 73, 76, 79, 86, 87, 91, 99 |
| layers.6.FFN | 60, 70, 73, 76, 80, 86, 95, 99 |

Table 24: Update steps added for $\text{Lift}_{mh}$ after BUA.

| Block | Added Steps |
|---|---|
| layers.0.FFN | 13, 14, 15, 22, 23, 27, 30, 32, 33, 38, 40, 43, 51, 52, 54, 57, 60, 63, 68, 71, 75, 78, 81, 82, 83, 85, 89, 90, 92, 93, 94, 95, 96, 97, 98, 99 |
| layers.5.SA | 27, 33, 43, 51, 60, 63, 65, 68, 71, 75, 81, 82, 85, 89, 92, 93, 95, 97 |
| layers.5.CA | 27, 33, 43, 51, 60, 63, 65, 68, 71, 75, 81, 82, 85, 89, 90, 92, 93, 94, 96, 97, 99 |
| layers.5.FFN | 27, 43, 51, 63, 65, 68, 71, 81, 82, 89, 92, 94, 96, 98 |

Table 25: Update steps added for $\text{Square}_{mh}$ after BUA.

| Block | Added Steps |
|---|---|
| layers.0.FFN | 1, 6, 7, 9, 10, 15, 20, 21, 23, 24, 28, 34, 41, 44, 49, 50, 58, 68, 69, 72, 76, 78, 81, 82, 84, 85, 86, 89, 91, 92, 93, 95, 96, 97, 98, 99 |
| layers.5.FFN | 6, 21, 28, 58, 72, 77, 81, 85, 92, 93, 96, 98 |
| layers.6.SA | 6, 21, 28, 58, 72, 81, 85, 92, 96 |
| layers.6.CA | 6, 21, 28, 72, 77, 81, 85, 92, 96 |

Table 26: Update steps added for $\text{Transport}_{mh}$ after BUA.

| Block | Added Steps |
|---|---|
| layers.0.SA | 1, 3, 5, 6, 10, 17, 19, 24, 29, 35, 40, 41, 42, 46, 47, 51, 52, 61, 62, 63, 66, 68, 71, 74, 76, 78, 81, 86, 88, 90, 92, 93, 94, 95, 96, 97, 98, 99 |
| layers.0.FFN | 3, 5, 6, 10, 17, 19, 24, 30, 35, 40, 41, 42, 46, 47, 52, 62, 63, 66, 68, 74, 76, 78, 80, 86, 88, 90, 91, 92, 94, 95, 96, 97, 98, 99 |
| layers.1.FFN | 1, 3, 5, 6, 10, 17, 19, 24, 30, 35, 41, 42, 46, 47, 63, 66, 68, 76, 78, 80, 88, 90, 91, 92, 95, 96, 97, 98, 99 |
| layers.5.FFN | 1, 3, 10, 17, 35, 46, 47, 63, 76, 90, 97 |
| layers.6.FFN | 17, 35, 46 |

Table 27: Update steps added for Block Pushing after BUA.

| Block | Added Steps |
|---|---|
| layers.0.FFN | 5, 10, 15, 16, 17, 27, 30, 38, 40, 41, 46, 52, 53, 58, 64, 70, 71, 76, 83, 86, 89, 92, 93, 96, 97, 98 |
| layers.6.FFN | 16, 40, 76, 96, 98 |
| layers.7.SA | 16, 40, 64, 76, 83, 89, 96, 98 |
| layers.7.CA | 16, 40, 64, 76, 89, 93, 96, 98 |

Table 28: Update steps added for Kitchen after BUA.

| Block | Added Steps |
|---|---|
| layers.0.SA | 1, 3, 6, 7, 8, 11, 14, 18, 25, 27, 38, 41, 46, 47, 55, 57, 59, 60, 68, 72, 78, 79, 84, 87, 88, 91, 92, 95, 96, 98 |
| layers.0.FFN | 1, 2, 5, 6, 8, 10, 11, 18, 27, 29, 41, 43, 47, 54, 55, 57, 60, 68, 72, 78, 79, 87, 88, 91, 92, 95, 96, 98 |
| layers.5.FFN | 6, 25, 29, 38, 47, 59, 60, 68, 92, 95, 96, 98 |
| layers.6.FFN | 6, 11, 25, 38, 60, 68, 88, 95 |
| layers.7.SA | 1, 6, 11, 25, 38, 60, 68, 88 |

## A.14 Visualization of Cache Behavior

We visualize the ground-truth features, the computed and cached features under the BAC schedule, and their absolute differences for each timestep in Figures 16 through 25, with cache update steps marked in red. Across all tasks, we observe two consistent phenomena. First, consecutive steps exhibit high feature similarity, confirming that high temporal redundancy makes caching naturally applicable. Second, the difference maps reveal distinct behaviors between reuse and update phases: during cache reuse, the absolute difference remains low, reflecting activation stability. Conversely, cache update steps show significant feature shifts, indicating that updates are effectively capturing necessary changes. Collectively, these visualizations validate the reliability of the BAC cache schedule.

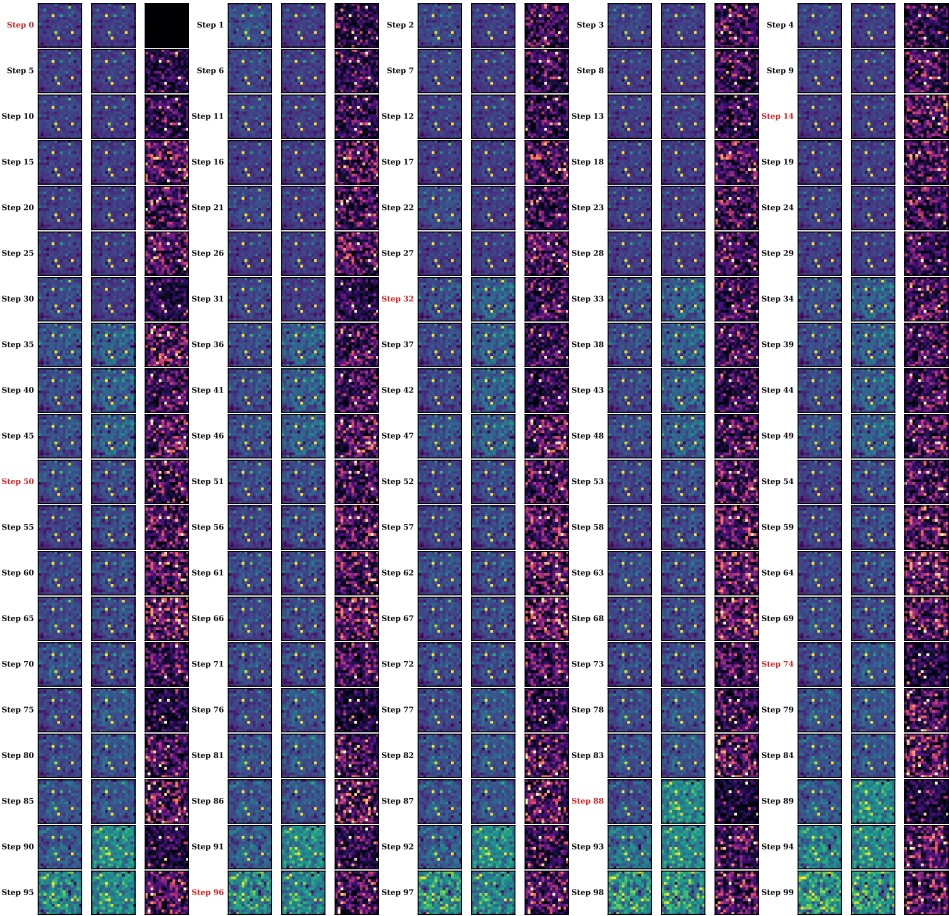

Figure 16: Feature heatmaps for Lift_ph.

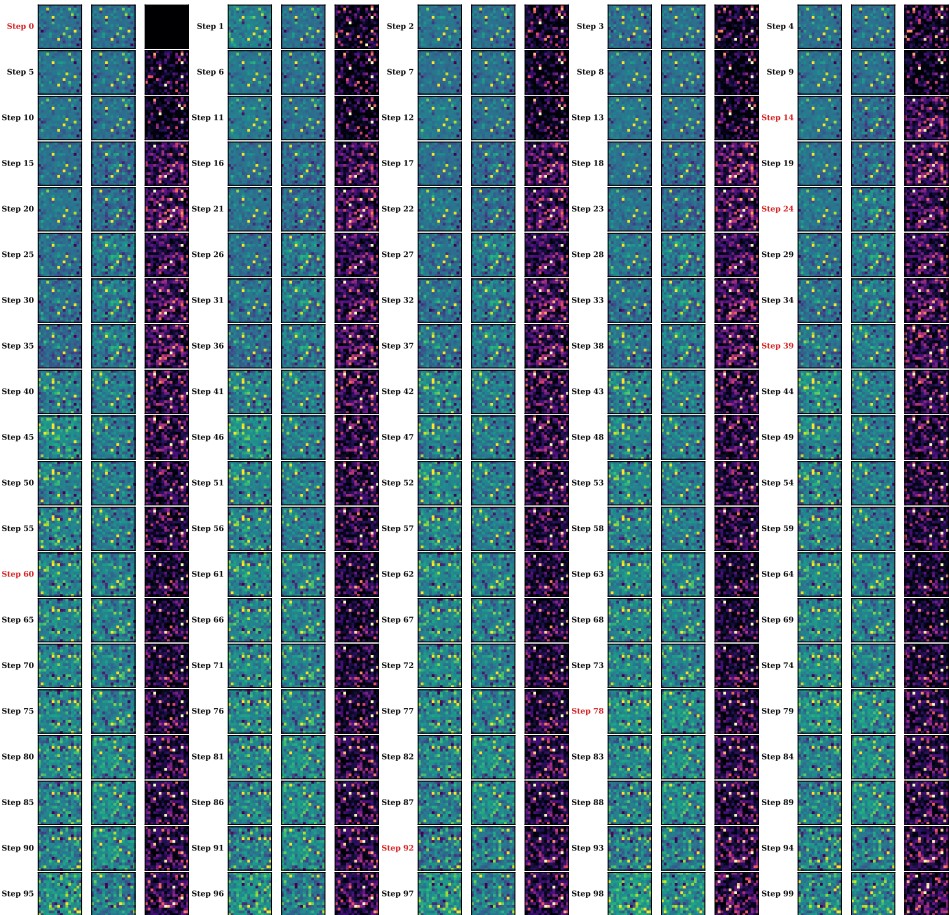

Figure 17: Feature heatmaps for Can_ph.

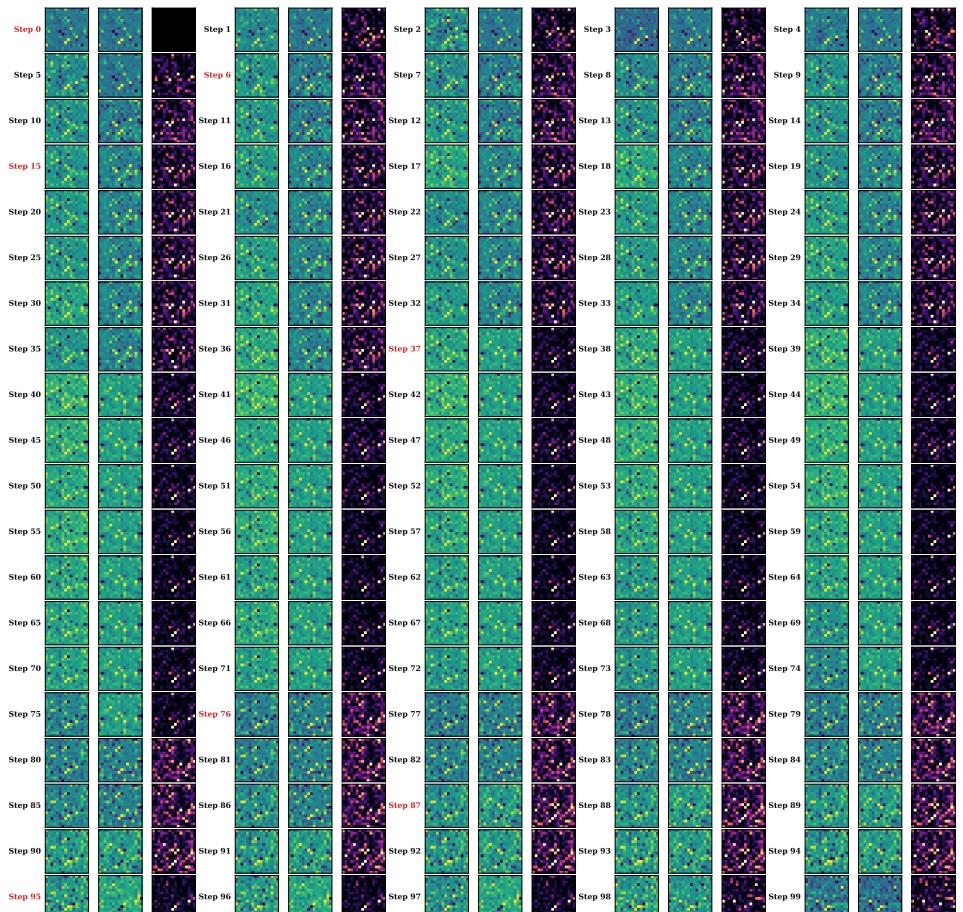

Figure 18: Feature heatmaps for Square_ph.

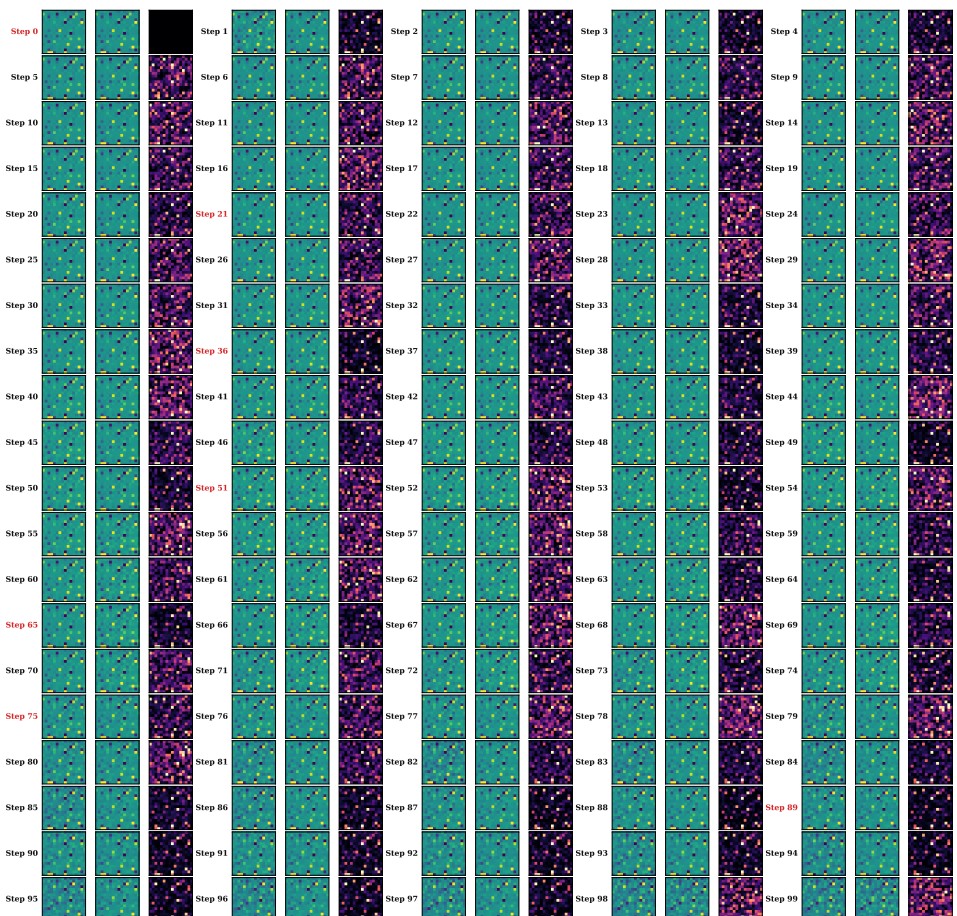

Figure 19: Feature heatmaps for Pusht.

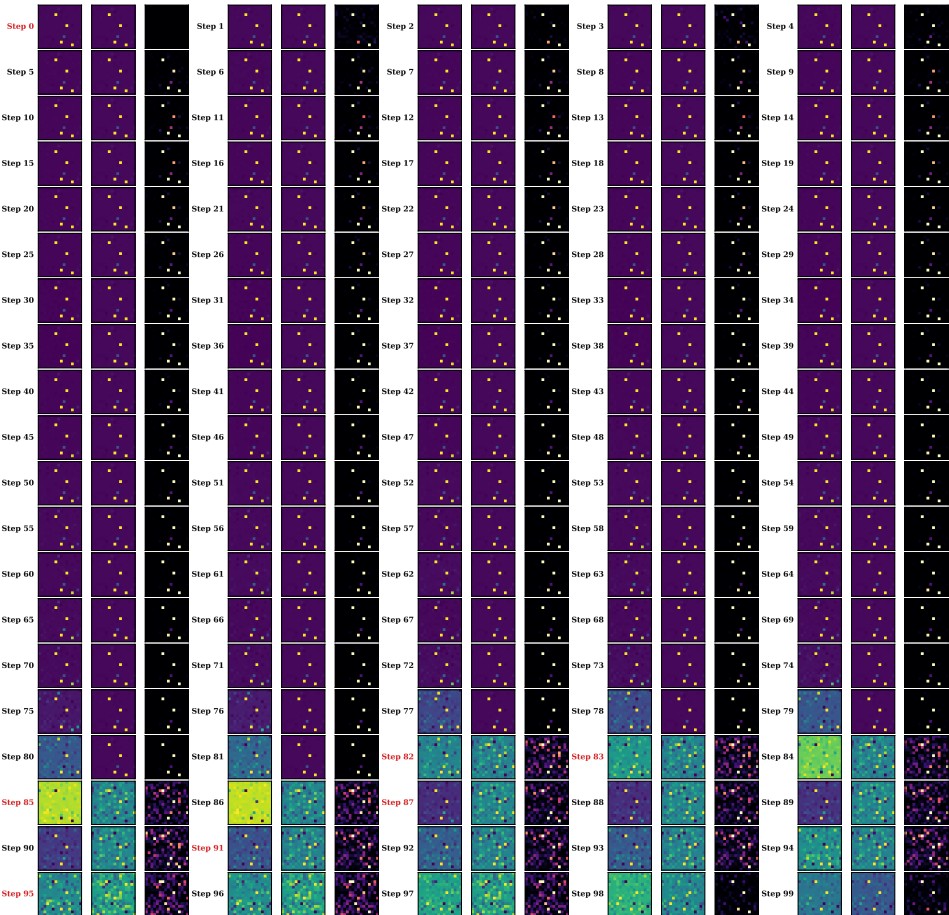

Figure 20: Feature heatmaps for Transport_ph.

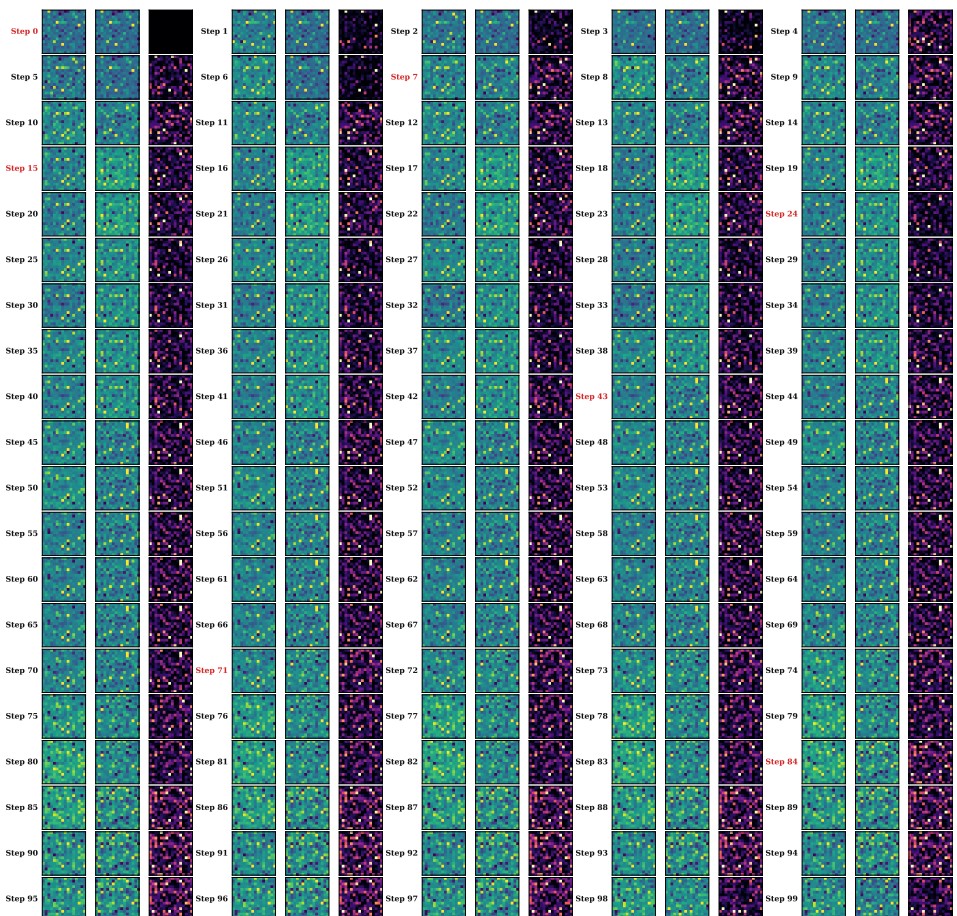

Figure 21: Feature heatmaps for Tool_hang.

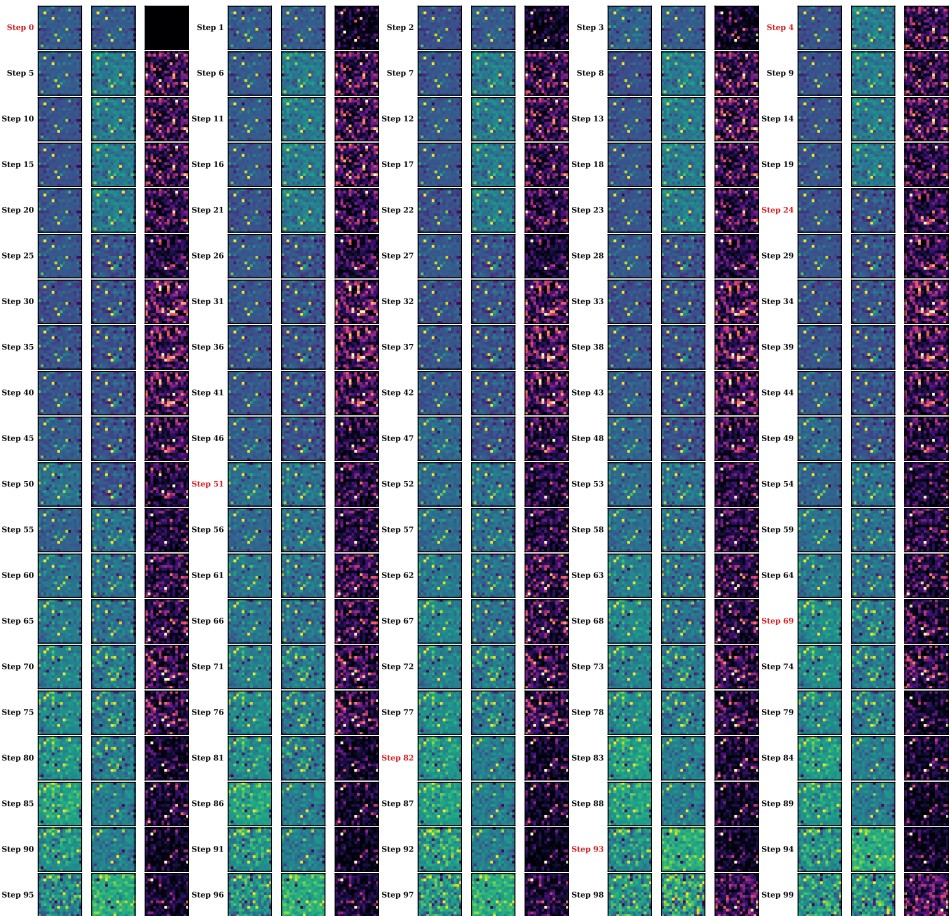

Figure 22: Feature heatmaps for Lift_mh.

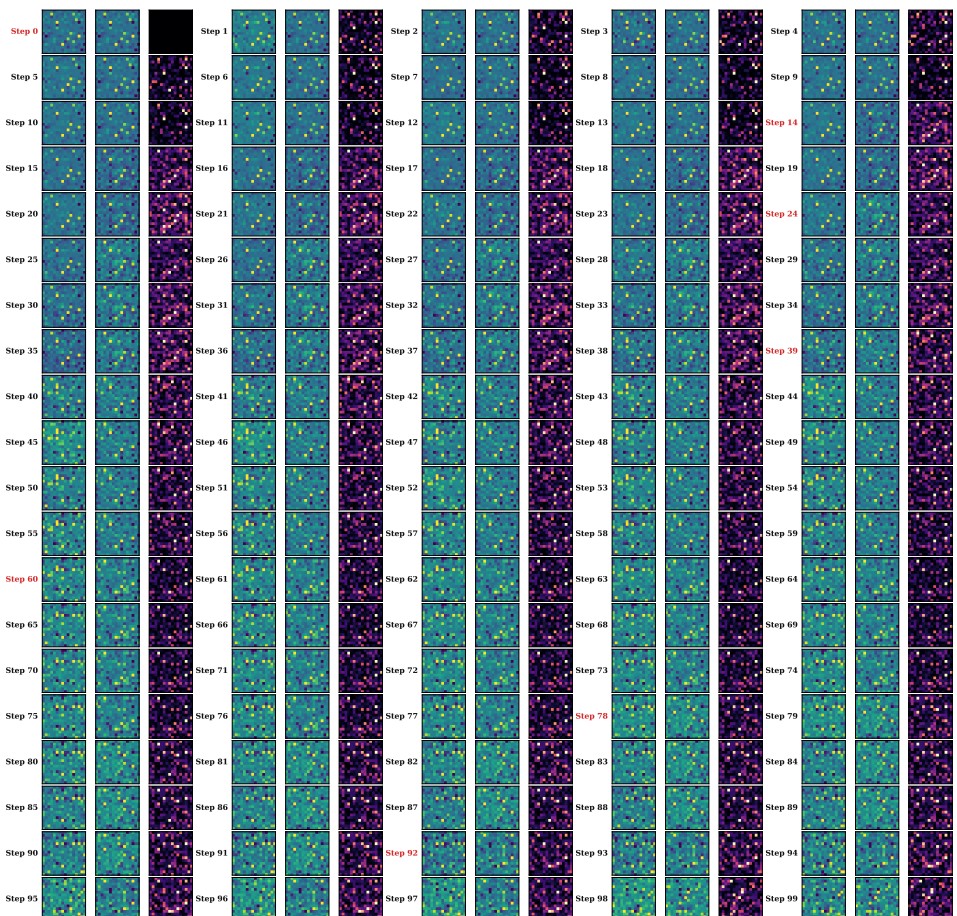

Figure 23: Feature heatmaps for Can_mh.

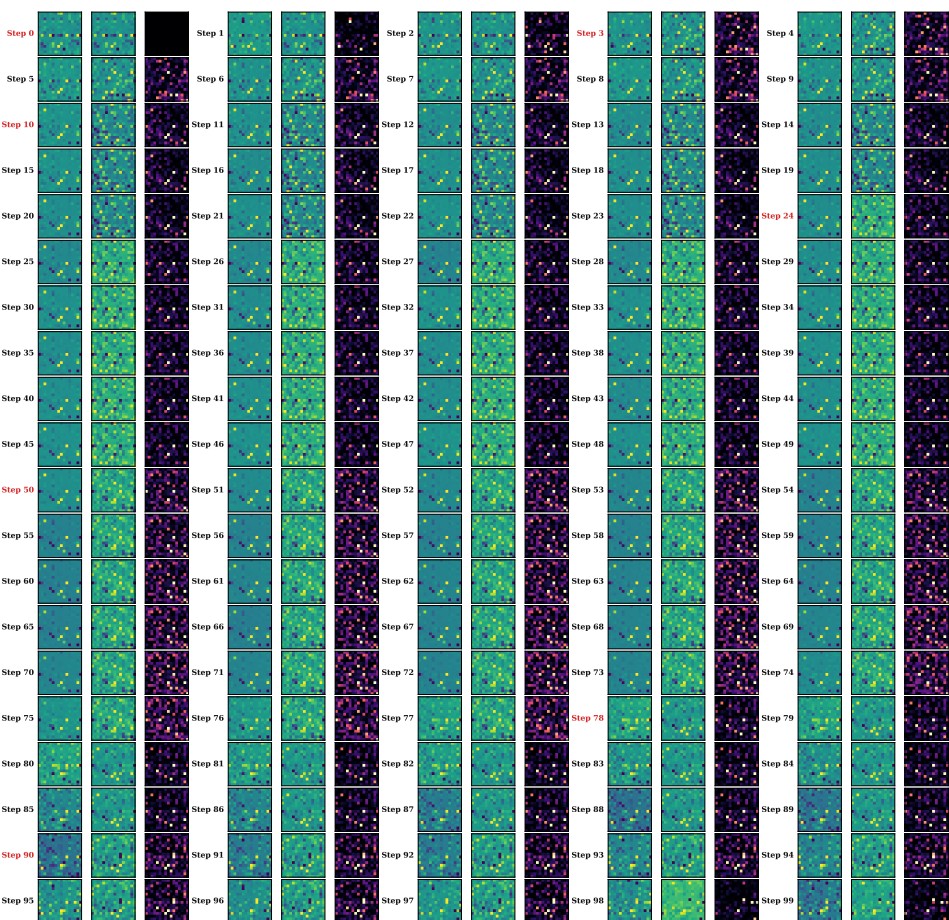

Figure 24: Feature heatmaps for Square_mh.

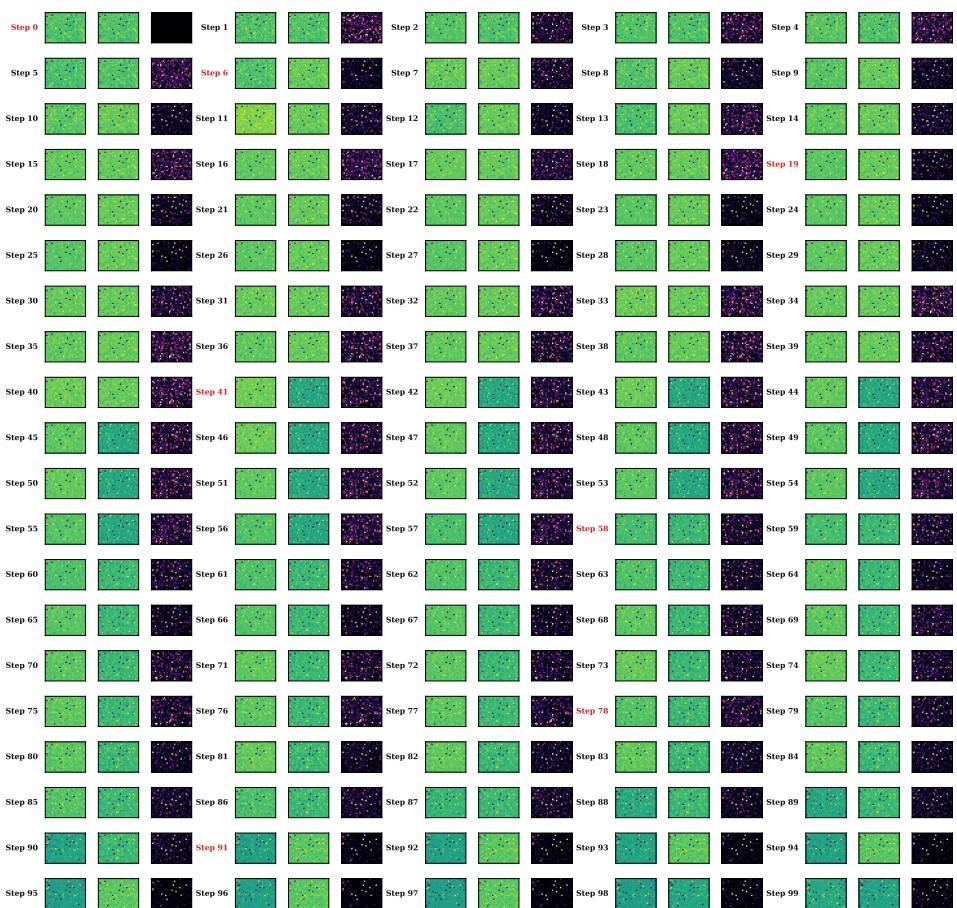

Figure 25: Feature heatmaps for Transport_mh.

## A.15 LARGE, DIVERSE EVIDENCE FOR HIGH EPISODE HOMOGENEITY

In this section, we provide both qualitative and quantitative evidence for high episode homogeneity.

In qualitative experiments, we visualize the similarity heatmap for the features between episodes in Figures 27 to 32. Specifically, for each task, we regenerate 50 episodes and extract features from 8 layers across 10 uniformly sampled timesteps. We compute a $50 \times 50$ cosine-similarity matrix across episodes, producing 80 matrices per task, giving a comprehensive view on episode hemogeneity across multimodal trajectories. Fig. 26 illustrates an example of the computed similarity matrix. Each entry represents the similarity between the features of two independent episodes. To make the visualization clearer, the range of all maps is restricted to $[0.8, 1.0]$. These figures clearly demonstrate a strong homogeneity across different episodes.

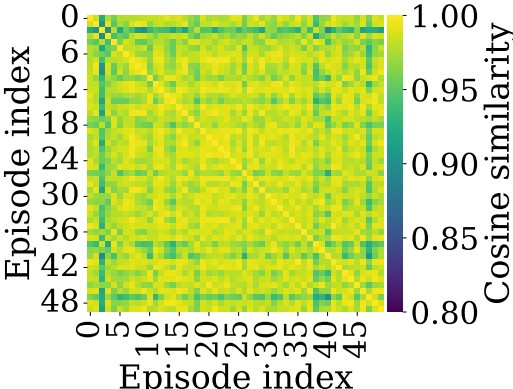

Figure 26: Inter-episode feature similarity for the Lift task at the FFN block of layer 7 at timestep 99.

In quantitative analysis, we conduct statistics on inter-episode similarities. As shown in Table 29, the average per-matrix mean similarity stays $\geq 0.96$ with 95th-percentile entries $\geq 0.99$, indicating

the similarity curve is essentially flat. Moreover, we find that differences across episodes emerge almost exclusively within the final 5% of denoising steps, this indicates that the vast majority of the diffusion process remains highly consistent across episodes, with only minor divergence introduced during the final stage. These results reflect a repeatable property of the denoising dynamics of the model.

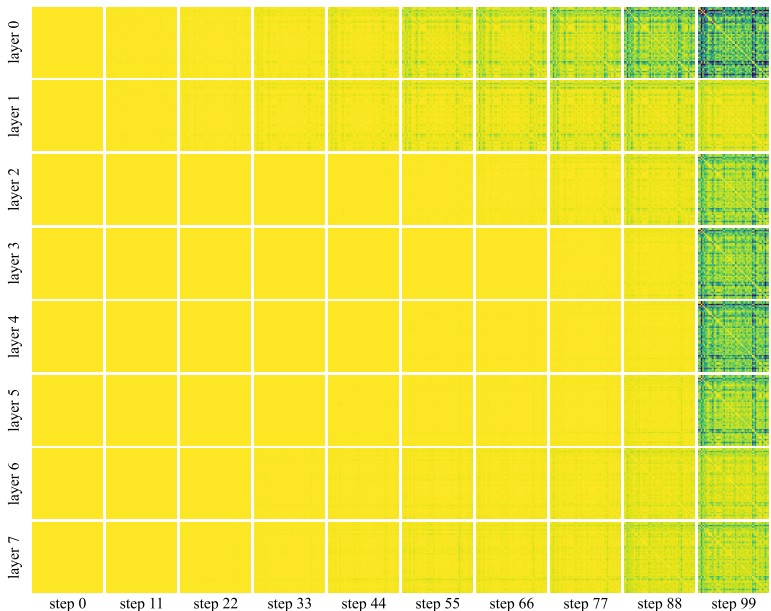

Figure 27: Inter-episode similarity pattern for Lift.

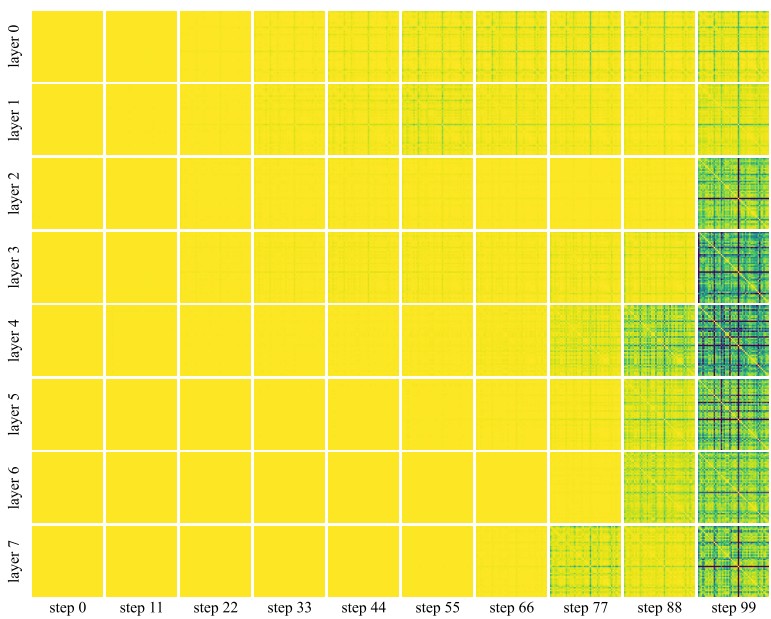

Figure 28: Inter-episode similarity pattern for Can.

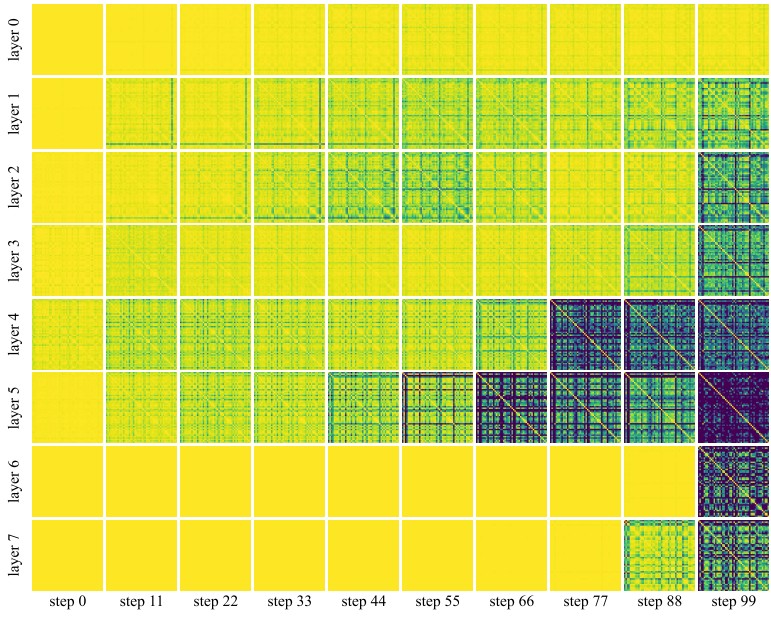

Figure 29: Inter-episode similarity pattern for Square.

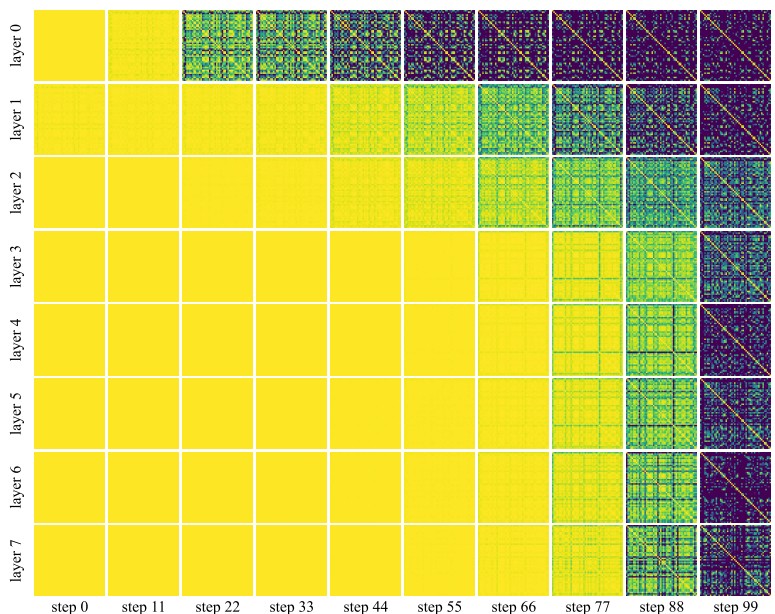

Figure 30: Inter-episode similarity pattern for Pusht.

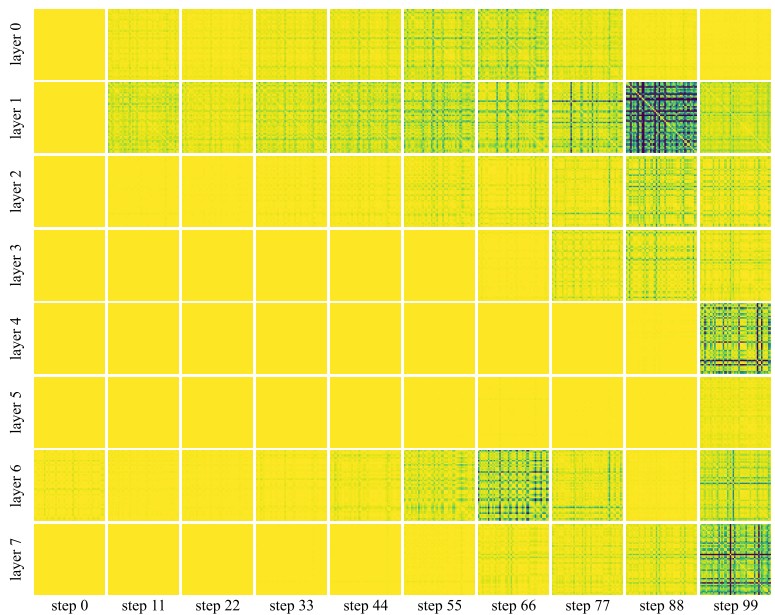

Figure 31: Inter-episode similarity pattern for Transport.

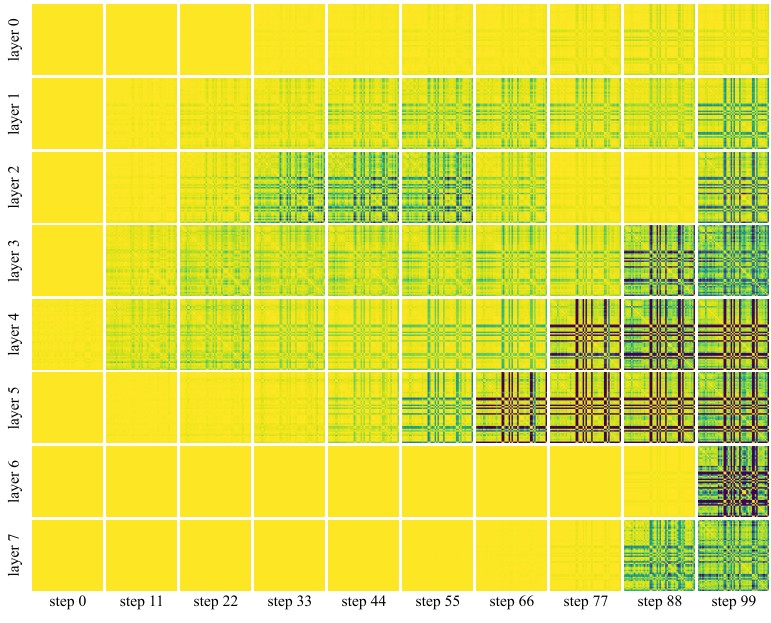

Figure 32: Inter-episode similarity pattern for Tool_hang.

| Task | AvgMean | MinMean | AvgStd | MaxStd | AvgP05 | AvgP95 |
|------|---------|---------|--------|--------|--------|--------|
| Can_mh | 0.9910 | 0.8206 | 0.0061 | 0.1414 | 0.9789 | 0.9980 |
| Lift_mh | 0.9767 | 0.7352 | 0.0156 | 0.1625 | 0.9465 | 0.9954 |
| Square_mh | 0.9744 | 0.6398 | 0.0268 | 0.2050 | 0.9291 | 0.9949 |
| Transport_mh | 0.9981 | 0.9393 | 0.0026 | 0.1017 | 0.9922 | 0.9998 |
| Can_ph | 0.9925 | 0.9049 | 0.0067 | 0.0769 | 0.9782 | 0.9986 |
| Lift_ph | 0.9937 | 0.9254 | 0.0048 | 0.0552 | 0.9839 | 0.9988 |
| Pusht | 0.9285 | 0.2885 | 0.0526 | 0.5010 | 0.8277 | 0.9930 |
| Square_ph | 0.9622 | 0.6829 | 0.0333 | 0.2729 | 0.8917 | 0.9936 |
| Tool_hang_ph | 0.9751 | 0.6957 | 0.0290 | 0.4019 | 0.9195 | 0.9983 |
| Transport_ph | 0.9920 | 0.8627 | 0.0085 | 0.1348 | 0.9752 | 0.9991 |

Table 29: Inter-episode similarity statistics across tasks.

## A.16 EVALUATING BAC UNDER DDIM SCHEDULERS

In this section, we investigate the performance of BAC when integrated with sampling-based methods, specifically focusing on the DDIM scheduler with a step size of $K = 40$. We apply BAC on top of DDIM across different update steps ($\mathcal{S} = 3, 5, 10$) and report the results in terms of Success Rate and Speedup.

Table 30: Performance of BAC integrated with DDIM ($K = 40$) across various benchmarks. Results are presented as (Success Rate / Speedup).

Benchmark on Proficient Human (PH) demonstration data.

| Method | Lift | Can | Square | Transport | Tool Hang | Push–T |
|--------|------|-----|--------|-----------|-----------|--------|
| DDIM ($K = 40$) | 1.00/1.00 | 0.95/1.00 | 0.84/1.00 | 0.75/1.00 | 0.47/1.00 | 0.55/1.00 |
| BAC ($\mathcal{S} = 3, n = 3$) | 1.00/3.22 | 0.56/3.49 | 0.64/3.67 | 0.00/3.14 | 0.00/3.43 | 0.44/3.15 |
| BAC ($\mathcal{S} = 5, n = 3$) | 1.00/2.41 | 0.95/3.22 | 0.87/2.85 | 0.58/2.97 | 0.30/3.08 | 0.53/3.23 |
| BAC ($\mathcal{S} = 10, n = 3$) | 1.00/2.32 | 0.94/2.41 | 0.86/2.53 | 0.76/2.33 | 0.47/2.38 | 0.55/2.48 |

Benchmark on Mixed Human (MH) demonstration data.

| Method | Lift | Can | Square | Transport |
|--------|------|-----|--------|-----------|
| DDIM ($K = 40$) | 0.99/1.00 | 0.93/1.00 | 0.74/1.00 | 0.34/1.00 |
| BAC ($\mathcal{S} = 3, n = 3$) | 0.98/3.44 | 0.54/3.63 | 0.04/3.71 | 0.22/3.55 |
| BAC ($\mathcal{S} = 5, n = 3$) | 0.98/3.14 | 0.87/3.23 | 0.68/3.33 | 0.30/2.94 |
| BAC ($\mathcal{S} = 10, n = 3$) | 0.99/2.29 | 0.90/2.43 | 0.75/2.45 | 0.33/2.20 |

Benchmark on multi-stage tasks. For Block-Pushing, $p_x$ denotes pushing $x$ blocks. For Kitchen, $p_x$ denotes interacting with $x$ or more objects.

| Method | Kitchen $p_1$ | Kitchen $p_2$ | Kitchen $p_3$ | Kitchen $p_4$ | BP $p_1$ | BP $p_2$ |
|--------|---------------|---------------|---------------|---------------|----------|----------|
| DDIM ($K = 40$) | 1.00/1.00 | 1.00/1.00 | 1.00/1.00 | 0.96/1.00 | 0.98/1.00 | 0.94/1.00 |
| BAC ($\mathcal{S} = 3, n = 3$) | 0.99/3.56 | 0.99/3.56 | 0.99/3.56 | 0.97/3.56 | 0.94/3.27 | 0.95/3.43 |
| BAC ($\mathcal{S} = 5, n = 3$) | 1.00/2.99 | 1.00/2.99 | 1.00/2.99 | 0.99/2.99 | 0.94/2.95 | 0.95/3.16 |
| BAC ($\mathcal{S} = 10, n = 3$) | 1.00/2.23 | 1.00/2.23 | 1.00/2.23 | 0.97/2.23 | 0.98/2.23 | 0.97/2.36 |

Even when integrated with the faster DDIM scheduler, the BAC algorithm demonstrates a significant speedup of 2.2 to $3.5\times$ while maintaining high success rates. This result emphasizes that the efficiency gains provided by BAC are not merely due to reductions in denoising steps, but rather stem from the inherent acceleration capabilities of BAC itself.

