# OpenReview forum: "Block-wise Adaptive Caching for Accelerating Diffusion Policy"
_ICLR.cc/2026/Conference — ICLR 2026 Poster_

### Official Review · Reviewer_N17f · 2025-10-22

**Soundness:** 3
**Presentation:** 3
**Contribution:** 3
**Rating:** 4
**Confidence:** 4

**Summary:**

The paper introduces Block-wise Adaptive Caching (BAC), a training-free method designed to accelerate inference for transformer-based Diffusion Policy models in robotic control. The method is composed of two primary components: an Adaptive Caching Scheduler (ACS), which employs dynamic programming to compute an optimal, non-uniform schedule of cache update steps for each transformer block by maximizing feature similarity; and a Bubbling Union Algorithm (BUA). The authors identify a novel "error surge phenomenon," where naive block-wise scheduling leads to catastrophic error propagation, particularly in Feed-Forward Network (FFN) blocks. The BUA is proposed as a targeted solution to mitigate this issue by enforcing upstream blocks with high estimated errors to update their caches in synchrony with their downstream FFN counterparts. Through experiments on several robotic manipulation benchmarks, the authors claim that BAC achieves up to a 3x inference speedup with negligible, or even slightly improved, performance compared to the full-precision model.

**Strengths:**

- While caching in diffusion models is an existing area of research, the primary contribution of this paper—the identification, analysis, and targeted mitigation of the inter-block "error surge phenomenon"—is novel and insightful. The proposed Bubbling Union Algorithm (BUA), motivated by this analysis, is an original and simple heuristic for a previously undocumented problem.

- The use of dynamic programming within the Adaptive Caching Scheduler (ACS) to find an optimal schedule under a well-defined objective function (maximizing feature similarity) is a principled and sound approach.

- The authors validate their method across a commendable range of benchmarks (RoboMimic, Push-T, Kitchen, etc.), data sources (PH, MH), and even extend it to a large-scale Vision-Language-Action model (RDT-1B), providing evidence for the potential generalizability of the approach.

**Weaknesses:**

- Insufficient Evidence for the Core Assumption of Homogeneity: The entire offline-online paradigm of the proposed method hinges on the crucial assumption of "high episode homogeneity"—that a cache schedule computed from one trajectory can generalize to others. The evidence provided for this cornerstone assumption is extraordinarily weak, consisting of a visual comparison between only two demonstration instances (demo 11001 vs. 20000) in Appendix A.9. Generalizing a universal property from two samples is scientifically unsound and constitutes anecdotal evidence at best. This fails to adequately address the challenge that the stochastic nature of diffusion models can lead to fundamentally different trajectories (i.e., different action modalities), which may invalidate a statically computed schedule.
- Absence of Real-World Hardware Experiments: All experiments are conducted in simulation, but the ultimate goal of robotic policy learning is deployment on physical hardware. The paper makes strong claims about enabling "real-time robotic control", but without a single real-world experiment, these claims remain unsubstantiated.

**Questions:**

1. The core assumption of "episode homogeneity" is foundational to your method but is supported by only two examples. Could you provide a more systematic study across a large, diverse set of episodes and initial random noises for a given task to statistically validate that the feature similarity structure ($s_k$ curve) remains stable? How does your offline schedule contend with the explicitly multimodal, stochastic nature of diffusion policy?

2. Could you provide qualitative video results comparing the behavior of a robot arm controlled by the baseline versus the BAC-accelerated policy? This would help visually verify that the claimed speedup does not introduce detrimental artifacts. Furthermore, do you have plans for or preliminary results from deploying this method on a physical robot to validate its real-world effectiveness and truly substantiate the claims of enabling real-time control?

---

> ### Author Response · Authors · 2025-11-26
> **Response to Reviewer N17f (Part 1)**
>
> Thank you for the time and effort in providing a thorough evaluation of our work. We appreciate your valuable comments and address specific concerns below.
>
>
> **W1 & Q1. Insufficient Evidence for the Core Assumption of Homogeneity**
>
> > Insufficient Evidence for the Core Assumption of Homogeneity: The entire offline-online paradigm of the proposed method hinges on the crucial assumption of "high episode homogeneity"—that a cache schedule computed from one trajectory can generalize to others. The evidence provided for this cornerstone assumption is extraordinarily weak, consisting of a visual comparison between only two demonstration instances (demo 11001 vs. 20000) in Appendix A.9. Generalizing a universal property from two samples is scientifically unsound and constitutes anecdotal evidence at best. This fails to adequately address the challenge that the stochastic nature of diffusion models can lead to fundamentally different trajectories (i.e., different action modalities), which may invalidate a statically computed schedule.
>
>
> **A1.**
> We thank the reviewer for this insightful observation. To address the concern that the high episode homogeneity assumption was only illustrated with two examples, we conducted a large-scale quantitative and qualitative analysis across all tasks in Appendix A.15.
>
> **Inter-episode similarity statistics across tasks**
> | Task | AvgMean | MinMean | AvgStd | MaxStd | AvgP05 | AvgP95 |
> | --- | --- |  --- | --- | --- | --- | --- |
> | Can_mh        | 0.9910 | 0.8206 | 0.0061 | 0.1414 | 0.9789 | 0.9980 |
> | Lift_mh       |  0.9767 | 0.7352 | 0.0156 | 0.1625 | 0.9465 | 0.9954 |
> | Square_mh     |  0.9744 | 0.6398 | 0.0268 | 0.2050 | 0.9291 | 0.9949 |
> | Transport_mh  |  0.9981 | 0.9393 | 0.0026 | 0.1017 | 0.9922 | 0.9998 |
> | Can_ph        |  0.9925 | 0.9049 | 0.0067 | 0.0769 | 0.9782 | 0.9986 |
> | Lift_ph       |  0.9937 | 0.9254 | 0.0048 | 0.0552 | 0.9839 | 0.9988 |
> | Pusht         |  0.9285 | 0.2885 | 0.0526 | 0.5010 | 0.8277 | 0.9930 |
> | Square_ph     |  0.9622 | 0.6829 | 0.0333 | 0.2729 | 0.8917 | 0.9936 |
> | Tool_hang_ph  |  0.9751 | 0.6957 | 0.0290 | 0.4019 | 0.9195 | 0.9983 |
> | Transport_ph  |  0.9920 | 0.8627 | 0.0085 | 0.1348 | 0.9752 | 0.9991 |
>
>
>
> In qualitative experiments, we visualize the similarity heatmap for the features between episodes in Figures 26 to 31. Specifically, for each task, we regenerate 50 episodes and extract features from 8 layers across 10 uniformly sampled timesteps. We compute a 50 $\times$ 50 cosine-similarity matrix across episodes, producing 80 matrices per task, giving a comprehensive view on episode hemogeneity across multimodal trajectories. These figures clearly demonstrate a strong homogeneity across different episodes.
>
> In quantitative analysis, we conduct statistics on inter-episode similarities. As shown in the table, the average per-matrix mean similarity stays $\geq$ 0.96 with 95th-percentile entries $\geq$ 0.99, indicating high similarity in inference pattern for different episodes. Moreover, we find that differences across episodes emerge almost exclusively within the final ~5\% of denoising steps. This indicates that the vast majority of the diffusion process remains highly consistent across episodes, with only minor divergence introduced during the final stage.
>
> These results reflect a consistent property of the denoising dynamics of the model, providing stronger evidence for high inter-episode homogeneity.

---

> ### Author Response · Authors · 2025-11-26
> **Response to Reviewer N17f (Part 2)**
>
> **W2 & Q2. Real-World Experiments and Qualitative Video Results**
> > All experiments are conducted in simulation, but the ultimate goal of robotic policy learning is deployment on physical hardware. The paper makes strong claims about enabling "real-time robotic control", but without a single real-world experiment, these claims remain unsubstantiated.
>
>
> **A2.** We thank the reviewer for raising this important concern regarding the absence. To address this, we have conducted real-world experiments to validate our method. The deployment videos and experimental results are now available on the project website: https://block-wise-adaptive-caching.github.io.
>
> > Could you provide qualitative video results comparing the behavior of a robot arm controlled by the baseline versus the BAC-accelerated policy? This would help visually verify that the claimed speedup does not introduce detrimental artifacts.
>
> Thank you for the suggestion! We have uploaded video demonstrations on our project website, which compare the behavior of the robot arm controlled by the baselines versus the BAC-accelerated policy. These videos clearly show that BAC leads to faster and smoother manipulation, without introducing any detrimental artifacts or destabilizing the robot’s movements. You can view the videos on the [project website](https://block-wise-adaptive-caching.github.io).
>
> > Furthermore, do you have plans for or preliminary results from deploying this method on a physical robot to validate its real-world effectiveness and truly substantiate the claims of enabling real-time control?
>
> Yes, we have deployed our approach on "Pick-and-Release" task, conducted on a Franka Research 3 robot arm. The real-world evaluation results demonstrate that our method achieves a wall-clock frequency of up to 45.1 Hz and significantly reduces task latency, achieving remarkably 63% and 71% success, as shown in the table below:
>
> | Method                    | Wall-Clock Frequency | End-to-End Latency | Success Rate |
> | :------------------------ | :------------------- | :----------------- | :----------- |
> | DDPM (Steps = 100)          | 7.8 Hz              | 180 s              | 3%           |
> | DDPM (Steps = 30)           | 19.4 Hz             | 54 s               | 22%          |
> | DDIM (Steps = 50)           | 15.6 Hz             | 25 s               | 52%          |
> | Uniform ($\mathcal S=20$) | 28.1 Hz             | 52 s               | 40%          |
> | BAC ($\mathcal S=7, n=3$) | 39.2 Hz             | 15 s               | 71%          |
> | BAC ($\mathcal S=5, n=3$) | 45.1 Hz             | 13 s               | 63%          |
>
> In real-world experiments, we observe significant failures with the DDPM and Uniform methods, which contrast sharply with their performance in simulation. This failure may be attributed to state obsolescence, as the inference frequency is much lower than the control frequency. By the time an action is generated, the robot's input state and capped observations are outdated (e.g., over 900 ms for each action chunk, generated by DDPM with 100 steps), causing the generated chunk to roll back and wander. These results underscore the critical need for real-time robotic control in physical environments, which requires both efficient generation and high fidelity. We have supplemented the real-world experimental results in the revised version.

---

> > ### Comment · Reviewer_N17f · 2025-11-27
> >
> > The authors have addressed most of my questions. I increased the score for this paper.

---

> > > ### Author Response · Authors · 2025-11-28
> > >
> > > We sincerely appreciate the positive feedback and are glad that our responses have addressed your concerns. We value your acknowledgment of the novelty and generalizability of BAC, and we thank you for the increased rating.

---

### Official Review · Reviewer_o36f · 2025-10-30

**Soundness:** 3
**Presentation:** 3
**Contribution:** 3
**Rating:** 6
**Confidence:** 3

**Summary:**

This paper proposes Block-wise Adaptive Caching (BAC), a training-free method to accelerate Transformer-based Diffusion Policies for real-time robot control. BAC adaptively caches and reuses intermediate block features using a dynamic scheduling algorithm and a bubbling mechanism to prevent error propagation. It achieves up to 3× faster inference without performance loss across multiple robotics benchmarks.

**Strengths:**

1. This paper proposes a method that achieves a 3× speedup, which is an important step toward making diffusion policies truly practical for real-world robotic control.
2. The paper offers an analysis of the internal feature dynamics of Diffusion Policy, revealing that feature similarity across timesteps is non-uniform among different Transformer blocks (SA, CA, FFN), which provides a solid foundation for the proposed block-wise adaptive method.
3.  Another major highlight is the discovery and explanation of the “Error Surge” phenomenon: naive block-level caching causes severe degradation in FFN blocks, and the authors convincingly attribute this to the amplification and propagation of upstream errors due to the absence of LayerNorm. This insight is broadly relevant to all Transformer-based caching approaches.

**Weaknesses:**

Proposition 3.1 is the paper’s only theoretical contribution, aiming to explain error propagation in the FFN blocks. While the first-order Taylor expansion (Appendix A.4) is directionally correct, it does not provide any quantitative bound, nor does it explain why the SA/CA blocks (which also contain $W_{out}$) do not exhibit the same error surge phenomenon.

**Questions:**

In Stage 1 of BUA, the method identifies “high-error” upstream blocks using a global $L_1$ norm (Eq. 13), which measures the average feature variation across all timesteps. However, the “error surge” analyzed in Section 3.3 is a localized event that occurs when an FFN block updates while its upstream block reuses a high-error cache at the same timestep. Why  is a global average metric $l_j$ the right choice for addressing an instantaneous error propagation problem? Wouldn’t a more appropriate measure be the instantaneous cache error at the specific timesteps where the FFN depends on the block?

---

> ### Author Response · Authors · 2025-11-26
> **Response to Reviewer o36f**
>
> Thank you for the time and effort in providing a thorough evaluation of our work. We appreciate your valuable comments and address specific concerns below.
>
> **W1. Quantitative Bound and Block-level Comparsion**
>
> > Proposition 3.1 is the paper’s only theoretical contribution, aiming to explain error propagation in the FFN blocks. While the first-order Taylor expansion (Appendix A.4) is directionally correct, it does not provide any quantitative bound, nor does it explain why the SA/CA blocks (which also contain ) do not exhibit the same error surge phenomenon.
>
> **A1.** We thank the reviewer for this insightful comment. We wish to clarify that the primary goal of this proposition was to formulate the mechanism of error amplification as a linear response operator. Built on this, the BUA algorithm avoids the amplification simply by reducing the original input error.
>
> Since the algorithmic design is not tightly coupled to the specific mechanisms inside FFN blocks, we provide a more in-depth discussion of the FFN-specific surge phenomenon in Appendix A.5. We summarize several potential contributing factors below.
>
> In FFN blocks, errors propagate sequentially through a layer normalization, the first linear transformation $W_1$, a GELU activation function, and the second linear transformation $W_2$. In contrast, error propagation in attention blocks involves layer normalization, the query $W_Q$, key $W_K$, and value $W_V$ projection matrices, and an output projection matrix that aggregates the multi-head attention outputs.
>
> As illustrated in Figure 3(b), a critical observation is that the two linear layers within FFN blocks, particularly the second layer, tend to significantly amplify upstream errors. Builing on the formulation, we further conducted additional experiments to examine the Frobenius norm distributions of weights across different blocks. We found that the magnitudes of $W_1$ and $W_2$ are approximately three times larger than those of $W_V$, as shown in Figure 7. This substantial difference in weight magnitudes likely contributes to the observed error surges within FFN blocks. Furthermore, the absence of intermediate normalization steps between the two linear layers in FFN blocks may exacerbate error amplification. Another intuitive explanation is that, compared with the transformations between latent spaces, the attention mechanisms are less sensitive to the errors.
>
> Importantly, BUA empirically identifies these surges and mitigates them by effectively reducing upstream errors. We appreciate the reviewer’s question and believe that future extensions of BUA may incorporate architectural-level insights to further address this phenomenon.
>
> ---
>
> **Q1. Error Level Metric.**
>
> >In Stage 1 of BUA, the method identifies “high-error” upstream blocks using a global
>  norm (Eq. 13), which measures the average feature variation across all timesteps. However, the “error surge” analyzed in Section 3.3 is a localized event that occurs when an FFN block updates while its upstream block reuses a high-error cache at the same timestep. Why is a global average metric
>  the right choice for addressing an instantaneous error propagation problem? Wouldn’t a more appropriate measure be the instantaneous cache error at the specific timesteps where the FFN depends on the block?
>
>
> **A2.** We agree with the reviewer that, in principle, the most appropriate metric would be the instantaneous upstream cache error for the to-be-updated FFN blocks. However, such a metric is fundamentally difficult to obtain in practice. The instantaneous cache error cannot be reliably simulated from the caching schedule alone, because its magnitude depends on the continuously evolving environmental dynamics during deployment. Consequently, directly estimating the true instantaneous error requires online access to ground-truth activations, which is infeasible in accelerated inference. We also note that prior attempts to estimate cache error online (e.g., TeaCache) have demonstrated limited performance, precisely because accurately fitting instantaneous error in real time is extremely challenging.
>
> For this reason, we approximate the upstream error level using an offline activation profiling procedure. This gives us a stable global indicator of which blocks tend to demonstrate large deviations in inference.
> Importantly, in Section 3.3 we do compute the “correct” instantaneous errors by measuring the distance between the accelerated and original model outputs. This analysis is used only to reveal and validate the error-propagation mechanism—not as a metric for BAC itself, since BAC is an offline sampling algorithm and cannot access these oracle quantities at test time.

---

### Official Review · Reviewer_p5ox · 2025-11-03

**Soundness:** 3
**Presentation:** 3
**Contribution:** 3
**Rating:** 6
**Confidence:** 3

**Summary:**

his paper introduces Block-wise Adaptive Caching (BAC), a training-free acceleration method for transformer-based Diffusion Policy models used in robotic control. Diffusion Policy models are powerful but computationally expensive due to their iterative denoising process. BAC addresses this by adaptively caching intermediate features at the block level, significantly reducing redundant computation without sacrificing performance.
Key Contributions:
1.	Block-wise Adaptive Caching (BAC): A novel caching strategy that adaptively updates and reuses features at the block level, tailored for Diffusion Policy architectures.
2.	Adaptive Caching Scheduler (ACS): Uses dynamic programming to determine the optimal update timesteps for each block, maximizing feature similarity and minimizing caching error.
3.	Bubbling Union Algorithm (BUA): Mitigates inter-block error propagation, especially in FFN blocks, by enforcing upstream updates when downstream blocks are updated.
4.	Extensive Experimental Validation: Demonstrates up to 3.55× speedup with no performance loss across multiple robotic tasks and models, including large-scale vision-language-action (VLA) models like RDT-1B.
5.	Training-Free and Plug-and-Play: BAC is easy to integrate into existing models without retraining or architectural changes.

**Strengths:**

This paper presents an original contribution by demonstrating that a block-level, training-free caching mechanism can significantly reduce the inference time of Diffusion Policy without compromising accuracy. Its strength lies in a clear problem statement—addressing the well-known 10 Hz to 30 Hz performance gap—and a novel insight: non-uniform temporal similarity across transformer blocks necessitates per-block caching schedules, while upstream errors can propagate and amplify into downstream FFN failures if not properly trapped. The authors support this with a concise error-propagation model and exhaustive simulations, ranging from small control tasks to a 1-B parameter VLA model, consistently achieving a 3× speedup with reproducible artifacts. Although the idea builds upon existing caching concepts, its novel reframing for Diffusion Policy, plug-and-play practicality, and immediate real-time performance gains render this work both high-quality and broadly significant for deploying generative models under strict latency constraints.

**Weaknesses:**

My primary concern is the absence of physical validation, as all experiments were conducted in simulation. Based on lessons from computer vision, we know that improvements in GPU-FLOPS do not always translate to reduced real-world latency. A brief hardware trace—even for a single arm and task—would help demonstrate that the acceleration persists in the presence of motor noise and timing jitter. I would also appreciate a comparison with standard diffusion acceleration techniques to validate the unique advantage of the proposed robot cache.

That said, the core idea is intuitive, the evaluation is thorough within the simulation framework, and the commitment to open-source the code is commendable. Should the authors add at least minimal real-robot validation and discuss the cache's robustness under more aggressive quantization or fewer denoising steps, I believe the paper would be of great value to the community.

**Questions:**

Q1. Practical Performance on Physical Hardware
The reported GPU-time improvements in simulation are promising. However, in real-world robotic systems, such gains can diminish due to factors like camera latency, USB bandwidth limitations, or motor jitter. Could the authors provide any empirical timing results—even from a single task on one robot arm—showing the actual wall-clock frequency achieved in a lab setting? This would help validate the practical applicability of the method.

Q2. Comparison with Fewer Denoising Steps
In image diffusion models, reducing denoising steps (e.g., from 1000 to 20) is a common acceleration strategy while maintaining acceptable output quality. Could the authors include a comparative analysis—such as a small plot—illustrating performance when using only 15 or 20 steps, both with and without BAC? This would clarify whether the proposed caching mechanism offers advantages beyond simply executing fewer denoising iterations.

Q3. Visualization of Cache Behavior
As a visual learner, I would find it greatly helpful to see a figure illustrating the cache's effect—for instance, an attention map or feature heatmap at a step where the cache decided not to update. Such a visualization would intuitively convey the underlying "similarity" concept without relying solely on mathematical formulations.

Q4. Hyperparameter Tuning for New Tasks
The paper mentions using S = 7 or 10. If a user applies the method to a new robot task, are there practical guidelines—such as "start with S = 8"—to avoid extensive hyperparameter search? A brief recommendation in the paper would greatly improve usability.

Q5. Code Accessibility
The commitment to open-source the code is appreciated. Would it be possible to share a GitHub link during the rebuttal period—even if kept private temporarily—so that reviewers can examine the license and potentially run a simple local example?

---

> ### Author Response · Authors · 2025-11-26
> **Response to Reviewer p5ox (Part 1)**
>
> Thank you for the time and effort in providing a thorough evaluation of our work. We appreciate your valuable comments and address specific concerns below.
>
> **W1 & Q1. The absence of physical validation.**
>
> > Practical Performance on Physical Hardware The reported GPU-time improvements in simulation are promising. However, in real-world robotic systems, such gains can diminish due to factors like camera latency, USB bandwidth limitations, or motor jitter. Could the authors provide any empirical timing results—even from a single task on one robot arm—showing the actual wall-clock frequency achieved in a lab setting? This would help validate the practical applicability of the method.
>
> **A1.** We thank the reviewer for highlighting the importance of validating our method on physical hardware. To address this, we have deployed our approach on a real-world robotic system. Detailed experimental results, along with real-world deployment videos, are available on our project website: [https://block-wise-adaptive-caching.github.io](https://block-wise-adaptive-caching.github.io).
>
> The experiments were conducted using a Franka Research 3 robot arm, equipped with a UGREEN CM717 camera. The inference was performed on a workstation with an NVIDIA GeForce RTX 4090D (48GB). We report the wall-clock frequency (Hz) and end-to-end task latency for a "Pick-and-Release" task. The real-world evaluation results are summarized in the table below:
>
> | Method                    | Wall-Clock Frequency | End-to-End Latency | Success Rate |
> | :------------------------ | :------------------- | :----------------- | :----------- |
> | DDPM (Steps = 100)          | 7.8 Hz              | 180 s              | 3%           |
> | DDPM (Steps = 30)           | 19.4 Hz             | 54 s               | 22%          |
> | DDIM (Steps = 50)           | 15.6 Hz             | 25 s               | 52%          |
> | Uniform ($\mathcal S=20$) | 28.1 Hz             | 52 s               | 40%          |
> | BAC ($\mathcal S=7, n=3$) | 39.2 Hz             | 15 s               | 71%          |
> | BAC ($\mathcal S=5, n=3$) | 45.1 Hz             | 13 s               | 63%          |
>
> The BAC method achieves a maximum inference frequency of 45.1 Hz, demonstrating significantly better performance and highlighting its practical applicability. These results underscore the demand for higher frequency operations in real-world robotic tasks. Real-world Experimental results have been contained in the new version of manuscript.
>
> ---

---

> ### Author Response · Authors · 2025-11-26
> **Response to Reviewer p5ox (Part 2)**
>
> **Q2. Comparison with Fewer Denoising Steps In image diffusion models.**
> > Comparison with Fewer Denoising Steps In image diffusion models, reducing denoising steps (e.g., from 1000 to 20) is a common acceleration strategy while maintaining acceptable output quality. Could the authors include a comparative analysis—such as a small plot—illustrating performance when using only 15 or 20 steps, both with and without BAC? This would clarify whether the proposed caching mechanism offers advantages beyond simply executing fewer denoising iterations.
>
>
> **A2.** To address the reviewer’s concern that the acceleration achieved by BAC might be explained simply by reducing the number of denoising iterations, we evaluate Diffusion Policy under a DDIM sampler, which already reduces the denoising steps from 100 to 40. We then apply BAC on top of DDIM with different update steps ($\mathcal{S}=3, 5, 10$). We report the results as follows with values formatted as **Success Rate / Speedup**.
>
> **PH (Proficient Human Data) Benchmark**
>
> | Method| Lift | Can | Square | Transport | Tool Hang | Push-T |
> | :---------------:| :---: | :---: | :---: | :---: | :---: | :---: |
> | DDIM (Steps=40) | 1.00 / 1.00 | 0.95 / 1.00 | 0.84 / 1.00 | 0.75 / 1.00 | 0.47 / 1.00 | 0.55 / 1.00 |
> | BAC($\mathcal S=3, n=3$) | 1.00 / 3.22 | 0.56 / 3.49 | 0.64 / 3.67 | 0.00 / 3.14 | 0.00 / 3.43 | 0.44 / 3.15 |
> | BAC ($\mathcal S=5, n=3$) | 1.00 / 2.41 | 0.95 / 3.22 | 0.87 / 2.85 | 0.58 / 2.97 | 0.30 / 3.08 | 0.53 / 3.23 |
> | BAC ($\mathcal S=10, n=3$) | 1.00 / 2.32 | 0.94 / 2.41 | 0.86 / 2.53 | 0.76 / 2.33 | 0.47 / 2.38 | 0.55 / 2.48 |
>
> **MH (Mixed Human Data) Benchmark**
>
> | Method | Lift | Can | Square | Transport |
> | :---: | :---: | :---: | :---: | :---: |
> | DDIM (Steps=40) | 0.99 / 1.00 | 0.93 / 1.00 | 0.74 / 1.00 | 0.34 / 1.00 |
> | BAC ($\mathcal S=3, n=3$) | 0.98 / 3.44 | 0.54 / 3.63 | 0.04 / 3.71 | 0.22 / 3.55 |
> | BAC ($\mathcal S=5, n=3$) | 0.98 / 3.14 | 0.87 / 3.23 | 0.68 / 3.33 | 0.30 / 2.94 |
> | BAC ($\mathcal S=10, n=3$) | 0.99 / 2.29 | 0.90 / 2.43 | 0.75 / 2.45 | 0.33 / 2.20 |
>
> **Multi-stage Benchmark**
>
> | Method | Kitchen ${p_1}$ | Kitchen $p_2$ | Kitchen $p_3$ | Kitchen $p_4$ | BP $p_1$ | BP $p_2$ |
> | :--- | :---: | :---: | :---: | :---: | :---: | :---: |
> | DDIM (Steps=40) | 1.00 / 1.00 | 1.00 / 1.00 | 1.00 / 1.00 | 0.96 / 1.00 | 0.98 / 1.00 | 0.94 / 1.00 |
> | BAC ($\mathcal S=3, n=3$) | 0.99 / 3.56 | 0.99 / 3.56 | 0.99 / 3.56 | 0.97 / 3.56 | 0.94 / 3.27 | 0.95 / 3.43 |
> | BAC ($\mathcal S=5, n=3$) | 1.00 / 2.99 | 1.00 / 2.99 | 1.00 / 2.99 | 0.99 / 2.99 | 0.94 / 2.95 | 0.95 / 3.16 |
> | BAC ($\mathcal S=10, n=3$) | 1.00 / 2.23 | 1.00 / 2.23 | 1.00 / 2.23 | 0.97 / 2.23 | 0.98 / 2.23 | 0.97 / 2.36 |
>
> Even on top of the faster DDIM scheduler, BAC yields an additional $2.2–3.5\times$ speedup while maintaining high success rates, demonstrating that its efficiency gain is not attributable to merely reducing denoising steps.
>
> ---
>
>
> **Q3. Visualization of Cache Behavior As a visual learner.**
> > Visualization of Cache Behavior As a visual learner, I would find it greatly helpful to see a figure illustrating the cache's effect—for instance, an attention map or feature heatmap at a step where the cache decided not to update. Such a visualization would intuitively convey the underlying "similarity" concept without relying solely on mathematical formulations.
>
> **A3.** Thank you for this helpful suggestion. We have directly visualized how the cache behaves at each denoising step in Appendix A.14.
> Specifically, we visualize the ground-truth features, the computed and cached features under the BAC schedule, and their absolute differences for each timestep in Figures 15 through 24, with cache update steps marked in red. Across all tasks, we observe two consistent phenomena.
> First, consecutive steps exhibit high feature similarity, confirming that high temporal redundancy makes caching naturally applicable. Second, the difference maps reveal distinct behaviors between reuse and update phases: during cache reuse, the absolute difference remains low, reflecting activation stability. Conversely, cache update steps show significant feature shifts, indicating that updates are effectively capturing necessary changes.

---

> ### Author Response · Authors · 2025-11-26
> **Response to Reviewer p5ox (Part 4)**
>
> **Q4. Hyperparameter Tuning for New Tasks.**
> > Hyperparameter Tuning for New Tasks The paper mentions using S = 7 or 10. If a user applies the method to a new robot task, are there practical guidelines—such as "start with S = 8"—to avoid extensive hyperparameter search? A brief recommendation in the paper would greatly improve usability.
>
> **A4.** We appreciate the suggestion regarding usability. Empirically, we find that $S \approx 20$ serves as a robust starting point that consistently yields lossless acceleration across varying tasks. From there, users can employ binary search to find the acceleration limit until a performance drop is observed, and then incrementally increase the steps to recover performance.
>
>
>
>
> ---
>
> **Q5. Code Accessibility.**
>
> > Code Accessibility. The commitment to open-source the code is appreciated. Would it be possible to share a GitHub link during the rebuttal period—even if kept private temporarily—so that reviewers can examine the license and potentially run a simple local example?
>
> **A5.** Thank you for your positive feedback on our commitment to open-sourcing the code. We are happy to provide code at https://anonymous.4open.science/r/Block-wise_Adaptive_Caching-1265/. The repository includes both the evaluation scripts for the simulation results and the real-world experimental results.

---

### Author Response · Authors · 2025-12-01
**Summary of Discussion Period**

We would like to express our deepest gratitude to the Area Chair and Program Chairs for their time and dedication in overseeing the review process. We also sincerely thank the reviewers for their insightful and constructive feedback, which has been invaluable in refining our work. To aid in the final assessment, we respectfully present a summary of the discussion period.

---

**Key Highlights of Reviews**

We are encouraged that **all reviewers reached a positive consensus on the quality of our submission**, unanimously giving ratings of 3 (Good) for Soundness, Presentation, and Contribution. Reviewers recognized our work as a “high-quality and broadly significant” contribution (Reviewer p5ox) with a “clear problem statement” (Reviewer p5ox). The method was praised for its “novel insight” (Reviewer p5ox) and the “original” discovery of the Error Surge phenomenon (Reviewers o36f, N17f). All reviewers highlighted its practicality, describing the approach as “training-free,” “plug-and-play” (Reviewer p5ox), and a “principled and sound approach” (Reviewer N17f) that represents “an important step toward making diffusion policies truly practical for real-world robotic control” (Reviewer o36f).

---
**Rebuttal Outcome and Consensus**

We are pleased to report that following our clarification and additional experiments, **Reviewer N17f has raised the score from 4 to 6**, noting that we have "addressed most of the questions". **This leads to a consensus of positive ratings across all reviewers (Scores: 6, 6, 6).** While Reviewers p5ox and o36f have not yet had the opportunity to provide further comments, we believe the comprehensive results and detailed analyses provided in our response have satisfactorily addressed their concerns.


---
**Major Revisions and Enhancements**

We have provided detailed responses and revisions to address the concerns raised by the reviewers. Below, we summarize the major improvements made in our revised manuscript:


+ **Real-World Robotic Validation (Reviewers p5ox, N17f)**: We have added real-world experiments in `Section 4.2`. BAC achieves 45.1 Hz and 63-71% success rate, significantly outperforming high-latency baselines. Demo videos are available on our [project page](https://block-wise-adaptive-caching.github.io).

+ **Extended Discussion on Error Surge (Reviewer o36f)**: We added `Appendix A.5` to extend the discussion on the theoretical analysis for the phenomenon. We demonstrate that FFN weight norms are $3\times$ larger than attention weights, identifying the structural reason why error surges occur in FFNs but not in attention blocks.

+ **Cache Behavior Visualizations (Reviewer p5ox):** We added visualizations of feature heatmaps reflecting cache behavior in `Appendix A.14` to intuitively demonstrate the "similarity" concept.

+ **Large-scale Evidence for Episode Homogeneity (Reviewer N17f):** We added large-scale evidence for episode homogeneity in `Appendix A.15`. Quantitative statistics (average similarity $\ge 0.96$) and qualitative heatmaps confirm that the homogeneity pattern remains stable across diverse episodes.

+ **Comparison with Fewer Denoising Steps (Reviewer p5ox):** We compared BAC against DDIM with fewer steps in `Appendix A.16`. Results show BAC yields an additional $2.2\times$ to $3.5\times$ speedup on top of DDIM while maintaining high success rates, confirming the advantages of fine-grained caching beyond simple step reduction.


+ **Code Accessibility (Reviewer p5ox):** We released the [anonymous source code](https://anonymous.4open.science/r/Block-wise_Adaptive_Caching-1265/), including comprehensive evaluation scripts and real-world deployment guidance, to ensure reproducibility.

---

Finally, we sincerely thank the AC and PCs again for their exceptional efforts in managing the OpenReview Incident and ensuring a fair and fruitful review process.

---

### Meta-Review · Area_Chair_hfaY · 2026-01-07

**Summary:**

This paper introduces Block-wise Adaptive Caching (BAC), a plug-and-play, training-free method designed to accelerate Diffusion Policy inference by up to 3x via intermediate feature caching. By analyzing the "error surge" phenomenon in FFN blocks, the authors propose a caching schedule that maintains high control precision. The submission is strengthened by a transition from purely simulated results to real-world robotic validation on a Franka arm, demonstrating that the method is practical for real-time control without retraining.

**Reviewer Concerns:**

Across reviewers, the primary concerns focused on the lack of physical experiments (p5ox, N17f), potential episode homogeneity affecting generalization (N17f), and comparisons with simple step reduction (p5ox). The rebuttal substantively addresses the validation gap by adding real-world deployment results (45.1 Hz inference) which satisfied both p5ox and N17f. To address generalization concerns, the authors provided large-scale quantitative analysis and similarity heatmaps (N17f). While o36f noted that the theoretical bound for error propagation remains a linear response formulation rather than a rigorous bound, they acknowledged this does not diminish the method's empirical value.

**Reviewer Scores:**

Given the unanimous positive consensus, N17f explicitly raised their score from 4 to 6 during the discussion. I expect p5ox to maintain their score as their requests for physical validation and code were fully met, and o36f likely stays at 6, viewing the theoretical limitation as a minor open point in an otherwise strong empirical paper.

---

### Decision · Program_Chairs · 2026-01-26

Accept (Poster)